# Evaluation of iterative Kalman smoother schemes for multi-decadal past climate analysis with comprehensive Earth system models

Javier García-Pintado [1] and André Paul [1]

[1]MARUM – Center for Marine environmental Sciences and Department of Geosciences, University of Bremen, Bremen, Germany

**Correspondence:** J. García-Pintado (jgarciapintado@marum.de)

**Abstract.** Paleoclimate reconstruction based on assimilation of proxy observations requires specification of the control variables and their background statistics. As opposed to numerical weather prediction (NWP), which is mostly an initial condition problem, the main source of error growth in deterministic Earth system models (ESMs) regarding the model low-frequency response comes from errors in other inputs: parameters for the small-scale physics, forcing and boundary conditions. Also,

comprehensive ESMs are nonlinear and as much only a few ensemble members can be run in current high performance computers. Under these conditions we evaluate two assimilation schemes, which (a) count on iterations to deal with nonlinearity, and (b) are based on low-dimensional control vectors to reduce the computational need. The practical implementation would assume that the ESM has been previously globally tuned with current observations, and that for a given situation there is a previous knowledge of the most sensitive inputs (given corresponding uncertainties), which should be selected as control variables.

The low dimension of the control vector allows for using full-rank covariances and resorting to finite difference sensitivities (FDS). The schemes are then a FDS implementation of the iterative Kalman smoother (FDS-IKS, a Gauss-Newton scheme), and a so-called FDS-multistep Kalman smoother (FDS-MKS, based on repeated assimilation of the observations). We describe the schemes and evaluate the analysis step for a data assimilation window in two numerical experiments: (a) a simple 1D energy balance model (Ebm1D; which has an adjoint code) with present-day surface air temperature from the NCEP/NCAR reanalysis

data as target, and (b) a multi-decadal synthetic case with the Community Earth System Model (CESM v1.2, with no adjoint). In the Ebm1D experiment, the FDS-IKS converges to the same parameters and cost function values than a 4D-Var scheme. For similar iterations than the FDS-IKS, the FDS-MKS results in slightly higher cost function values, which are however still substantially lower than those of an ensemble transform Kalman filter (ETKF). In the CESM experiment, we include an ETKF with Gaussian Anamorphosis (ETKF-GA) implementation as potential nonlinear assimilation alternative. For three iterations,

both FDS schemes get cost functions values that are close between them and (with about half the computational cost) lower than those of the ETKF and ETKF-GA (these with similar cost function values). Overall, the FDS-IKS seems more adequate for the problem, with the FDS-MKS potentially more useful to damp increments in early iterations of the FDS-IKS.

# 1 Introduction

Earth system models (ESMs) to simulate the Earth system and global climate are usually developed using the present and recent historical climates as references, but climate projections indicate that future climates will lie outside these conditions. Paleoclimates very different from these reference states therefore provide a way to assess whether the ESM sensitivity to forcings is compatible with the evidence given by paleoclimatic records (Kageyama et al., 2018). Coupled atmosphere-ocean general circulation models (AOGCMs) and comprehensive ESMs have enabled the paleoclimate community to gain insights into internally generated and externally forced variability, and to investigate climate dynamics, modes of variability (e.g.; Ortega et al., 2015; Zanchettin et al., 2015), and regional processes in detail (e.g.; PAGES 2k-PMIP3 group, 2015). However, AOGCMs and comprehensive ESMs demand high computational resources, which severely limits the length and number of affordable model integrations in current High Performance Computers (HPCs). Thus, the last millenium ensemble with the Community Earth System Model (CESM) (CESM-LME; Otto-Bliesner et al., 2016), which is still a considerable achievement, has only $m = 10$ members for the full-forcing transient simulations for the years 850–2005 in the Common Era. Also, the multi-model ensemble in the Paleoclimate Model Intercomparison Project (PMIP) for experiments contributing to the Coupled Model Intercomparison Project (CMIP, since CMIP6) relies on coherent modelling protocols followed by the paleoclimate modelling teams in independent HPCs (e.g.; Jungclaus et al., 2017). On the other hand, the gathering and analysis of existing and new paleoclimate proxy records to create multiproxy databases also relies on collective efforts focused on specific time spans as, for example, the *global multiproxy database for temperature reconstructions of the Common Era* (PAGES2K) (PAGES2k Consortium, 2017), or the *multiproxy approach for the reconstruction of the glacial ocean surface* (MARGO) (MARGO Project Members, 2009), which focuses on the Last Glacial Maximum (LGM), a period between 23 and 19 thousand years before present (BP). The quantitative fusion of comprehensive ESMs and paleoclimate proxy observations should provide deeper insight into past climate low-frequency variability, to which (here and throughout the article) we refer to variability on times scales of 30–50 years or longer (e.g.; as Christiansen and Ljungqvist, 2017). However, this fusion is hampered by the high computational demand of AOCGMs and comprehensive ESMs.

The issue of fusing data into models arises in scientific areas that enjoy a profusion of data and use costly models. In the geophysical community this is referred to as inverse methods and data assimilation (DA), whose aim is finding the best estimate of the state (the *analysis*), by combining information from the observations and from the numerical and theoretical knowledge of the underlying governing dynamical laws. Most known DA methods stem from Bayes' theorem (Lorenc, 1986), and each is made practical by making approximations (Bannister, 2017). In numerical weather prediction (NWP) the assimilation is mostly an initial condition problem. In contrast, the climate of a sufficiently long trajectory is typically much less sensitive to initial conditions, being essentially a sample of the underlying true model climate contaminated by a small amount of deterministic noise due to the finite integration interval (Annan et al., 2005b). The low-frequency errors in deterministic ESMs are so mostly dependent on model errors, including the parameters for the small-scale physics, and errors in forcings and boundary conditions.

DA has been used as a technique for low-frequency past climate field reconstruction (CFR), with real case studies as, e.g., assimilation of marine sediment proxies of sea surface temperature (SST) in a regional ocean model of North Atlantic at the termination of the Younger Dryas (YD) cold interval (Marchal et al., 2016), and synthetic studies, as assimilation of tree-ring-width into an atmospheric GCM (AGCM) (Acevedo et al., 2017), analysis of time-averaged observations (Dirren and

Hakim, 2005), or evaluation of particle filters for paleodata assimilation (Dubinkina et al., 2011). Including model parameters as control variables, early work in climate analysis was done by Hargreaves and Annan (2002), who evaluated a Monte Carlo Markov Chain (MCMC) method with a simple ESM. Later, the technique of state augmentation with model parameters (e.g.; Friedland, 1969; Smith et al., 2011) and an ensemble Kalman filter (EnKF) (Evensen, 1994) was used by Annan et al. (2005a) and Annan et al. (2005b) in synthetic experiments with an Earth system model of intermediate complexity (EMIC) and with

an AGCM coupled to a slab ocean, respectively. The additional issue of sparsity in paleoclimate proxies was addressed by Paul and Schäfer-Neth (2005) for climate field reconstructions with an EMIC and manual tuning. More recent applied work, in part motivated by the nonlinearity of climate models, has used four dimensional variational DA (4D-Var). Thus, Paul and Losch (2012) applied 4D-Var with a conceptual climate model, and Kurahashi-Nakamura et al. (2017) used 4D-Var with the Massachusetts Institute of Technology general circulation model (MITcgm) for ocean state estimation considering joint initial

conditions, atmospheric forcings, and an ocean vertical diffusion coefficient as control variables to analyse the global ocean state in equilibrium conditions during the Last Glacial Maximum (LGM). We share the motivation of this recent work but put the focus on deterministic and comprehensive ESMs. As, in general, these models are not suited to automatic differentiation (AD) and the development of hand-coded tangent linear and adjoint models is out of reach (so, standard 4D-Var and related hybrid approaches as En4DVar are not applicable), we seek assimilation strategies that take into account the nonlinearity in

ESMs and the computational constraints with current HPCs for low-frequency analysis.

The question remains about how should one choose the control vector for the assimilation. One possibility is to resort to ensemble methods, which involve a low-rank representation of covariances. For example, the (adjoint-free) iterative ensemble Kalman smoother (IEnKS) in Bocquet and Sakov (2014), counts on iterations to deal with nonlinearity. Also, the IEnKS has been evaluated in a synthetic study with the low-order model Lorenz-95 by state augmentation (Bocquet and Sakov, 2013). In

principle, this could allow for selecting a relatively high-dimensional control vector and include model parameters. However, the ensemble (low-rank) covariances and sensitivities would be noisy because of the small ensemble size. An alternative is to resort to low-dimensional control vectors, so that the covariance of the control vector is explicitly represented and the low dimension allows for an estimation of the sensitivity of the observation space to the control vector by conducting individual perturbation experiments of the control variables; i.e., a finite difference sensitivity (FDS) estimation. With respect to adjoint

sensitivities, FDS has the disadvantages that the computing cost is proportional to the dimension of the state vector and that the choice of the perturbations is critical. Too small perturbations lead to numerical errors, while too high perturbations lead to truncation errors. An advantage is that FDS consider the full physics of the nonlinear model.

In any case, in a practical application of such a low-dimensional control vector approach (whose dimension would be imposed by computational constraints), the selection of the control variables should be carefully done. From all available

model inputs, the selected control variables (given their respective background uncertainties) and model should try to explain

the most of the observed variability. In turn, this assumes (a) a general need to perform sensitivity analysis beforehand, and (b) that the model has been previously comprehensively tuned. The exclusion of relatively less sensitive inputs from the control vector, and the previous tuning would reduce possible compensation effects (i.e., that increments in the control vector due to the assimilation take partial responsibility for errors elsewhere). Nonetheless, some error compensation will alway

be present (for example, this is intrinsic to the common tuning of the coupled ESM, which follows tuning of individual components) and very difficult to deal with. A striking example is given by Palmer and Weisheimer (2011), who report how an inadequate representation of horizontal baroclinic fluxes resulted in a climate model error equal in magnitude and opposite to the systematic error caused by insufficiently represented vertical orographic wave fluxes. Thus, the selected control vector has the responsibility of embracing the model climate background uncertainty and their updated values would possibly compensate

for non-accounted errors. For low-frequency climate analysis it is very likely that after some years of model integration, the modelled climate is less sensitive to (reasonable) initial conditions than to other possible inputs. Thus, one would generally select the the most sensitive parameters for the model physics and forward operators, forcings and boundary conditions for a given situation as control vector.

Also, regarding initial conditions in a sequence of multi-decadal and longer data assimilation windows with transient forc-

ings, there is no clear consensus about how should one approach the initialization at each DAW. For example, Holland et al. (2013) indicate that inizialization had little little impact (in general, limited to a couple of years) on Artic and Antartic sea ice predictability in the Community Climate System Model 3 (CCSM3) in a perfect-model framework. However, in a later synthetic study including assimilation with a perfect-model framework and an Earth system model of intermediate complexity (EMIC), Zunz et al. (2015) obtained a similar interannual predictability ($\sim$3 years), but noted that the initialization for the

DAW can still influence the state at multi-decadal timescales (although with a larger impact of external forcing). Among others, we mention these examples as sea ice has been related to changes in atmospheric circulation patterns and teleconnections with the tropical Pacific and Atlantic oceans (e.g.; Meehl et al., 2016; Marchi et al., 2018), and these relatively fast climate dynamics can, e.g., influence the onset or termination of glacial conditions and stronger climate changes. On the other hand, given the limited predictability at multidecadal timescales (and to reduce computational costs), the reinitialization in paleo-DA

has often be removed after the assimilation altogether, and a common initial integration is applied as background climate for a number of DAWs. This has been named as *off-line* assimilation (e.g.; Steiger et al., 2014; Hakim et al., 2016; Klein and Goosse, 2017; Acevedo et al., 2017). The perspective of the *off-line* approach can be modified when model parameters are included in the control vector because, as opposed to initial conditions, the impact of model parameters does not decay with the model integration. In general, updated model parameters as part of the assimilation as well as their physically consistent

climate would then be used as (augmented) initial conditions for a subsequent DAW.

Throughout this study, all observations available during a data assimilation window (DAW) are assimilated in parallel (e.g., as Sakov et al., 2010). This has been termed four-dimensional data asimilation or asynchronous data assimilation, and is also commonly referred to as the smoothing problem (Sakov et al., 2010). Here we choose the term *smoother* for the evaluated schemes, but they could as well be termed asynchronous (or four-dimensional) *filters*. This study focuses on evaluating two

assimilation schemes for low-frequency past climate reconstruction. They are based on finite difference sensitivities (FDS) and

low-dimensional control vectors, and rely on iterations to account for nonlinearity. The schemes are then a FDS implementation of the iterative Kalman smoother (FDS-IKS, a Gauss-Newton scheme), and an alternative named as FDS-multistep Kalman smoother (FDS-MKS, based on repeated assimilation of the observations). Other paleoclimate assimilation issues, as the sparsity and measurement error characteristics (e.g.; temporal autocorrelation), and the representation error, or the development of

forward operators for specific proxies (*proxy system models*) (e.g.; Evans et al., 2013; Dee et al., 2016) and further complexities in the model-observation comparison (e.g.; Goosse, 2016; Weitzel et al., 2018) are not addresed in this manuscript.

The rest of this article is organised as follows. In section 2, within the broader context of the joint state-parameter estimation problem, we summarise the strong constraint incremental 4D-Var formulation (Courtier et al., 1994) from a perspective where the state vector is augmented with the model parameters to arrive, under given assumptions, to the formulation of the increments

in the iterative Kalman smoother (IKS) scheme as implemented here. Then we describe the sensitivity estimation and the two schemes, the FDS-IKS and the FDS-MKS in a concise, algorithmic format. The description of our implementation of the Gaussian Anamorphosis (GA), which we applied along with an ensemble transform Kalman filter (ETKF-GA) as alternative nonlinear assimilation approach, is also included in section 2. In section 3 we conduct an experiment with a simple 1D energy balance model (Ebm1D) and present day NCEP/NCAR reanalysis surface air temperature as target, and in section 4 we conduct

an identical twin experiment with CESM, assimilating MARGO-like data (MARGO uncertainties, timescales and locations) as example of paleoclimate observing dataset. Ebm1D is subject to automatic differenciation, so that we applied 4D-Var and ETKF as benchmark schemes. CESM lacks an adjoint, and we applied ETKF and ETKF-GA as benchmark schemes. The experimental setup, results and discussions are given in each case. We finish with conclusions in section 5.

## 2  Assimilation schemes

### 2.1  Analysis approach

The problem is to estimate the mean state (seasonal and annual means) of a past climate state along a time window for multidecadal and longer time scales. From a variational perspective, in numerical weather prediction (NWP) this would be referred to as a four-dimensional variational data assimilation (4D-Var) problem, where the initial conditions of a model integration are estimated subject to model dynamics and according to background and observation uncertainties within a data

assimilation window (DAW). In NWP, the background (or *prior*) is normally given by a previous model forecast. In this article, time $t_k$ and its index $k$ measure time relative to the start of the DAW, which is $t_0$, using conventions similar to those of 4D-Var. We consider a discrete nonlinear time invariant dynamical system model where $\mathbf{x}_k \in \mathbb{R}^n$ is the state vector at time $t_k$, and $\boldsymbol{\theta}_k \in \mathbb{R}^q$ is a vector of selected inputs in addition to initial conditions (model parameters, forcings and boundary conditions). We assume a gridded system, where specification of the model state and other inputs at time $t_k$ uniquely determines the model

state at all future times, and that $\boldsymbol{\theta}_k$ is constant during the DAW (i.e., $\boldsymbol{\theta} \equiv \boldsymbol{\theta}_{k+1} = \boldsymbol{\theta}_k$). For example, within $\boldsymbol{\theta}$, one can include a constant error term (a bias to be estimated as part of the asimilation) in a prescribed transient radiative constituent (e.g.; a $CH_4$ time series).

Then, we consider an augmented state vector $\mathbf{z}$:

$$\mathbf{z} = \begin{bmatrix} \mathbf{x} \\ \boldsymbol{\theta} \end{bmatrix}, \tag{1}$$

and an augmented deterministic nonlinear dynamics operator $\mathcal{M} : \mathbb{R}^{n+q} \to \mathbb{R}^{n+q}$, such that

$$\mathbf{z}_{k+1} = \mathcal{M}(\mathbf{z}_k), \quad k = 0, 1, \ldots \tag{2}$$

Observations at time $t_k$ are represented by the vector $\mathbf{y}_k^o \in \mathbb{R}^{p_k}$ and related to the model state by

$$\mathbf{y}_k^o = \mathbf{y}_k^{\mathbf{z}} + \boldsymbol{\epsilon}_k \equiv \mathcal{H}_k(\mathbf{z}_k) + \boldsymbol{\epsilon}_k, \tag{3}$$

where $\mathcal{H}_k : \mathbb{R}^{n+q} \to \mathbb{R}^{p_k}$ is a deterministic nonlinear observation operator that maps from the augmented state $\mathbf{z}_k$ to the observation space, and $\boldsymbol{\epsilon}_k \in \mathbb{R}^{p_k}$ is a realisation of a noise process, which consists of mesurement errors and representation errors (errors due to unresolved scales and processes, observation-operator errors, and pre-processing or quality-control errors; Janjić et al., 2017). We assume $\boldsymbol{\epsilon}_k$ is a Gaussian variable with mean $\mathbf{0}$ and covariance matrix $\mathbf{R}_k$. The error covariance matrix of a

state $\mathbf{z}_k = [\mathbf{x}_k^T, \boldsymbol{\theta}^T]^T$, where the superscript "T" denotes matrix transposition, at any time $t_k$ within the DAW is

$$\mathbf{P}_k = \begin{bmatrix} \mathbf{P}_{\mathbf{x}\mathbf{x}_k} & \mathbf{P}_{\mathbf{x}\boldsymbol{\theta}_k} \\ (\mathbf{P}_{\mathbf{x}\boldsymbol{\theta}_k})^T & \mathbf{P}_{\boldsymbol{\theta}\boldsymbol{\theta}} \end{bmatrix}, \tag{4}$$

where $\mathbf{P}_{\mathbf{x}\mathbf{x}_k} \in \mathbb{R}^{n \times n}$ is the error covariance matrix of $\mathbf{x}_k$, $\mathbf{P}_{\boldsymbol{\theta}\boldsymbol{\theta}} \in \mathbb{R}^{q \times q}$ is the error covariance matrix of $\boldsymbol{\theta}$, and $\mathbf{P}_{\mathbf{x}\boldsymbol{\theta}_k} \in \mathbb{R}^{n \times q}$ is the error covariance between $\mathbf{x}_k$ and $\boldsymbol{\theta}$. The goal in 4D-Var is then to find the initial state $\mathbf{z}_0$ that minimizes a non-quadratic cost function given by

$$\mathcal{J}_1(\mathbf{z}_0) = \frac{1}{2}\|\mathbf{z}_0 - \mathbf{z}_0^b\|_{\mathbf{P}_0^{-1}} + \frac{1}{2}\|\hat{\mathbf{y}}^o - \hat{\mathcal{H}}(\mathbf{z}_0)\|_{\hat{\mathbf{R}}^{-1}}, \tag{5}$$

where $\|\mathbf{a}\|_{\mathbf{A}^{-1}}^2 \equiv \mathbf{a}^T \mathbf{A}^{-1} \mathbf{a}$. The first term (the background term, $\mathcal{J}_b$) measures the deviation between $\mathbf{z}_0^b$ and $\mathbf{z}_0$, with the background-error covariance matrix $\mathbf{P}_0$ as $L_2$ norm. The second term (the observation term, $\mathcal{J}_o$) measures the deviation between $\hat{\mathbf{y}}^o \in \mathbb{R}^p$ (where $p = \sum_{k=0}^{n_k} p_k$, indicating all observations throughout the DAW) and its model equivalent $\hat{\mathbf{y}}^{\mathbf{z}} \equiv \hat{\mathcal{H}}(\mathbf{z}_0)$, using the observation-error covariance matrix $\hat{\mathbf{R}}$ as $L^2$ norm. $\hat{\mathcal{H}}(\mathbf{z}_0) : \mathbb{R}^{n+q} \to \mathbb{R}^p$ is a generalised observation operator maping

from the augmented initial state to all the observations at any time in the DAW (i.e., $\hat{\mathcal{H}} \equiv \mathcal{H} \circ \mathcal{M}$). The maximum *a posteriori* estimation in (5) is also known as conditional mode estimation, or maximum of the conditional density. As presented by Lorenc (1986) from a Bayesian view, this is the maximum likelihood of the state under a Gaussian assumption for the various error terms. The cost function (5) is subject to the states satisfying the nonlinear dynamical system (2) and is known as *strong* constraint variational formulation, while the additional inclusion of a term for the model error would lead to a *weak* constraint

4D-Var. The solution to the functional $\mathcal{J}_1(\mathbf{z}_0)$ is $\mathbf{z}_0^a$, with the resulting states along the DAW referred to as the *analysis*.

In general, an exact solution cannot be found. In the incremental formulation of 4D-Var, the solution to (5) is approximated by a sequence of minimizations of quadratic cost functions. Thus, incremental 4D-Var has first an outer loop, where $\mathbf{z}_0^l$ provides the

current approximation and initially, for $l = 1$, $\mathbf{z}_0^1 = \mathbf{z}_0^b$. The *innovations* are then given by the residual between the observations and the mapping of the initial state in the current approximation into observation space

$$\delta\hat{\mathbf{y}}^l = \hat{\mathbf{y}} - \hat{\mathcal{H}}(\mathbf{z}_0^l), \tag{6}$$

where the computation of the initial state mapped to observation space, $\hat{\mathcal{H}}(\mathbf{z}_0^l)$, has the following way:

$$\left[\hat{\mathcal{H}}(\mathbf{z}_0^l)\right]_k = \mathcal{H}_k\left[\mathcal{M}(\mathbf{z}_0^l, t_0, t_k)\right] = \mathcal{H}_k(\mathbf{z}_k^l). \tag{7}$$

Then, incremental 4D-Var has an inner loop, where two approximations are conducted. First

$$\mathcal{H}_k(\mathbf{z}_k) - \mathcal{H}_k(\mathbf{z}_k^l) \approx \mathbf{H}_k(\mathbf{z}_k^l)(\mathbf{z}_k - \mathbf{z}_k^l), \tag{8}$$

where $\mathbf{H}_k(\mathbf{z}_k^l)$ is the Jacobian of $\mathcal{H}_k(\bullet)$ evaluated at $\mathbf{z}_k^l$. $\mathbf{H}_k$ is referred to as the *tangent linear operator* in the DA literature. Second, the dynamical model is also linearized, obtaining the *tangent linear model* (TLM):

$$\mathbf{M}_{0:k} = \frac{\partial\mathcal{M}(\mathbf{z}_0^l, t_0, t_k)}{\partial\mathbf{z}_0}, \tag{9}$$

so that $\hat{\mathbf{H}} = [(\mathbf{H}_0)^T, (\mathbf{H}_1\mathbf{M}_{0:1})^T, \ldots (\mathbf{H}_1\mathbf{M}_{0:n_k})^T]^T$, leading to the generalized linearization

$$\hat{\mathcal{H}}(\mathbf{z}_0) - \hat{\mathcal{H}}(\mathbf{z}_0^l) \approx \hat{\mathbf{H}}(\mathbf{z}_0^l)\delta\mathbf{z}_0, \tag{10}$$

where $\delta\mathbf{z}_0 = \mathbf{z}_0 - \mathbf{z}_0^l$ is the increment. By considering (6) and (10), the generalised error term $\hat{\boldsymbol{\epsilon}}$ in (5) for all observations in the DAW can be expressed as

$$\begin{aligned}\hat{\boldsymbol{\epsilon}} &= \hat{\mathbf{y}}^o - \hat{\mathcal{H}}(\mathbf{z}_0) \\ &= \delta\hat{\mathbf{y}}^l - \hat{\mathcal{H}}(\mathbf{z}_0) + \hat{\mathcal{H}}(\mathbf{z}_0^l) \\ &\approx \delta\hat{\mathbf{y}}^l - \hat{\mathbf{H}}^l\delta\mathbf{z}_0, \end{aligned} \tag{11}$$

where $\hat{\mathbf{H}}^l \equiv \hat{\mathbf{H}}(\mathbf{z}_0^l)$. This approximation of $\hat{\boldsymbol{\epsilon}}$ is introduced in (5) leading to a quadratic cost function with the increment $\delta\mathbf{z}_0$ as argument:

$$\mathcal{J}_2(\delta\mathbf{z}_0) = \frac{1}{2}\|\delta\mathbf{z}_0 - (\mathbf{z}_0^b - \mathbf{z}_0^l)\|_{\mathbf{P}_0^{-1}} + \frac{1}{2}\|\delta\hat{\mathbf{y}}^l - \hat{\mathbf{H}}^l\delta\mathbf{z}_0\|_{\hat{\mathbf{R}}^{-1}}, \tag{12}$$

    The minimizations of $\mathcal{J}_2(\delta\mathbf{z}_0)$ is the inner loop, which is conducted by gradient descent algorithms (e.g.; Lawless, 2013)
until it meets a given criterion, yielding an optimal $\delta\mathbf{z}_0^{l+1}$. Then, the outer loop takes control, where the estimate of the initial state is updated with the estimated increment: $\mathbf{z}_0^{l+1} = \mathbf{z}_0^l + \delta\mathbf{z}_0^{l+1}$. Incremental 4D-Var has been shown to be an inexact Gauss-Newton method applied to the original nonlinear cost function (Lawless et al., 2005).

    In our context in this paper, we assume Gaussian statistics and a perfect-model framework except for the sources of model uncertainty in $\mathbf{z}_0$. Thus, the conditional mode given by minimization of Eq. (12) is also the conditional mean (also called the
*minimum variance* estimate) given by the explicit solution

$$\delta\mathbf{z}_0^{l+1} = \mathbf{z}_0^b - \mathbf{z}_0^l + \mathbf{K}^l[\delta\hat{\mathbf{y}}^l - \hat{\mathbf{H}}^l(\mathbf{z}_0^b - \mathbf{z}_0^l)], \tag{13}$$

where $\mathbf{K}^l$ is known as the Kalman gain matrix, given by

$$\mathbf{K}^l = \mathbf{P}_0(\hat{\mathbf{H}}^l)^{\mathrm{T}}[\hat{\mathbf{H}}^l\mathbf{P}_0(\hat{\mathbf{H}}^l)^{\mathrm{T}} + \hat{\mathbf{R}}]^{-1}. \tag{14}$$

So, the inner loop is omitted and the state vector is explicitly updated as

$$\begin{aligned} \mathbf{z}_0^{l+1} &= \mathbf{z}_0^l + \delta\mathbf{z}_0^{l+1} \\ &= \mathbf{z}_0^{\mathrm{b}} + \mathbf{K}^l(\hat{\mathbf{y}}^{\mathrm{o}} - \hat{\mathcal{H}}(\mathbf{z}_0^l) - \hat{\mathbf{H}}^l(\mathbf{z}_0^{\mathrm{b}} - \mathbf{z}_0^l)). \end{aligned} \tag{15}$$

Thus, as incremental 4D-Var, the iterative approach described by Eq. (15) gives an approximation to the conditional mode or
maximum likelihood of the cost function (5). Iterative methods have a long history for DA applications in nonlinear systems.
Jazwinski (1970) considers local (conducted over a single assimilation cycle) and global (conducted over several assimilation
cycles) iterations of the extended Kalman filter (EKF). Local iterations of the Kalman filters are designed to deal with nonlinear
observation operators and non-Gaussian errors. The locally iterative (extended) Kalman filter (IKF) is a Gauss-Newton method
for approximating a maximum-likelihood estimate (Bell and Cathey, 1993), and actually it is algebraically equivalent to nonlin-
ear three dimensional variational (3D-Var) analysis algorithms (Cohn, 1997). For the first loop, the IKF is identical to an EKF
(see e.g.; Jazwinski, 1970). Later, Bell (1994) showed that the iterative Kalman smoother (IKS) represents a Gauss-Newton
method to obtain an approximate maximum likelihood, as was shown later for incremental 4D-Var (Lawless et al., 2005). The
IKF and the IKS circumvent the need for choosing a step size, which is sometimes a source of difficulty in descent methods.
However, as Gauss-Newton methods, not even local convergence is guaranteed. Eq. (15) is actually akin to the formulation of
the IKF, but generalized to a DAW, and it is so an IKS. It differs, though, from the IKS formulation in (Bell, 1994), in that (15)
is a strong constraint version (cost term for the model neglected) without the backward pass (see Fig. (1) in Bell, 1994).

Now, there remains to see how one would apply a scheme as Eq. (15) for multidecadal and longer term paleoclimate analysis.
In the last years there is a growing effort toward development of stochastic physical parameterizations in weather forecast
and climate models. However, stochastic parameterization is still in its infancy in comprehensive ESMs. For deterministic
parameterizations, on which we focus here, the model climate converges to its own dynamical attractor, and the climate of a
sufficiently long model trajectory is typically much less sensitive to initial conditions than to other model inputs.

In general, ensemble methods rely on perturbations $\delta\mathbf{z}_0$ of the control vector to estimate the sensitivity matrix $\hat{\mathbf{H}}^l$. Two
general kind of simulations and climate analysis are of interest to the paleoclimate community; the so-called *equilibrium*
simulations, and the transient ones. Equilibrium simulations are subject to solar forcing prescribed to a specific calendar year
and fixed radiative constituents in the atmosphere, representing a situation when the past climate is considered stable. The
goal is to evaluate the low-frequency seasonal and annual means and variability within these relative long-term stable climate
conditions (e.g.; Eeemian, LGM or mid-Holocene). In these simulations, the model is integrated until it reaches equilibrium
conditions. Typically, starting from a standard climatology from the World Ocean Atlas, it takes a few thousand years for an
ESM integration to converge to its equilibrium (even in the deepest ocean) for stationary forcings. After an initial spin-up
similar to the equilibrium conditions, transient simulations with EMSs use then the corresponding time-varying solar forcing
and normally use prescribed time series of radiative atmospheric constituents reconstructed from observations for the time

window of interest, as well as transient boundary conditions. From an assimilation perspective, irrespective of whether the analysis is for equilibrium or transient forcing, the perturbation of model parameters and other inputs introduces a shock at the start of the model integration (in some way, analogous to shocks in unbalanced ocean-atmosphere coupled models initialized with uncoupled data assimilation in NWP). And further than the initial oscillations, the model climatology in the first years of model integration is not consistent with the perturbed parameters. Thus, the estimation of the sensitivities $\hat{\mathbf{H}}^l$ can be spurious

in the first years of model integration, and also the innovations do not result from the model climatology. It is convenient then to set up a model integration time threshold and to disregard sensitivities earlier than this time. For such purpose, we loosely define a model *quasi-equilibrium* condition as the situation in which a model climatology is in reasonable physical consistency with the control vector (initial conditions plus model parameters, etc.). With equilibrium simulations a *quasi-equilibrium* time $t_q$ can, for example, be evaluated based on the convergence on the Meridional overturning circulation maximum value,

where each ensemble member will converge towards its own attractor. By then, the correlations between the atmosphere and surface climate, and ocean mixed layer and the observation space should be fully developed. Then, low-frequency climate means (annual and/or seasonal) after the integration time to *quasi-equilibrium*, $t_q$, would be evaluated against the climate proxy database (e.g.; MARGO for the mean annual and seasonal climate across the LGM) to obtain the sensitivity $\hat{\mathbf{H}}^l$ and the innovations $\delta \hat{\mathbf{y}}^l$ at each iteration. This is the approach followed by the experiments in this study.

For the more general transient forcing situation, in a current DAW, the effects from a perturbed control vector and from transient forcing are entangled. As opposed to the equilibrium simulations, integration times here match physical forcing times. Thus, also observations earlier than a specified $t_q$ should be better disregarded. Note that by *quasi-equilibrium* here, we do not refer to a model state, which will be transient as the forcings, but (as before) to a situation in which the model state is physically consistent with all the input given in the control vector. One could prescribe the forcings and boundary conditions

to those at the start of the integration time (the DAW), and use a $t_q$ estimated from these equilibrium forcings as surrogate for the transient $t_q$. However, this would increase computations. Alternatively, one can set up a $t_q$ based on previous experience and tests for equilibrium forcings.

In both cases, it is unlikely that errors in initial conditions are among the most sensitive errors out of all possible inputs for the evaluated integration times after *quasi-equilibrium*. Thus, for low-frequency past climate analysis, it should be generally

aceptable to exclude $\mathbf{x}_0$ from the control variables, leading to a reduced problem. To simplify the notation, we then define $\mathcal{G}(\bullet)$ as a generalised (deterministic) observation operator mapping a vector $\boldsymbol{\theta}$ into the observation space; $\mathcal{G} : \mathbb{R}^q \mapsto \mathbb{R}^p$, which follows

$$[\mathcal{G}(\boldsymbol{\theta})]_k = \left[ \hat{\mathcal{H}}(\mathbf{z}_0) \right]_k \Bigg|_{t_k \geq t_q}, \tag{16}$$

where $t_q$ represents the model integration time to *quasi-equilibrium*. Instead of (5), a reduced problem is posed now by mini-

mization of the non-quadratic cost function

$$\mathcal{J}(\boldsymbol{\theta}) = \frac{1}{2} \|\boldsymbol{\theta} - \boldsymbol{\theta}^{\mathrm{b}}\|_{\mathbf{P}_{\boldsymbol{\theta}\boldsymbol{\theta}}^{-1}} + \frac{1}{2} \|\hat{\mathbf{y}}^{\mathrm{o}} - \mathcal{G}(\boldsymbol{\theta})\|_{\mathbf{R}^{-1}}. \tag{17}$$

After the assimilation, a forward integration with the updated $\boldsymbol{\theta}^{\mathrm{a}}$ leads to its physically consistent climate estimate. The sensitivity matrix, or Jacobian of $\mathcal{G}$, is noted as $\mathbf{G} \in \mathbb{R}^{p \times q}$. The trivial substitutions into the incremental formulation (12) and its solution (15), with estimation of $\mathbf{G}$ via finite differences, lead to the finite difference sensitivity iterative Kalman smoother (FDS-IKS), which is summarised in section 2.3. The finite difference sensitivity multistep Kalman smoother (FDS-MKS), described in 2.4 is an alternative approach to deal with nonlinearity.

## 2.2 Background error covariances and sensitivity estimation

The current implementation of variational assimilation (with atmospheric models) is now different in each operational NWP center. A recent review of operational methods of variational and ensemble-variational data assimilation is given by Bannister (2017). However, for many geophysical models, codes are not suited to automatic differentiation and it is extremely complex to develop and maintain tangent linear and adjoint codes. This has motivated more recent research toward (adjoint-free) ensemble DA methods for high-dimensional models. Lorenc (2013) recommended to use the term 'EnVar' for variational methods using ensemble covariances. Thus, Liu et al. (2008) proposed (the later called) 4DEnVar, as a way to estimate the generalised sensitivities of the observation space to initial conditions based on an ensemble of model integrations within a 4D-Var formulation. Gu and Oliver (2007) introduced a scheme called Ensemble Randomized Maximum Likelihood Filter (EnRML) for on-line nonlinear parameter estimation, which was later adapted as a smoother, the batch-EnRML, by Chen and Oliver (2012). The EnRML estimates sensitivities by multivariate linear regression between the ensemble model equivalent of the observations and the ensemble perturbations to the control vector. These are so mean ensemble sensitivities. The iterative EnKF (IEnKF) is similar to the EnRML, but uses instead and ensemble square root filter, and rescales the ensemble perturbation (deflated) before and (inflated) after propagation of the ensemble as alternative to estimate the sensitivities (Sakov et al., 2012). In this way the estimated sensitivites are more local about the current estimation. An extension to the IEnKF as a fixed-lag smoother led to the IEnKS in Bocquet and Sakov (2014). However, while in the 4D-Var approach the computing cost of the sensitivites with the adjoint method is independent of the dimension of the control vector in the cost function, in numerical estimates of the sensitivities the computing cost increases with the size of the control vector. The low-rank property of the ensemble covariances in ensemble methods means that sampling error problems will inevitably arise when the number of ensemble members is small in comparison with the size of the control vector. Instead, as indicated in above, in the low-dimensional control vector schemes evaluated here, the background-error covariance is a full-rank explicit matrix.

While the relation between $\mathbf{G}$ and $\mathbf{P}_k$ is implied in the previous section, it is instructive to look at it in some detail. Let us consider the case of a specific observation time $t_k$. The Kalman gain matrix (disregarding the loop index, if any) for the components of the model parameters, which we denote in this section as $\mathbf{K}_{k\boldsymbol{\theta}}$, can be expressed in the two following ways:

$$
\begin{aligned}
\mathbf{K}_{k\boldsymbol{\theta}} &= \mathbf{P}_k \mathbf{H}_k^{\mathrm{T}} [\mathbf{H}_k \mathbf{P}_k (\mathbf{H}_k)^{\mathrm{T}} + \mathbf{R}_k]^{-1} \\
&= \mathbf{P}_{\boldsymbol{\theta}\boldsymbol{\theta}} \mathbf{G}_k^{\mathrm{T}} [\mathbf{G}_k \mathbf{P}_{\boldsymbol{\theta}\boldsymbol{\theta}} (\mathbf{G}_k)^{\mathrm{T}} + \mathbf{R}_k]^{-1},
\end{aligned}
\tag{18}
$$

where the first way is the standard one in Kalman smoother expressions including parameter estimation via state vector augmentation, and the second one is a parameter space formulation, which we apply here. Both are equivalent, but the covariance

information in $\mathbf{P}_k$ has been transferred to $\mathbf{G}_k$, or sensitivity matrix, in the second expression. Let us further consider the case that at $t_k$ there is a single observation $y$ of a state variable within the vector $\mathbf{z}_k$, denoted as $\mathbf{x}_{k_y}$, and we focus on the representer matrix for a single parameter $\boldsymbol{\theta}_i$. The covariance between $\boldsymbol{\theta}_i$ and the observation $y$ is expressed in both cases as

$$
(\mathbf{P}_k \mathbf{H}_k^{\mathrm{T}})[\boldsymbol{\theta}_i, y] = \sigma_{\mathbf{x}_{k_y} \boldsymbol{\theta}_i} \frac{\partial y}{\partial \mathbf{x}_{k_y}},
$$

$$
(\mathbf{P}_{\boldsymbol{\theta\theta}} \mathbf{G}_k^{\mathrm{T}})[\boldsymbol{\theta}_i, y] = \sum_{j=1}^{q} \sigma_{\boldsymbol{\theta}_j \boldsymbol{\theta}_i} \frac{\partial y}{\partial \boldsymbol{\theta}_j}, \tag{19}
$$

which, as $\frac{\partial y}{\partial \boldsymbol{\theta}_j} = \frac{\partial y}{\partial \mathbf{x}_{k_y}} \frac{\partial \mathbf{x}_{k_y}}{\partial \boldsymbol{\theta}_j}$, indicates that the linear equality

$$
\quad \sigma_{\mathbf{x}_{k_y} \boldsymbol{\theta}_i} = \sum_{j=1}^{q} \sigma_{\boldsymbol{\theta}_j \boldsymbol{\theta}_i} \frac{\partial \mathbf{x}_{k_y}}{\partial \boldsymbol{\theta}_j}, \tag{20}
$$

is taken from a bottom-up approach in the parameter space formulation, where all sources of uncertainty are specifically evaluated to compose the covariance $\sigma_{\mathbf{x}_{k_y} \boldsymbol{\theta}_i}$. In our experiments with ETKF and ETKF-GA, with parameter augmentation, the first alternative in Eq. (19) is used, and the generalized sensitivity matrix $\mathbf{G}$ is not explicitly computed. So, for comparison with the finite differences sensitivity (FDS) schemes, we estimate an ensemble-based average sensitivity matrix by solving

for $\bar{\mathbf{G}}$ in $\Delta \mathbf{Y} = \bar{\mathbf{G}} \Delta \boldsymbol{\theta}$, where $\Delta \boldsymbol{\theta} \in \mathbb{R}^{q \times m}$ is the matrix of random model parameter perturbations drawn from $\mathbf{P}_{\boldsymbol{\theta\theta}}$ around the background values, and $\Delta \mathbf{Y} \in \mathbb{R}^{p \times m}$ are the resulting perturbations in the observation space. In an iterative approach the sensitivities would need to be evaluated for perturbations around the current estimate.

Alternatively, finite differences sensitivity (FDS) directly samples from the conditional probability density function (pdf) of the perturbed variable, as the remaining control variables are kept to their current estimate. However, the computing require-

ments in FDS are linearly proportional to the size of the input vector, and its numerical estimation of derivatives is inaccurate and the associated error can be unacceptably large due to inadequate choice of the finite differencing step size. High perturbations increase the truncation error, which increases linearly with the perturbation magnitude, while as the magnitude of the perturbation gets smaller the accuracy of the differentiation degrades by the loss of computer precision (Dennis and Schnabel, 1996). It is possible to do more than one perturbation experiment by sampling from the conditional pdf for each parameter,

and estimating the sensitivity by univariate regression, which ameliorates the problem of non-optimal perturbations, and we include a few tests in this sense for the first experiment in this study (see Appendix A). It would be, however, computationally difficult to do more than one perturbation per control variable with comprehensive ESM and long integrations (and one might as well resort then to ensemble approaches). The sensitivity estimates by forward finite differences at a loop $l$ (initially, the background) are then computed at each column $\mathbf{G}_{:,i}^l$ as:

$$
\quad \mathbf{G}_{:,i}^l \approx \frac{\mathcal{G}(\boldsymbol{\theta}^l + \delta \boldsymbol{\theta}_i) - \mathcal{G}(\boldsymbol{\theta}^l)}{\delta \theta_i}, \ i = 1, \ldots, q, \tag{21}
$$

where for each parameter, $\delta \theta_i$ is a small perturbation (or variation) to the current approximation of $\theta_i$ and $\delta \boldsymbol{\theta}_i$ is the vector $\mathbf{0} \in \mathbb{R}^q$ but with element $\theta_i$ replaced by $\delta \theta_i$. As indicated, the estimation of sensitivities by local finite difference approximations results from sampling the conditional density function in the control vector space.

## 2.3 Finite difference sensitivity iterative Kalman smoother

The algorithm we describe here, denoted as *finite difference sensitivity iterative Kalman smoother* (FDS-IKS), is a Gauss-Newton scheme akin to the IKF and the IKS. The 'FDS' acronym to clarifies that the scheme is (a) expressed in terms of explicit sensitivities to all variables in the control vector, and (b) these local sensitivites are estimated numerically by individual perturbation experiments for each variable in the control vector. The scheme then uses a full-rank representation of the background-error covariance matrix (hence, it is not called an *ensemble* method). It is a smooother rather than a filter as it assimilates (future) observations along a DAW to update the control variables, applicable since the start of the DAW. After iterations are stopped (due to convergence criterion or reaching a maximum iteration number), a model reintegration of with the updated control vector $\boldsymbol{\theta}^{\mathrm{a}}$ leads to the analysis (or climate field reconstruction) over the DAW.

For any natural number $l$, the FDS-IKS provides the update

$$\boldsymbol{\theta}^{\mathrm{a}} = \boldsymbol{\theta}^l, \qquad \mathbf{P}^{\mathrm{a}}_{\boldsymbol{\theta}\boldsymbol{\theta}} = \mathbf{P}^l_{\boldsymbol{\theta}\boldsymbol{\theta}}.$$

The sequences $\{\boldsymbol{\theta}^l : l \geq 0\}$ and $\{\mathbf{P}^l_{\boldsymbol{\theta}\boldsymbol{\theta}} : l \geq 0\}$ are defined inductively as follows

$$\boldsymbol{\theta}^0 = \boldsymbol{\theta}^{\mathrm{b}}, \qquad \mathbf{P}^0_{\boldsymbol{\theta}\boldsymbol{\theta}} = \mathbf{P}^{\mathrm{b}}_{\boldsymbol{\theta}\boldsymbol{\theta}},$$

$$\boldsymbol{\theta}^{l+1} = \boldsymbol{\theta}^b + \mathbf{K}^l[\mathbf{y} - \mathcal{G}(\boldsymbol{\theta}^l) - \mathbf{G}^l(\boldsymbol{\theta}^{\mathrm{b}} - \boldsymbol{\theta}^l)], \tag{22}$$

$$\mathbf{P}^{l+1}_{\boldsymbol{\theta}\boldsymbol{\theta}} = (\mathbf{I} - \mathbf{K}^l\mathbf{G}^l)\mathbf{P}^{\mathrm{b}}_{\boldsymbol{\theta}\boldsymbol{\theta}}, \tag{23}$$

where for notational convenience

$$\mathbf{G}^l \equiv \mathbf{G}(\boldsymbol{\theta}^l), \tag{24}$$

and

$$\mathbf{K}^l = \mathbf{P}^{\mathrm{b}}_{\boldsymbol{\theta}\boldsymbol{\theta}}(\mathbf{G}^l)^{\mathrm{T}}\left(\mathbf{G}^l\mathbf{P}^{\mathrm{b}}_{\boldsymbol{\theta}\boldsymbol{\theta}}(\mathbf{G}^l)^{\mathrm{T}} + \mathbf{R}\right)^{-1}. \tag{25}$$

Equations (22) and (25) show that, as in the IKS and incremental 4D-Var, the FDS-IKS uses the initial background error statistics $\mathbf{P}^b_{\boldsymbol{\theta}\boldsymbol{\theta}}$ during all iterations. The updated $\mathbf{P}^{\mathrm{a}}_{\boldsymbol{\theta}\boldsymbol{\theta}}$ is just calculated in the last iteration.

## 2.4 Finite difference sensitivity multistep Kalman smoother

Here, a multistep approach is conducted by inflating the observation-error covariance matrix $\mathbf{R}$ and recursively applying a standard Kalman smoother over the assimilation window with the inflated $\mathbf{R}$ and the same observations. The multistep idea of inflating $\mathbf{R}$ for repeated assimilation of the observations was proposed by Annan et al. (2005a) and further clarified and

applied by Annan et al. (2005b) for an atmospheric GCM using the EnKF with parameter augmentation. Their approach is designed for steady-state cases, where time-averaged climate observations corresponding to a long DAW can be assumed as constant along a sequence of smaller assimilation sub-windows in which the DAW is divided. The model parameters are then sequentially updated in small increments and the loss of balance in a more general nonlinear model should be reduced with the multistep approach (Annan et al., 2005b). The inflation weights are such that in a linear case, after the predefined sequence of

integrations/assimilations, the solution is identical to that of the single step scheme along the whole DAW.

The multistep strategy was termed multiple data assimilation (MDA) by Emerick and Reynolds (2013) in the context of ensemble smoothing, and was then further developed by Bocquet and Sakov (2013, 2014) in their iterative ensemble Kalman smoother (IEnKS) where the weights for the inflation of $\mathbf{R}$ were applied in overlapping data assimilation windows (MDA IEnKS). We apply here the MDA strategy to a recursive formulation of the KF in term of FDS estimates for the dual of

the observation space to the control vector, and here we denote the scheme as finite difference sensitivity multistep Kalman smoother (FDS-MKS). As the inflation of $\mathbf{R}$ results in a reduced influence of the observations at each iteration, the increments in the early iterations are relatively reduced with respect to the FDS-IKS, making the FDS-MKS is potentially more stable. We note, however, that the inflation of $\mathbf{R}$ modifies the direction of the increment in nonlinear cases. Thus it does not converges to the same (local) minimum than the FDS-IKS (or 4DVar), but to an approximate point in the control space. In contrast with the

scheme in Annan et al. (2005b), the FDS-MKS considers recursive integration along the complete DAW (and non-overlappings DAWs). Thus, it is not restricted to steady-state conditions.

The scheme considers the total increment in the state vector that would result from the linear assimilation of one specific observation, and alternatively conducts a recursive sequence of assimilations of the same observation whose sum of fractional increments equals the total increment. This is achieved by considering that the observation error variance at loop $l$ is the

product of an inflation factor $\boldsymbol{\beta}_l$ and the variance of the "complete" observation: $\sigma_{y_l}^2 = \beta_l \sigma_y^2$. As the (linear) increment is inversely proportional to the observation error variance, for the total increment to be the same in both situations the condition that $(\sigma_y^2)^{-1} = \sum_{l=1}^{N_l} (\sigma_{y_l}^2)^{-1} = \sum_{l=1}^{N_l} (\beta_l \sigma_y^2)^{-1}$ must be fulfilled. This leads to the constraint

$$\sum_{l=1}^{N_l} \boldsymbol{\beta}_l^{-1} = 1. \tag{26}$$

Here, the sensitivity matrix $\mathbf{G}$ is estimated at each recursive step (iteration), similarly to the FDS-IKS. However, the error

covariance $\mathbf{P}_{\boldsymbol{\theta\theta}}$ is also updated at each iteration. Hence, as indicated in (20), the covariance between any climatic variable and a parameter $\theta_i$ ($\sigma_{\mathbf{x}_{k_y} \theta_i}$) is also affected by the sensitivity of the climatic variable to other parameters. The FDS-MKS is a recursive direct method, as attemps to solve the problem in a pre-specified finite sequence of iterations $N_l$:

$$\boldsymbol{\theta}^{\mathrm{a}} = \boldsymbol{\theta}^{N_l}, \qquad \mathbf{P}_{\boldsymbol{\theta\theta}}^{\mathrm{a}} = \mathbf{P}_{\boldsymbol{\theta\theta}}^{N_l}.$$

The sequences $\{\boldsymbol{\theta}^l : l = 0, \ldots, N_l\}$ and $\{\mathbf{P}_{\boldsymbol{\theta\theta}}^l : l = 0, \ldots, N_l\}$ are defined inductively as follows

$\boldsymbol{\theta}^0 = \boldsymbol{\theta}^{\mathrm{b}}, \qquad \mathbf{P}_{\boldsymbol{\theta\theta}}^0 = \mathbf{P}_{\boldsymbol{\theta\theta}}^{\mathrm{b}},$

$$\boldsymbol{\theta}^{l+1} = \boldsymbol{\theta}^l + \mathbf{K}^l[\mathbf{y} - \mathcal{G}(\boldsymbol{\theta}^l)], \tag{27}$$

$$\mathbf{P}_{\boldsymbol{\theta\theta}}^{l+1} = (\mathbf{I} - \mathbf{K}^l \mathbf{G}^l)\mathbf{P}_{\boldsymbol{\theta\theta}}^l, \tag{28}$$

where for notational convenience

$$\mathbf{G}^l \equiv \mathbf{G}(\boldsymbol{\theta}^l), \tag{29}$$

and

$$\mathbf{K}^l = \mathbf{P}_{\boldsymbol{\theta\theta}}^l (\mathbf{G}^l)^{\mathrm{T}} \left( \mathbf{G}^l \mathbf{P}_{\boldsymbol{\theta\theta}}^l (\mathbf{G}^l)^{\mathrm{T}} + \beta_l \mathbf{R} \right)^{-1}. \tag{30}$$

Regarding $\boldsymbol{\beta}$, a possible step size approach is to set the inflation weight constant for all the iterations, which given (26) leads to $\boldsymbol{\beta} = N_l \mathbf{1}$, where the column vector $\mathbf{1} \in \mathbb{R}^{N_l}$ has all values set to 1. However, as the iterations proceed, the updated

background covariance decreases so the fractional increments get smaller. A more even distribution of fractional increments, with likely improved stability, can be given by decreasing inflation weights as the iterations proceed, so initial weights are relatively higher. Thus, among other possible solutions for the inflation factor at step $l$, here we adopt the expression

$$\boldsymbol{\beta}_l = (N_l - l + 1) \sum_{n=1}^{N_l} n^{-1}, \tag{31}$$

which satisfies requirement (26). A numerical advantage of inflating $\mathbf{R}$ for the multiple data assimilation approach is that it

reduces the condition number of the matrix to be inverted in the assimilation at each iteration. A practical advantage of the FDS-MKS with respect to the FDS-IKS is that the number of iterations is predefined. With given computer resources and model computational throughput statistics, it is possible to evaluate how many ESM integrations are affordable, and set the FDS-MKS inflation weights and computing schedule accordingly. This does not mean, though, that the FDS-IKS would not converge closer to the (local) minimum of the cost function with the same iterations. Also, without specific consideration of

constraint (26), the idea of inflating $\mathbf{R}$ has also been considered by previous studies as a mechanism to improve initial sampling for the EnKF (Oliver and Chen, 2008), and also to damp model changes at early iterations in Newton-like methods (Wu et al., 1999; Gao and Reynolds, 2006).

## 2.5  Early-stopped iterations for the FDS-MKS

The computational cost of the ESMs integrations is much higher than that of the assimilation steps as considered in the FDS-

MKS for a low-dimensional control vector. In this study, we do not evaluate adaptive strategies for the planning of the weights in the FDS-MKS. However, the evolution of the increments in the control variables along the iterations could potentially be used to guide the size of $\beta_l$ at each loop, and even to conduct an early stopping of the iterations. At each iteration, it is possible

to compute the standard update using the corresponding pre-planned weight $\boldsymbol{\beta}_l$, and simultaneously to compute an alternative update with early termination of the iterations by applying a *completion* weight $\boldsymbol{\beta}_l^c$ that both terminates the iterations and fulfils the condition (26) as alternative to $\boldsymbol{\beta}_l$ in (30):

$$\boldsymbol{\beta}_l^c = \left[ 1 - \sum_{j=1}^{l-1} \boldsymbol{\beta}_j^{-1} \right]^{-1} . \tag{32}$$

Comparison of the sequence of increments given by the fractional steps of the FDS-MKS with those with simultaneous early-stopped solutions may be used to support replanning of weights and even to decide on an early stopping of the iterations, using the update given by using the completion weight as final solution.

## 2.6 ETKF and Gaussian anamorphosis

The ensemble Kalman filter (EnKF) was introduced by Evensen (1994). It makes it possible to apply the Kalman filter to high-dimensional discrete systems, when the explicit storage and manipulation of the system state error covariance is impossible or impractical. The EnKF methods may be characterized by the application of the analysis equations given by the Kalman filter to an ensemble of forecasts. One of the main differences among the several proposed versions of ensemble Kalman filters is how the analysis ensemble is chosen. Ensemble square-root filters use deterministic algorithms to generate an analysis ensemble with the desired sample mean and covariance (e.g.; Bishop et al., 2001; Whitaker and Hamill, 2002; Tippett et al., 2003). Here, in our experiments with global model parameters, we use the mean-preserving ensemble transfom Kalman filter (ETKF), or "symmetric solution", described by Ott et al. (2004), and also referred to as "spherical simplex" solution by Wang et al. (2004). The mean-preserving ETKF is unbiased (Livings et al., 2008; Sakov and Oke, 2008).

Still, for the (En)KF to be optimal, three special conditions need to apply: (1) Gaussianity in the prior, (2) linearity of the observation operator, and (3) Gaussianity in the additive observational error density. In order to better deal with nonlinearity, a number of studies have addressed the use of transformation of the model background and observation to obtain a Gaussian distribution, in a way that the (En)KF can be applied under optimal conditions. This preprocessing transformation step is known as Gaussian anamorphosis (GA) (e.g.; Chìles and Delfiner, 2012). The GA procedure was introduced into the context of data assimilation by Bertino et al. (2003), and has been applied for many years in the field of geostatistics (e.g.; Matheron, 1973; Deutsch and Journel, 1998).

It is not standard, however, how the GA should be applied in the context of DA (Amezcua and Leeuwen, 2014). The process of GA involves transforming the state vector and observations $\{\mathbf{z}, \mathbf{y}\}$ into new variables $\{\tilde{\mathbf{z}}, \tilde{\mathbf{y}}\}$ with Gaussian statistics. The (En)KF analysis is computed with the new variables, and the resulting analysis is mapped back into the original space. For the transformations, the GA makes use of the integral probability transform theorem.

In a theoretical framework and with simple experiments, Amezcua and Leeuwen (2014) evaluated several approaches using univariate GA transformations. As key point, they found that when any of above (1)–(3) conditions are violated the analysis step in the EnKF will not recover the exact posterior density in spite of any transformation. Also, they concluded that when ensemble sizes are small and the knowledge of the conditional $p_{y|x}(y|x)$ is not too precise, it is perhaps better to rely on

independent marginal transformations for both a state variable $x$ and observation $y$ than on joint transforms. For field variables, one can consider them to have homogeneous distributions so that each kind of model variable is transformed using the same monovariate anamorphosis function at all grid points of the model (e.g.; Simon and Bertino, 2009, 2012), or to apply local transformation functions at different gridpoints (Béal et al., 2010; Doron et al., 2011; Zhou et al., 2011). In both cases, the GA in these studies has been applied to the filtering problem. In the context of low-frequency past climate analysis, the temporal dimension has to be considered. For example, point (2) above would refer to the linearity in the generalized observation operator, which includes the model dynamics. Given the considerations in Amezcua and Leeuwen (2014), the sparsity of low-frequency paleoclimate records, and the lack of homogeneity in global ESM variables, here we follow the approach in Béal et al. (2010).

In our implementation of the ETKF we augmented the state vector with the model equivalent of the observations. We evaluated transformations of the control variables as well as transformations in sea surface temperature (SST), as observed variable. We transformed the control variables marginally. Regarding SST, due to sparsity and heterogeneity, we consider it is not possible to estimate the marginal distribution of the low-frequency paleo-climate observations with enough confidence to support a transformation. Thus, in our experiments we estimated the marginal distribution of the model equivalent of the SST observations, as derived from the background ensemble, and also used the same transformation for the SST observations. The transformation then operates in the marginals in independent way at each gridpoint:

$$\tilde{x} = \Phi_{\tilde{x}}^{-1}(P_x(x)) \qquad g_1(\bullet) = g_{x \to \tilde{x}}(\bullet) = \Phi_{\tilde{x}}^{-1}(P_x(\bullet))$$

$$\tilde{y} = \Phi_{\tilde{x}}^{-1}(P_x(y)) \qquad g_2(\bullet) = g_{x \to \tilde{x}}(\bullet) = \Phi_{\tilde{x}}^{-1}(P_x(\bullet)), \tag{33}$$

where $P_\xi(\xi)$ denotes the cumulative density function (cdf), and $\Phi_{\tilde{\xi}}(\bullet)$ explicitly indicates that the cdf in the transformed space is that of a Gaussian random variable. For comparison, Eq. (33) corresponds to transformacion (c) in Amezcua and Leeuwen (2014). Tests with standard ETKF are included in the two experiments below. A test with ETKF including GA as just described is included in experiment 2, with CESM.

As indicated, here we use empirical cumulative density functions (cdf's) for the anamorphosis based on the background ensemble. The risk of using the tails of the transformation function during the anamorphosis of the ensemble is significant. Thus we obtain linear tails following Simon and Bertino (2009, 2012), which consists of extrapolating to infinity the first and last segments of the interpolation function with the same slope. In practice, we just extrapolated, in each direction, until twice the original range of the Gaussian variable $\tilde{x}$. Then, we truncate (only) the physical coordinate in tail points in the transformation function to its physical bound in case it is exceeded. Then, we set the two first moments of the target Gaussian variable $\tilde{x}$ to those of the original ensemble (see section 4.4 in Bertino et al., 2003).

## 3  Experiment 1: 1D energy balance model

### 3.1  Model description

This experiment is based on a conceptual one-dimensional, South-North, energy-balance model (Ebm1D), for which Paul and Losch (2012) (PL2012 hereafter) conduct a number of 4D-Var experiments. Ebm1D is based on a) the difference between

absorbed solar radiation $Q_{\mathrm{abs}}$ and outgoing longwave radiation $F^{\uparrow}_{\infty}$ at the top of the atmosphere (TOA) on the one hand, and b) the divergence of the horizontal heat transport $\Delta F_{\mathrm{ao}}$ on the other hand. In Ebm1D, the climate is expressed in terms of just the zonally averaged surface temperature $T_{\mathrm{s}}$.

PL2012 evaluate several climate conditions and uncertain parameter scenarios, including present-day and Last Glacial Maximum (LGM) climate states. Then, with the model constrained by the present-day and LGM parameter estimates, they conduct climate projections under several $CO_2$ forcing scenarios. We revisit their PD1 scenario; a present-day test with five (scalar) parameters, summarized in Table 1, as control variables. Here we summarize the model in relation with these parameters, and the reader is referred to PL2012 for a thorough description of all model parameters and related equations. The ocean mixed layer depth, $H_{\mathrm{o}}$, controls the effective heat capacity of the atmosphere-ocean system. $A$ is a constant term in the calculation of the outgoing longwave radiation $F^{\uparrow}_{\infty}$, which also depends linearly on the surface temperature and the logarithm of the ratio of the actual value of the atmospheric $CO_2$ concentration to a reference value (Eq. 6 in PL2012). Meridional heat transport is treated as a diffusive process driven by latitudinal temperature gradients, where the horizontal heat transport depends linearly on a thermal diffusion coefficient $K_{\mathrm{ao}}(x)$, given by $K_{\mathrm{ao}}(x) = K_o(1 + K_2 x^2 + K_4 x^4)$, where $K_0$, $K_2$, and $K_4$ are the remanining three parameters included in the control vector (Table 1), and $x = \sin \Phi$, where $\Phi$ is latitude.

## 3.2 Observations and cost function

As observations, we took surface air temperature (SAT) derived from the NCEP/NCAR reanalysis data (Kalnay et al., 1996). From the reanalysis data we first calculated global zonal means of SAT. Then, we obtained SAT means for present-day climate at each grid cell (i.e., latitude) for winter (January, February and March; JFM) and summer (July, August and September; JAS) in the Northern Hemisphere. These zonal averages of SAT, $T_{\mathrm{s}}$, were the target for the analysis. The mean of the last 10 years, out of 100 years of model integration, were taken as model equivalent of the observations. That is, each grid cell in the 1D model has one observation and model equivalent for winter (JFM) and similarly for summer (JAS) in the cost function, defined by:

$$\mathcal{J}(\boldsymbol{\theta}) = \frac{1}{2}\|\boldsymbol{\theta} - \boldsymbol{\theta}^b\|_{\mathbf{P}_{\boldsymbol{\theta\theta}}^{-1}} + \frac{1}{2}\|\hat{\mathbf{y}} - \mathcal{G}(\boldsymbol{\theta})\|_{\mathbf{W}^{\frac{1}{2}}\mathbf{R}^{-1}\mathbf{W}^{\frac{1}{2}}}, \tag{34}$$

where $\mathbf{W}$ is a diagonal matrix, whose diagonal is a vector of weights $\mathbf{w} \in \mathbb{R}^p$ given to the observations. This cost function can be written as $\mathcal{J}(\boldsymbol{\theta}) = \mathcal{J}_{\mathrm{b}}(\boldsymbol{\theta}) + \mathcal{J}_{\mathrm{o}}(\boldsymbol{\theta})$. The observational target and control variables are identical to those in PL2012, but they did not include a background term $\mathcal{J}_{\mathrm{b}}$ in the cost function. As PL2012, we assumed that observation errors are uncorrelated ($\mathbf{R}$ is diagonal), with all observations having a standard deviation $\sigma_{T_0} = 1^{\circ}$ C. The explicit division of the norm for $\mathcal{J}_{\mathrm{o}}$ in terms of $\mathbf{R}^{-1}$ and $\mathbf{w}$ facilitates the comparison of scenarios as a function of increasing observational weight.

## 3.3 Experimental setup

As PL2012, we set the grid resolution to $10^{\circ}$. We assumed a diagonal $\mathbf{P}_{\boldsymbol{\theta\theta}}^b$, with standard deviations given in Table 1, which we considered as reasonable. Other than the parametric uncertainty we considered a perfect-model assumption, which is overly optimistic in this specific case as, in addition to the 1D Earth climate representation, there are strongly simplified physics in

the energy balance model. While PL2012 also assume this perfect-model framework, they point out to a number of specific structural model errors. Thus, the control variables will attempt to compensate for the unaccounted uncertainties in either of the evaluated estimation approaches.

For the PD1 tests, we made the observation weights $\mathbf{w}$ proportional to the area of the zonal band (i.e., decreasing toward the poles) with $\Sigma_{i=1}^{p} w_i = 1$, and we compared the FDS-MKS, the FDS-IKS, the ETKF ($m = 60$ members) and 4D-Var. We evaluated a two-step and a three-step FDS-MKS (a one-step FDS-MKS equals a FDS-EKS, or first iteration of the FDS-IKS). To evaluate the resilience of the FDS schemes to high perturbations, we conducted three tests with different perturbation sizes for each FDS scheme. In each one, the perturbation applied to each of the control variables was proportional to its background standard deviation by a factor *sdfac*. This perturbation factor was $sdfac \in \{0.001, 0.01, 0.1\}$. The evaluation of the cost function for the ETKF (as a smoother) is conducted with a single forward integration of the mean of the posterior control variables. In the 4D-Var minimization, as PL2012, we use a variable memory Quasi-Newton algorithm as implemented in M1QN3 by Gilbert and Lemaréchal (1989), and to compute the gradient we use a discrete adjoint approach with the tangent and adjoint codes generated automatically by the "Transformation of Algorithms in Fortran" code (TAF, http://www.autodiff.org; Giering and Kaminski, 1998; Giering et al., 2005). The number of simulations can be higher than the number of iterations as the minimizer M1QN3 takes a stepsize determined by a line search that sometimes reduces the initial unit stepsize (see Gilbert and Lemaréchal, 1989). For 4D-Var, as a stopping criterion we required a relative precision on the norm of the gradient of the cost function of $10^4$.

We conducted a number of aditional tests to compare the convergence of the FDS-IKS versus the FDS-MKS as higher weight is given to the observations. These were named PD2 and PD3, corresponding to $\Sigma_{i=1}^{p} w_i = 3$, and $\Sigma_{i=1}^{p} w_i = 5$, respectively. As these weights increase, the effect of the regularization by $\mathcal{J}_b$ decreases, and one can expect the convergence of the Gauss-Newton scheme (the FDS-IKS) to be more difficult. A few of these aditional tests evaluate how/if the convergence of the FDS-IKS can be improved (mostly in a low-regularization situation) by increasing the number of perturbations per parameter (that is, the ensemble size). The results of these aditional tests are briefly described here and expanded in Appendix A.

**Table 1.** 1D energy balance model. PD1 tests. Parameter definition and first-guess values.

| Symbol | $(\mu, \sigma)$ | Units | Description | References[1] |
|---|---|---|---|---|
| $H_o$ | (70.0, 15.0) | m | Ocean mixed-layer depth | (Hartmann, 1994, p. 84) |
| *Linearized longwave radiation* | | | | |
| $A$ | (205.0, 7.0) | W m$^{-2}$ | Constant term | (Hartmann and Short, 1979, set 2) |
| *Diffusion coefficients* | | | | |
| $K_0$ | $(1.5 \times 10^5, 1.5 \times 10^5)$ | m$^2$s$^{-1}$ | Constant term | |
| $K_2$ | (-1.33, 0.75) | | Second-order coefficient | (North et al., 1983) |
| $K_4$ | (0.67, 0.6) | | Fourth-order coefficient | (North et al., 1983) |

[1] References just for the mean values.

## 3.4 Results

Here we provide a succinct summary of the estimation process. Broader explanation of the model climatology in relation with the control variables is given in PL2012. The background sensitivity of the 10-yr mean surface temperature $T_s$ to the control variables is shown in Fig. 1, where, to ease comparison, the sensitivity matrix $\mathbf{G}$ is scaled by multiplying each of its columns by the assumed background standard deviation of the corresponding parameter. Fig. 1a shows mean ensemble sensitivities estimated from the background ensemble for the ETKF, and Fig. 1b shows local finite difference sensitivities (FDS) estimated with perturbations using $sdfac = 0.001$. Note that these background FDS are identical for both the FDS-IKS and the FDS-MKS in all scenarios (PD1, PD2, and PD3) for the same $sdfac$. Each plot has its own scale to avoid flattened lines in the ensemble sensitivity plot. For the three control variables composing the thermal diffusion coefficient $K_{ao}$, FDS are more than twice higher than the corresponding mean ensemble sensitivities. However, the sensitivities to the ocean mixed-layer depth $H_o$, and to the constant term $A$ in the longwave radiation are quantitatively similar in both cases. In both, $A$ is negatively correlated (as expected) with $T_s$ at all latitudes but with a relatively low sensitivity, while the rest of the control variables show a rather neutral, but negative, sensitivity in the tropical belt and a positive sensitivity increasing toward the poles (nearly symmetrical off the Equator), with weaker scaled sensitivities for the ocean mixed-layer depth $H_o$. In both cases, additional plots (not shown) depict summer sensitivities similar to the corresponding winter ones. Also, FDS with $sdfac = 0.01$ (winter and summer) are very similar to those shown in Fig. 1b. FDS with $sdfac = 0.1$ are also very close to those in Fig. 1b, but slightly lower for the $K_{ao}$ components, toward those in Fig. 1a.

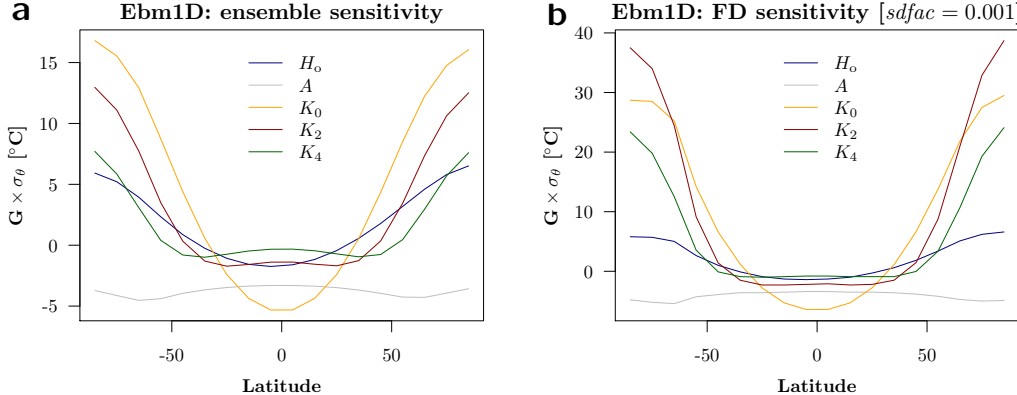

**Figure 1.** Ebm1D experiment. Background sensitivity of winter surface air temperature to the control variables estimated as (a) ensemble sensitivity (for the ETKF) and (b) finite difference sensitivity (for the FDS-IKS and the FDS-MKS) with $sdfac = 0.001$. Sensitivities are scaled by the standard deviation of the control variables, and each line refers to a column in $\mathbf{G}$. Each plot uses its own scale to ease visualization.

For the PD1 scenario, Fig. 2 summarizes the convergence, including the FDS-IKS, the 2-step and 3-step FDS-MKS, and 4D-Var; and Table 2 compares the posterior control variables and the corresponding cost function values for all the evaluated schemes. For comparison, the convergence in Fig. 2 is scaled as a function of the number of simulations, which, for five control

variables, is six simulations per iteration in the forward FDS schemes. Convergence details are given in Appendix A. For the PD1 scenario, 4D-Var took 141 simulations and 111 iterations to converge to the minimum with the convergence criterion indicated in section 3.3. A relative improved convergence by including the regularization term $\mathcal{J}_b$ can be seen by comparison with PL2012, whose cost function only considered the $\mathcal{J}_o$ term, and took 236/190 (simulations/iterations) to converge (Table 3 in PL2012). In any case, Fig. 2 shows that the 4D-Var convergence becames apparently similar to that of the FDS-IKS tests from simulation 40 onwards. In Fig. 2, the three convergence series for the FDS-IKS with the three different perturbation parameters ($sdfac \in \{0.001, 0.01, 0.1\}$) are represented with the same symbol. Starting with the same background cost function value $\mathcal{J}$, the two series with $sdfac \in \{0.001, 0.01\}$ show identical results, while the series with $sdfac = 0.1$ is the one that has a higher cost after the first iteration but still reunites with the other two series after the second iteration. However, the variations in $sdfac$ had a very minor effect on the FDS-MKS schemes. For the FDS-MKS schemes, we focus now on the end of their corresponding iterations. The 2-step FDS-MKS, for all $sdfac$ values, gives final $\mathcal{J}$ values that are close but slightly higher than those of the FDS-IKS at the same number of iterations. The same happens with the 3-step FDS-MKS with respect the third iteration of the FDS-IKS. For 4D-Var, the convergence goes slightly faster than for the FDS-IKS at simulation number six (first iteration of FDS-IKS), but then it goes slower than for the FDS-IKS after that.

Table 2 for PD1 scenario shows that the posterior values of the control variables, as well as corresponding cost function values, are nearly identical for 4D-Var and the FDS-IKS (values shown for $sdfac = 0.001$ perturbations, but also similar for the higher $sdfac$ values). The 3-step FDS-MKS also (shown for $sdfac = 0.001$) converges to relatively similar control variables. In this case, also the FDS-EKS (first iteration of FDS-IKS) obtained lower cost function values than the ETKF. Although both obtained similar values for $H_o$, $A$, and $K_0$; in the ETKF case, the values of $K_2$ and $K_4$ (with a clear nonlinear relation with the surface air temperature) diverged from the minimum obtained by 4D-Var and the FDS-IKS. This is related with the background departure from the minimum and the mean sensitivites used by the ETKF. In general, one would not expect a FDS-EKS to perform better than an ETKF with denser sampling (bigger ensemble) as is this case. Table 2 also indicates the posterior standard deviations for the Kalman based schemes. Non-diagonal values of $\mathbf{P}^a_{\theta\theta}$ are not shown. There are not high differences among the various posterior standard deviations for the Kalman based schemes, with some values being higher in one scheme, and others higher in a different scheme. In summary, the linear approaches (ETKF, and FDS-EKS) obtained some reduction in the cost function values with respect to the background, but the rest of the schemes obtained substantially lower cost function values, with the FDS-IKS and 4D-Var converging to the same minimum and getting the lowest $\mathcal{J}$ values. It is possible an alternative minimization for the strong-constraint 4D-Var would have converged faster. In any case, the FDS-IKS has shown to have a fast convergence in this experiment. Interestingly, Fig. 2 shows that the first fraction of all FDS-MKS schemes had a substantially lower cost value than either 4D-Var or any of the FDS-IKS tests. This, along with the resilience of the FDS-MKS to relatively high perturbations, supports the strategy of using a combination of the FDS-MKS in early iterations of a Newton-like scheme as the FDS-IKS, akin to Wu et al. (1999) or Gao and Reynolds (2006). Alternatively, one can conduct a line search along the direction given by the FDS-IKS increments at each iteration. Further details for this experiment are in Appendix A, focusing on the convergence of FDS-IKS versus FDS-MKS as the observational weight increases.

**Figure 2.** Ebm1D experiment. Convergence as a function of the number of simulations for the scenario PD1 ($\Sigma_{i=1}^p w_i = 1$). Each iteration of the evaluated finite difference schemes requires 6 simulations (background plus number of perturbations, with one perturbation per parameter). Two-step and three-step FDS-MKS are included. Each FDS scheme includes the convergence for the three perturbation tests (details in text).

**Table 2.** Ebm1D PD1 tests. Parameter estimation and cost function values[1].

| Parameter | Background | 4D-Var | FDS-MKS[2] | FDS-IKS | FDS-EKS | ETKF$_{60}$[3] |
|---|---|---|---|---|---|---|
| $H_o$ | $70.0 \pm 15.0$ | 60.8 | $55.7 \pm 13.7$ | $60.9 \pm 13.3$ | $62.2 \pm 14.1$ | $62.1 \pm 9.73$ |
| $A$ | $205.0 \pm 7.0$ | 209.2 | $209.4 \pm 1.94$ | $209.2 \pm 1.96$ | $208.8 \pm 1.94$ | $208.6 \pm 2.07$ |
| $K_0$ | $1.5E05 \pm 1.5E05$ | 2.2E05 | $2.1E05 \pm 4.83E04$ | $2.2E05 \pm 6.7E04$ | $2.0E05 \pm 4.9E04$ | $2.1E05 \pm 6.7E04$ |
| $K_2$ | $-1.33 \pm 0.75$ | -1.25 | $-1.06 \pm 0.37$ | $-1.20 \pm 0.38$ | $-1.33 \pm 0.43$ | $-1.42 \pm 0.50$ |
| $K_4$ | $0.67 \pm 0.6$ | 0.32 | $0.12 \pm 0.40$ | $0.31 \pm 0.39$ | $0.35 \pm 0.47$ | $0.70 \pm 0.46$ |
| $\mathcal{J}_o(\boldsymbol{\theta})$ | 14.21 | 8.83 | 8.98 | 8.82 | 11.08 | 11.83 |
| $\mathcal{J}(\boldsymbol{\theta})$ | 14.22 | 9.47 | 9.55 | 9.47 | 11.56 | 12.20 |

[1] FDS schemes with $sdfac = 0.001$. Values are identical for $sdfac = 0.01$, and slightly different with minor increase in cost function values for $sdfac = 0.1$ (see Appendix A).

[2] Three-step FDS-MKS. Details of cost function convergence for the two-step FDS-MKS shown in Appendix A.

[3] ETKF subindex indicates the ensemble size. Cost function obtained by re-integration of the model with the mean updated parameters.

## 4 Experiment 2: CESM

### 4.1 Experimental setup

The experiment 2 is a synthetic test with the Community Earth System Model (CESM1.2), a deterministic ESM. The CESM component set used here comprises the Community Atmosphere Model version 4 (CAM4), the Parallel Ocean Program version 2 (POP2), the Community Land Model (CLM4.0), the Community Ice CodE (CICE 4) as sea-ice component, the River Trans-

port Model (RTM), and the CESM flux coupler CPL7. The coupler computes interfacial fluxes between the various component models (based on state variables) and distributes these fluxes to all component models while insuring the conservation of fluxed quantities. Land ice is set as boundary conditions, and the wave component is not active. The configuration uses preindustrial forcings and it is a standard component set named as B1850CN in the CESM1.2 list of *compsets*. We use a $\sim 4°$ horizontal resolution regular finite volume (FV) grid for the atmospheric and land components, a FV grid with a displaced pole centered at Greenland $\sim 3°$ (version 7) for the ocean and sea ice components, and a $0.5°$ FV grid for the river runoff component (this is also a standard set of component grids with short name f45_g37 in CESM1.2). For comparison, this is a coarser resolution than that of the recent CESM Last Millennium Ensemble (Otto-Bliesner et al., 2016).

Here we focus on the analysis for a single DAW and equilibrium forcing and, as adequate, introduce some comments regarding practical implementations for real cases, including the case of transient forcings. The identical twin assimilation experiment is designed to approach a case of past climate reconstruction with sparse observations, as usual in pre-instrumental climate analysis. Specifically, we use the features of available observations of near sea surface temperature for the Last Glacial Maximum (LGM) from the "multiproxy approach for the reconstruction of the glacial ocean surface" (MARGO) database (MARGO Project Members, 2009). The LGM has received great attention in the paleoclimate community for its relevance to understand climate feedbacks and future climate projections, and specifically the MARGO database has been profusely used for qualitative and quantive model-data comparisons (e.g.; Kageyama et al., 2006; Hargreaves et al., 2011; Waelbroeck et al., 2014) as well as in dynamical reconstruction, with the ocean model MITgcm and 4D-Var, of the upper-ocean conditions in the LGM Atlantic (Dail and Wunsch, 2014) and the global ocean (Kurahashi-Nakamura et al., 2017). For the purpose of this study, it is not so important that the actual climate of the model matches that of the LGM but that the case study is realistic from the estimation point of view. Thus we just make use of the MARGO database characteristics (location, seasonality and uncertainty), but conduct a synthetic experiment for preindustrial climate conditions. To do so, before starting the experiment we spun up CESM for $1\,200$ yr starting from Levitus climatology with standard preindustrial conditions to reach an equilibrium state. Then, we used the restart files from the end of the spin up time to create a 60 yr control simulation (as synthetic *truth*), in which in addition to the preindustrial forcings and boundary conditions, we added a flux term to the ocean, as detailed below.

To create the background ensemble we perturbed a number of parameters for the (deterministic) physics in both the ocean and the atmosphere components, as well as input greenhouse gases and an additional influx of water into the North Atlantic ocean. As indicated in the introduction, the selected control variables have the responsibility of creating all the background uncertainty in a perfect-model scenario, and through the assimilation they will try to compensate for any unaccounted model error elsewhere. In a step-by-step approach, here all perturbed model parameters and forcings were included as control variables in the assimilation. An obvious (still synthetic) and very useful extension would be to perturb a wider set of model parameters and/or forcings and boundary conditions (e.g.; various ice sheet configurations, or alternative freshwater influx) and explicitly evaluate the compensation effect and climate reconstruction results by using subsets of the perturbed inputs as control vector. Here, the selected parameters for model physics and radiative constituents are relevant to the global energy budget of the Earth system, but not neccessarily the most sensitive model inputs for multidecadal and longer scales. In real cases, the selection of

control variables (if the control vector is to be kept low-dimensional) should be done carefully and generally based on previous global sensitivity analyses.

We included an influx of water into the North Atlantic from melting in the Greenland ice sheet (GIS) to the true run and as control variable. This flux was homogeneously distributed along the coast of Greenland and at the ocean surface, and it is appealing to explore as control variable because the Atlantic meridional overturning circulation (AMOC) plays a critical role in maintaining the global ocean heat and freshwater balance, and it is commonly acknowledged that the North Atlantic deep-water (NADW) formation is key in sustaining the AMOC (e.g.; Delworth et al., 1997), while in turn freshwater flux in the North Atlantic, along with surface wind forcing, ocean tides and convection, provides the energy for the NADW formation (e.g.; Gregory and Tailleux, 2011). Adding this freshwater flux (or *freshwater hosing*) makes the identification of the model parameters more complicated, but it is realistic to expect that current paleoclimate melting estimates can hold some bias and it is useful to know how the evaluated schemes deal with this possibility. In real cases, flux terms have been used in paleoclimate modelling as a mean to account for model errors. So, they relax the perfect-model assumption in a parametric way. Here, the estimated flux term attempts to correct the mean state towards the observations along with the model parameters. As far as the authors know, this is the first experiment with a comprehensive ESM which attempts (even in a synthetic way) a joint flux and model parameter estimation for climate field reconstruction, as these are more commonly seen as competing strategies.

We initiated the background with biased control variables with respect to the truth and a zero mean Greenland ice sheet freshwater flux. We used reasonable uncertainties in the control variables derived from previous publications. Separate analyses (weakly coupled assimilation) for different model components (atmosphere, ocean, land) may be inconsistent. In our setup, all observations are allowed to directly impact model parameters from any component in the Earth system model. This is known as strongly coupled data assimilation. Both truth and background simulation were branched from the same initial conditions, which allowed to use relatively short integration times (60 yr) in the experiment. In a real case with steady-state forcings (e.g.; estimation of real LGM climate state by assimilating the MARGO database), the model should be integrated even longer towards *quasi-equilibrium* to ensure that errors in the initial conditions will not affect the analysis (or they should be accounted for). Also, each model equivalent of the observations has to be mapped into the corresponding spatio-temporal domain of each paleoclimate proxy observation. Similar to the previous experiment, for the FDS schemes, we set the perturbations for each control variable as equal to their standard deviation multiplied by a perturbation factor *sdfac*. For computational reasons we only tested $sdfac \in \{0.001, 0.1\}$.

The cost function was as Eq. (17), where the set of control variables used for the experiment is summarised in Table 4. Sections 4.2 and 4.3 give a brief information of the atmospheric and ocean components of CESM as used in this experiment. For the rest of the model components we used default configurations for the indicated CESM *compset*. Given that adjoint codes are not available for CESM, here we alternatively tested an ETKF (with $m = 60$ ensemble members) including Gaussian anamorphosis (ETKF-GA) as possible nonlinear approach, which has a negligible extra cost over a standard ETKF. We also evaluated the 3-step FDS-MKS, the FDS-IKS (with 3 as maximum number of iterations), and the ETKF, also with $m = 60$.

## 4.2 CAM

We used the Community Atmosphere Model version 4 (CAM4) as atmospheric global circulation model (AGCM) component. A comprehensive description of CAM4 can be found in Neale and Coauthors (2011). Precipitation and the associated latent heat release drive the Earth's hydrological cycle and atmospheric circulations, and many model processes in AGCMs, including deep and shallow convection and stratiform cloud micro- and macrophysics, are responsible for the partitioning of precipitation through competition for moisture and cooperation for precipitation generation (Yang et al., 2013). Cumulus convection is a key process for producing precipitation and redistributing atmospheric heat and moisture (Arakawa, 2004), and consequently, the global radiative budget (Yang et al., 2013). Since AGCMs are unable to resolve the scales of convective processes, various convection parameterization schemes (CPSs) have been developed based on different types of assumptions. The CPS usually includes multiple tunable parameter, which are related to the subscale internal physics and are thought to have wide ranges of possible values (e.g.; Yang et al., 2012). Also, the dependence of CPS parameters on model grid size and climate regime is an important issue for weather and climate simulations (Arakawa et al., 2011). In addition, AGCMs include parameterization of macrophysics, microphysics, and subgrid vertical velocity and cloud variability to simulate the subgrid stratiform precipitation.

Here we used CAM4 with the Zhang and McFarlane (1995, hereafter ZM) deep convection scheme, and the Hack (1994) shallow convection scheme. For representation of stratiform microphysics we used the scheme by Rasch and Kristjánsson (1998); a single-moment scheme that predicts the mixing ratios of cloud droplets and ice. Regarding cloud emisivity, clouds in CAM4 are grey bodies with emissivities that depend on cloud phase, condensed water path, and the effective radius of ice particles. By default, the CAM4 physics package uses prescribed gases except for water vapor. In CAM4, the principal greenhouse gases whose longwave radiative effect is included are $H_2O$, $CO_2$, $O_3$, $CH_4$, $N_2O$, CFC11, and CFC12. $CO_2$ is asumed to be well mixed. As the use of prescribed specie distributions is computationally less expensive than prognostic schemes, for long term paleoclimate analysis we would generally favour the use of prescribed greenhouse gases, for example as given by the recently published 156 kyr history of the atmosperic greenhouse gases by Köhler et al. (2017). Still, we would acknowledge that these emerging dataset have an associated uncertainty and that it is generally appropriate to include the most influential ones as control variables in the climate analysis so their errors can be estimated as part of the assimilation.

In this study, as perturbed parameters and control variables we selected parameters related to the ZM deep convection scheme and the relative humidity thresholds for low and high stable cloud formation. Also, within the radiative constituents, we included invariant surface values of $CO_2$ and $CH_4$ as control variables. Table 4 shows the control variables in both CAM and POP2, and Table 5 shows the *true* run values in column $\mathbf{x}^t$.

This experiment is based on equilibrium simulations. Regarding real cases in transient conditions, $\boldsymbol{\theta}$ could, e.g., include a constant error term (i.e., a bias) for a transient greenhouse gas datasets as prescribed radiative constituent in the atmospheric model (e.g.; from Köhler et al., 2017). The estimated bias would be updated for succesive DAWs. One could think of more complicated autocorrelated error models for the greenhouse gas dataset (e.g.; García-Pintado et al., 2013), but it seems highly unlikely to us that available proxy datasets for low-frequency climate variability can constraint model errors further than simple biases in GHG forcing. We did not evaluate any parameter in relation with indirect effects of aerosol to cloud nucleation and

autoconversion, despite the overall effect of aerosol to cloud albedo and cloud lifetime, and so to climate, remains largely uncertain (Chuang et al., 2012).

## 4.3 POP2

As ocean component, we used POP2 (Smith et al., 2010). Subgrid scale mixing parametrization includes horizontal diffusion and viscosity and vertical mixing. For horizontal diffusion we chose an anisotropic mixing of momentum, and the Gent and Mcwilliams (1990) parameterization, which forces mixing of tracers to take place along isopycnic surfaces, with activated submesoscale mixing. The main drawback in the Gent and Mcwilliams (1990) scheme is that it nearly doubles the running time with respect to other simpler schemes. For vertical mixing, we chose the K-profile parameterization (KPP) of Large et al. (1994). In the KPP mixing, interior mixing coefficient (viscosity and diffusivity) are computed at all model interfaces on the grid as the sum of individual coefficients corresponding to a number of different physical processes. The first coefficients are denoted as background diffusivity $\kappa_\omega$ and background viscosity $\upsilon_\omega$ (not to be confused with the "background" in assimilation terminology), which represents diapycnal mixing due to internal waves and other mechanisms in the mostly adiabatic ocean. Other coefficient are associated with shear instability mixing, convective instability, and diffusive convective instability. The background viscosity is allowed to vary with depth, but here we assumed a depth-constant vertical viscosity $\kappa_\omega = $ bckgrnd_vdc1, where bckgrnd_vdc1 is a model input parameter. The model then computes $\upsilon_\omega = P_r \kappa_\omega$, where $Pr$ is the dimensionless Prandtl number (set to $P_r = 10$ in the model).

As control variables in POP2 we chose the Gent-Williams isopycnic tracer diffusion parameter and the (constant with depth) KPP background viscosity, both with default values for the *truth*. A third control variable in POP2 was the total freshwater influx from the Greenland ice sheet, which we distributed homogeneously along the coast of Greenland and only at the ocean surface.

## 4.4 Observations

The observational dataset is composed of point samples of climate averages for the last 20 years out of a total 60 yr integration time of a true simulation. The synthetic observations were located at the horizontal locations and 10 m depth of the MARGO database, and the sampling characteristics reproduce those of the MARGO. The MARGO database is a synthesis of six different proxies and is considered to represent the combined expertise of at least a sizeable fraction of the LGM paleocommunity. The observational uncertainty was taken from the MARGO database as input to the assimilation, but we did not add any error to the synthetic observations. MARGO provides observations (or reconstructions) of near sea surface temperature (SST), for the Last Glacial Maximum (LGM). The proxy types on which the SST estimates are based are a) microfossil-based: planktonic foraminifera, diatom, dinoflagellate cyst and radiolarian abundances, and b) geochemical palaeothermometers: alkenones unsaturation ratios ($U_{37}^{K'}$) and planktonic foraminifera Mg/Ca. Details of the database are given in MARGO Project Members (2009).

In summary, MARGO provides seasonal means for Northern Hemisphere winter (January, February and March; JFM) and summer (July, August and September; JAS), as well as annual means. However, the data availability for each of the three

temporal means (winter, summer, and annual) is different for each proxy type. Specifically, diatoms are just available for Souther Hemisphere summer; dinoflagelates, foraminifera, and Mg/Ca are available for the three temporal means; and $U_{37}^{K'}$ are only available as annual means. The observation errors are assumed uncorrelated, but each individual record in MARGO contains a specific uncertainty. Mapped into the SST space, the range of standard deviations in MARGO is within 0.79 and 4.87 °C, with relatively homogeneous uncertainty ranges among proxy types. Fig. 3 shows the type and location of the proxy
5    data. In addition to the locations and uncertainty, we emulated the temporal mean availability of the observations, with all temporal means calculated over the last 20 yr of the integration time of the true simulation. Thus, the synthetic observations in the experiment, as well as those in MARGO, impose a less rectrictive constraint not only in areas in which observations are more sparse, but also in those locations for which just one season or just annual means are available.

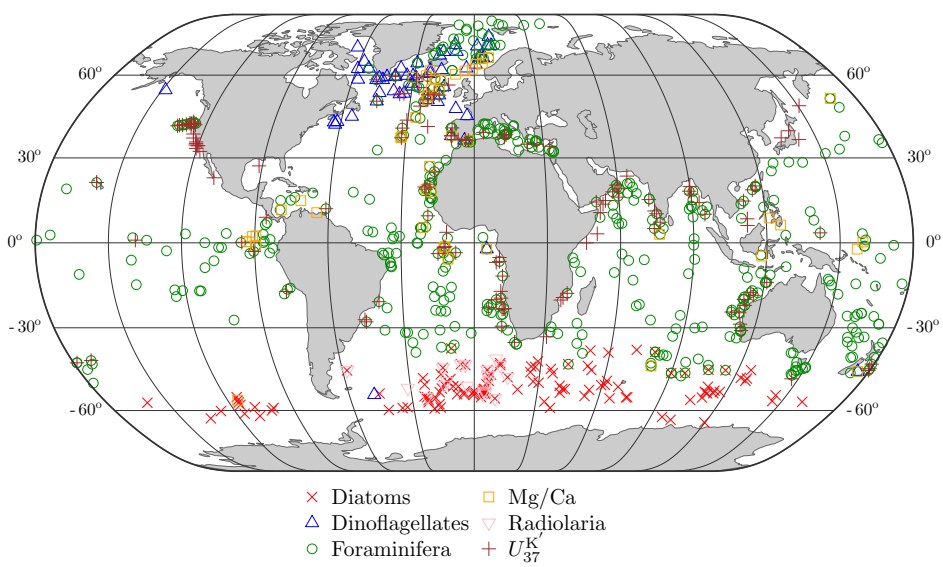

**Figure 3.** MARGO data coverage (MARGO Project Members, 2009).

## 4.5 Results

10    ### 4.5.1 Nonlinearity and Gaussian anamorphosis transformations

The need for nonlinear estimation is justified based on the assumed nonlinear relationship between the control variables and the observation space. In this experiment, nonlinearity is imposed by the Earth system model, which directly generates SST; the observed variable. In a more general case of past climate analysis, the forward operator (*proxy system model*) can impose further nonlinearity when the observed variables are direct proxy records (e.g.; foraminiferal counts, ring tree widths, speleothems,
15    etc.). In addition to model and forward operator nonlinearity, non-Gaussianity in the control variables also renders (En)KF non-

optimal. Here we conducted a test with ETKF including the Gaussian anamorphosis (GA) transformation (ETKF-GA). The test transforms the control variables, whose background deviations from Gaussianity here derive from imposed bounds, and the SST in both the model equivalent of the observations, and the observations themselves, with the strategy explained in section 2.6. In this section, we show an example of non-Gaussianity and nonlinearity in this experiment, which are the motivation for the iterative FDS schemes evaluated in this manuscript as well as for testing the Gaussian anamorphosis. Specific estimation results are given in section 4.5. Note that no transformation has been applied to the FDS schemes.

In the ETKF-GA test, we conducted the marginal Gaussian anamorphosis for all the variables in the control vector. Table 3 shows the p-values of the Shapiro-Wilk normality test (Shapiro and Wilk, 1965) conducted on the original (raw) control vector, and the anamorphosed ones (ana). The second column shows that Gaussianity is improved in all cases, most of them even reaching a $p-value = 1$. We note that despite the GA was applied to all the control vector, the raw samples of the background control variables had also a $p-value > 0.05$ in all cases. For the SST observations, we evaluated the marginal normality of the raw SST with the Shapiro–Wilk test. Then, we only applied the GA transformations at those locations with the background SST deviating significantly from Gaussianity.

**Table 3.** CESM experiment. P-values of Shapiro-Wilk normality test for the ensemble of original control variables (Raw) and transformed by Gaussian anamorphosis (Ana).

| Parameter | Raw | Ana |
| --- | --- | --- |
| CAM.cldfrc_rhminh | 0.80 | 1.00 |
| CAM.cldfrc_rhminl | 0.07 | 0.69 |
| CAM.zmconv_c0_lnd | 0.14 | 0.98 |
| CAM.zmconv_c0_ocn | 0.22 | 0.87 |
| CAM.zmconv_ke | 0.72 | 1.00 |
| CAM.ch4vmr | 0.54 | 1.00 |
| CAM.co2vmr | 0.24 | 1.00 |
| POP2.bckgrnd_vdc1 | 0.95 | 1.00 |
| POP2.hmix_gm_nml.ah | 0.97 | 1.00 |
| POP2.imau_gis | 0.40 | 0.99 |

The minimum relative humidity for high stable cloud formation (CAM.cldfrc_rhminl parameter), given its background uncertainty, was shown to have a strong effect on SST in the experiment. In most of locations of the global ocean the marginal background SST had a non-Gaussian but still unimodal cdf. Especially complicated was the North Atlantic, with strongly bimodal background distributions in some locations. This case is depicted in Fig. 4. As shown in Table 3, the CAM.cldfrc_rhminl parameter had a sharp increase in its background Gaussianity in the anamorphosed variable, and Fig. 4a describes the transformation of SST in a location in the North Atlantic ocean. Specifically, this is the observation with the highest negative innovation in the experiment, and it seems to represents the most complicated case in the test. Apart from the high (negative) innovation,

the raw observation falls well in the tail of the transformation, so its forward and backward transformations are rather sensitive to the dealing of the tails in the anamorphosis function, as mentioned in section 2.6 and Simon and Bertino (2009, 2012). In addition, there is the effect of bimodality in SST in this case. As a result, despite both marginal transformations (control variable and SST) have a clear increase in their marginal Gaussianity, their joint distribution is not bivariate Gaussian, as seen in Fig. 4b. This generic possibility also very well described by Amezcua and Leeuwen (2014). Appendix B shows an example of the more general case, where the Gaussian anamorphosis does not show specific issues.

### 4.5.2 Sensitivities and minimization

Table 5 shows the values of the control variables for the true simulation, $\mathbf{x}^{\mathrm{t}}$, the background (or *prior*), $\mathbf{x}^{\mathrm{b}}$, and the analysis (or *posterior*) estimates, $\mathbf{x}^{\mathrm{a}}$, for the evaluated schemes, as well as the value of the cost function $\mathcal{J}$ for the background and each estimation. The observational term in the cost function, $\mathcal{J}_{\mathrm{o}}$, is also shown to give an indication of the relative contribution from each cost function term. $\mathcal{J}_{\mathrm{o}}$ is calculated by reintegration of the updated (or *posterior*) control vector in the FDS schemes, and reintegration of the mean of the updated control vector ensemble for the ETKF and the ETKF-GA. An initial experiment with the perturbation factor $sdfac = 0.001$ showed extremely high sensitivities, and the the corresponding FDS-EKS (or first step of the FDS-IKS) had a very small increments in the control vector variables. Alternative the scale of sensitivites for a perturbation factor $sdfac = 0.1$ was more in agreement with the ensemble sensitivites of the ETKF background (and higher increments in a good direction), and we decided to apply this perturbation factor to both the FDS-MKS and FDS-IKS. Still, according to the previous experiment, these high perturbation factors should favor the (more regulated) FDS-MKS. All the following results refer to $sdfac = 0.1$. Due to limits in the computational quota, we did not test alternative perturbation factors in this experiment. It is very likely that smaller perturbations (e.g.; $sdfac \sim 0.01$) would have resulted in improved results for the FDS schemes, and mostly for the FDS-IKS. Thus, in a way, results here may be interpreted as relative low performance bounds for the FDS schemes under the given experimental assumptions.

All schemes obtained a substantial reduction in the value of the cost function with respect to the background, which had a $\mathcal{J}(\boldsymbol{\theta}^{\mathrm{b}}) = 373.39$. Within the evaluated schemes, the 3-step FDS-MKS obtained the lowest cost function value with $\mathcal{J}(\boldsymbol{\theta}^{\mathrm{a}}) = 51.85$. The cost for the FDS-IKS (stopped at the third iteration) was slightly higher ($\mathcal{J}(\boldsymbol{\theta}^{\mathrm{a}}) = 55.20$). However, the observational term $\mathcal{J}_{\mathrm{o}}(\boldsymbol{\theta}^{\mathrm{a}})$ was lower for the FDS-IKS. The $m = 60$ member ETKF resulted in a higher cost value ($\mathcal{J}(\boldsymbol{\theta}^{\mathrm{a}}) = 66.43$), and the ETKF-GA in a very similar $\mathcal{J}(\boldsymbol{\theta}^{\mathrm{a}}) = 66.8$. As expected, the FDS-EKS (or first iteration of the FDS-IKS), resulted in the highest cost value, with $\mathcal{J}(\boldsymbol{\theta}^{\mathrm{a}}) = 93.13$, as it is linear (as the ETKF) and also the FDS reduced exploration of the sensitivity is noisier than that of the ETKF with $m = 60$ members. The cost function for the thee iterations of the FDS-IKS is shown in Table B1.

As seen, regarding the cost function, the Gaussian anamorphosis (as implemented here) does not improve the minimization with respect to the ETKF, although the transformation serves to obtain a lower value for the observational term $\mathcal{J}_{\mathrm{o}}$. Out of several tranformation possibilities, Amezcua and Leeuwen (2014) concluded that using the cdf estimated from the ensemble model equivalent of the observations to also transform the observations (as we have done here, given the considerations indicated in section 2.6) was the worst option. In any case, we have also seen here a specific example that shows that even if

marginal Gaussianity is improved in both transformed control variables and the model equivalent of the observations, the joint cdf is not guaranteed to be bivariate Gaussian (Fig. 4). It is also interesting to note that the slope of the scatterplot examples shown in Fig. 4 is positive for the two separate branches off an apparent switch occurring in SST (with a sharp negative step in SST around CAM.clfrc_rhminl=0.88) at this location in the North Atlantic. Thus, the slope for a linear regression in this relation would have been clearly negative, as opposed to the slopes in the individual two branches. This switch in SST here

seems to be associated to a displacement in the North Atlantic gyre and (possibly coupled) cloud development. This general negative slope in SST with respect to CAM.cldfrc_rhminl is clearly shown at the given location in the ensemble sensitivity plot in Fig. 6. The switch also indicates that the background values and perturbation sizes are key for FDS estimation, and supports why the corresponding SST plot for FDS in Fig. 6 differs substantially from the ensemble sensitivity at the given location (note that the sensitivity about the background is the same for both FDS schemes; the FDS-MKS and the FDS-IKS).

Additional sensitivity plots (not shown) also indicate a positive mean ensemble sensitivity of both the total cloud cover and the large-scale precipitation with respect to the CAM.clfrc_rhminl at this location, while the sensitivity of the cloud cover to CAM.clfrc_rhminl becomes negative for latitudes lower than 50°N in the North Atlantic. A major point here is that the Gaussian anamorphosis cannot solve the strong nonlinear relationship in this case. Also, the MARGO coverage is far denser in the North Atlantic than in the rest of the global ocean. Thus, the updates in global parameters in the control vector are stronly

influenced by the sensitivities in the area. The MARGO coverage in the North Atlantic can be adequate regarding the analysis of the glacial climate at the LGM, and here it has been useful to show that the anamorphosis, as applied here, has not able to solve the strong nonlinearities found in the North Atlantic area. Still, in other conditions with available proxy time series, one could possibly apply alternative transformations for the observations based on their own statistics, and result could perhaps improve. Also, a further evaluation of the specific contribution from each observation to the increments could serve to design

quality controls and improved ensemble approaches. Looking at the posterior estimates in Table 5, it does not seem possible to derive any general conclusion about the benefit of the Gaussian anamorphosis in ensemble Kalman schemes for multidecadal climate analysis at the view of this experiment, but further exploration is needed.

The lower cost function values of the FDS schemes with respect to the ETKF (with/without GA) suggests a benefit in the more limited (and noisier) but iterated local sensitivity estimation. Also, the computational cost in the FDS tests was about half

of the ETKF (and ETKG-GA). Regarding the estimation of specific control variables, all of the evaluated schemes had some variables for which the estimation, starting from the background, went in the wrong direction with respect to the true values. For example, while the closest estimate to the true value of the relative humidity threshold for high stable cloud formation (cldfrc_rhminh) was given by the FDS-MKS, with a slight overshooting (0.81 versus 0.80 for the truth). It may have been that this slight overshooting has partially compensated for the effect of other control variables. Thus, the FDS-MKS estimates of the

freshwater flux from the Greenland influx went in the wrong directions, as were the estimates for the autoconversion coefficients in the Zhang-McFarlane deep convection scheme. On the other hand, the FDS-IKS had a total increment in cldfrc_rhminh in the wrong direction, but had the Greenland influx total increment in the right direction. The ETKF did not show any overshooting, but had some control variable increments going in the wrong direction. For the ocean background vertical diffusitivity, the only

two schemes for which there was some, albeit minor, improvement in the estimate were the ETKF and the FDS-IKS. Still, the improvement is so light in these cases that it could be a random effect.

The perturbations in the FDS may have been far from optimal for sensitivity estimation regarding their effect on the model SST at the locations (including depth) of the observations for the integration times. Table B2 shows the corresponding standard deviation for the background and posterior estimates in the ensemble and iterated FDS schemes. In general, the posterior variance of the control vector was higher for the ETKF and the ETKF-GA and a bit smaller in the FDS schemes, but also the relative reduction in variances with respect to the background was not systematically lower or higher for any specific scheme.

In sensitivity studies, the conditional sensitivity exploration of the FDS has also been termed one-at-a-time (OAT) sensitivity analysis. The difference between the local sensitivities of the FDS and the mean sensitivities of the ensemble for the ETKF may affect the estimation of the various parameters in different degrees. For comparison, with a prescribed SST, Covey et al. (2013) evaluated the sensitivity of the radiation balance at the top of the atmosphere in the model CAM to a large number of input parameters with both an OAT exploration and an alternative Morris one-at-a-time (MOAT) sampling (somehow closer to the mean ensemble sensitivites). For example, they found with both methods that the highest sensitivity of the upward shortwave flux (solar energy reflected back to space by the atmosphere and surface) was with respect to cldfrc_rhminl, out of 21 evaluated CAM parameters. As cldfrc_rhminl is the threshold of the relative humidity value at which low-level water vapor starts to condense into cloud droplets, their result is consistent with the role of thick stratus clouds in reflecting sunlight.

This study does not attemp to give an in-depth analysis of the assimilation results for the corresponding climate field reconstructions. However, we summarise some results of the spatial patterns shown in the climate reconstructions and give examples of sensitivities as estimated by the FDS schemes and the ETKF. Fig. 5 shows, in general, a similar absolute bias reduction for the FDS-MKS and the ETKF in both general magnitude and spatial patterns for both SST and SSS. For SST, the most problematic area is that where most of the observations come from the diatom locations in the MARGO database. A reason for that seems to be that observations for diatoms locations are just the 20 yr means for winter in the Northern Hemisphere, which reduces its impact on the climate analysis with respect to other observation types. Still, in general the FDS-MKS has slightly less areas where the absolute bias in the SST could not be reduced. Also the negative effect of the assimilation, regarding absolute bias reduction, on SSS which is shown for the ETKF for the North Atlantic, the Bering strait, and in some areas of the Artic ocean is negligible in the FDS-MKS. While these unobserved areas (from the point of view of the MARGO database) remain largely unconstrained, the FDS-MKS seeems generally more able to correct for the biases in areas with more observations and simultaneously not having a negative effect in areas far from the observations.

As an example of estimated sensitivities in the ocean, Fig. 6 shows the sensitivity of the sea surface temperature (SST) and sea surface salinity (SSS) to cldfrc_rhminl in CAM, as mean ensemble sensitivities and FDS about the background value. In the case of SST the general pattern of both sensitivities is quite similar, except for the much more negative sensitivity shown at the North Atlantic ocean above 50° of latitude for the ETKF, as detailed above. The second loop of the FDS-MKS (not shown) shows a similar pattern to the first loop. However the sensitivity in the third loop (not shown), approaches more the sensitivity of the ETKF and also shows a similar negative sensitivity area, in extent and magnitude, in the North Atlantic. Something similar happens to the sensitivity estimates for the SSS, where the FDS in the first and second loops of the FDS-MKS are

reasonably similar to the mean one from the ensemble background for the ETKF, with the major differences being in the Artic ocean, around the Bering strait, and the North Atlantic. The third loop of the FDS-MKS (not shown) also shows the more homogeneous band of sensitivity between the coasts of Canada and Europe shown for the ETKF.

Vertical diffusion in the ocean determines ocean heat uptake and in turn the air-sea heat flux and atmospheric heat transport, but also sea-surface temperature, evaporation and atmospheric moisture transport. Regions more sensitive to the ocean

vertical difussion would be coastal upwelling systems (e.g.; Namibia), the equatorial oceans and the Southern Ocean in case of upwelling, but also the North Atlantic Ocean in case of downwelling (deep-water formation), which is key in sustaining the Atlantic meridional overturning circulation (AMOC) (e.g.; Delworth et al., 1997). Thus, as a further example in the ocean, we find also interesting to show a sensitivity example for deeper ocean layers considering the short integration time (60 years) of the experiment. For this integration time the deep ocean is far from reaching equilibrium, but the maximum value of the AMOC

has reasonably converged in the perturbed members for the ETKF and in the FDS-MKS members for the three iterations. Fig. 7 depicts the sensitivity of the AMOC to the background vertical diffusity ($\kappa$; POP2 parameter bckgrnd_vdc1) for the ETKF and the three iterations of the FDS-MKS. It can be seen that the general pattern have some similarities in all cases, but also noticeable differences. The first iteration of the FDS-MKS (FDS-MKS.f01) is quite close to the mean estimate of the ETKF, except for the high sensitivity area around 30°N and 1.5 km deep, which is mostly missing in the FDS-MKS.f01. The pattern

of the sentitivity in the second iteration (FDS-MKS.f02) is similar to that in the first iteration but with much higher values. The third loop again has reduced sensitivities and keeps missing the deeper high sensitivity area given by the mean sensitivity of the background ensemble for the ETKF. This warrants further study to analyse with more detail why, despite being sampling different regions of the parameter space, the conditional parameter sampling of the FDS-MKS does not show in any case the deeper high value area estimated by the mean sensitivity.

As last related example, Fig. 8 shows the sensitivity of some atmospheric variables to the same ocean background vertical diffusion parameter obtained from the ensemble sensitivity (for ETKF) and the first iteration of the FDS schemes (for FDS-IKS or FDS-MKS). The atmospheric variables are the 2 m air temperature (T2M), the convective precipitation (PRECC), the large-scale precipitation (PRECL), and the total cloud cover (CLDTOT). In general, the four variables show some similarities between the ensemble sensitivities and the FDS but also some important differences. The 2 m air temperature air sensitivity

patterns are in a rather good agreement, with the ensemble sensitivity mostly resembling a smoothed version of the FDS, although also diverging in some areas as Northern Europe around Scandinavia and Northern Asia. The relative amplified sensitivity seen in the FDS plot leads also to some areas reaching a negative sensitivity in the FDS estimates, while having a low but still positive sensitivity in the ensemble plot, as happens to the south of 30°S in the Atlantic and Pacific around South America. Part of the differences are for sure due to the more global sensitivity explored in the ensemble, but also it is pending

to seee how different perturbation sizes in the FDS would have affected its sensitivity pattern and amplitude. With different patterns, a similar comparison can be done between the ensemble sensitivity and the FDS for the convective precipitation, which in both cases show a higher (in absolute values) sensitivity in the tropics, and a rather reasonable agreement, although very different in the tropical Eastern Pacific to the south of the Equator. The tropical climate may indicate the indirect effects of changes in evaporation, atmospheric convection and cloud formation. Higher differences result for the large-scale precipitation,

although some similarities are also present; as the trough stretching from the tropical Eastern Pacific, across South America, to the South of Africa, or the high sensitivity in the tropical Western Pacific. The total cloud cover finds a substantial difference between the sensitivity plots above $\sim 30°$N. However, the delineation (0 isoline) between the negative and positive sensitivites is substantially similar in both sensitivites. In general, despite the differences between the two sensitivity estimates (due to the different exploration of the control vector space), the similarities, along with the reasonable sensitivity patterns shown for the AMOC for the same ocean parameter $\kappa_\omega$, point to the usefulness of the strongly coupled assimilation for the low-frequency climate analysis, where observation from one component of the Earth system is allowed to influence the state and parameters for a different component.

An important last consideration is that the assimilation will just attempt to minimise (or get the first moments) of the cost function. So the assumed background statistics in the cost function are instrumental in controlling the control vector increments in the assimilation and the resulting climate field reconstruction (CFR). In this synthetic experiment the source of errors is known, and we assume a perfect-model framework except for the assumed uncertainties in the chosen control vector. However, the real applied situation does not know about real model errors. As described in the introduction, the control vector increments will compensate for non-accounted errors. Although the minimization can highly reduce the value of a cost function and improve the corresponding CFR, it does not necessarily imply that updated parameters for the model physics (or their moments), as part of the control vector, actually correspond to improved (extrapolable) model physics. For example, the use of the posterior model parameters can potentially lead to improved climate simulations for other prospective climatic conditions, but not necessarily. Thus, it is important to distinguish between the use of the assimilation methods for CFR including parameter estimation, and the trust one can have in the estimated model parameters for future climate projections under very different climatic conditions. A fair caveat note was recently given by Dommenget and Rezny (2017) regarding the use of flux corrections as an alternative to parameter estimation in CGCMs. As they indicated, the compensating error risk when using parameter estimates for one specific observation dataset for future projections can be eliminated by using flux corrections instead to estimate the CFR. However, as flux corrections (with specific shapes) serve as a parametric way of expressing a model error, but also try to account for errors in boundary conditions (e.g.; in melting in a non-modelled ice sheet), which are specific for a climatic condition, the estimated flux corrections do not serve to improve the climate projections either.

All in all, our experiment with CESM is an example of joint estimation of flux correction, model parameters and forcing errors. The goal is to analyse the low-frequency past climate, and at the very least the divergence between the parameters for the model physics estimated for multidecadal and longer past climate conditions and the values estimated for present conditions should serve to evaluate the reasons for the divergences, which points back to our introductory opening reference (Kageyama et al., 2018).

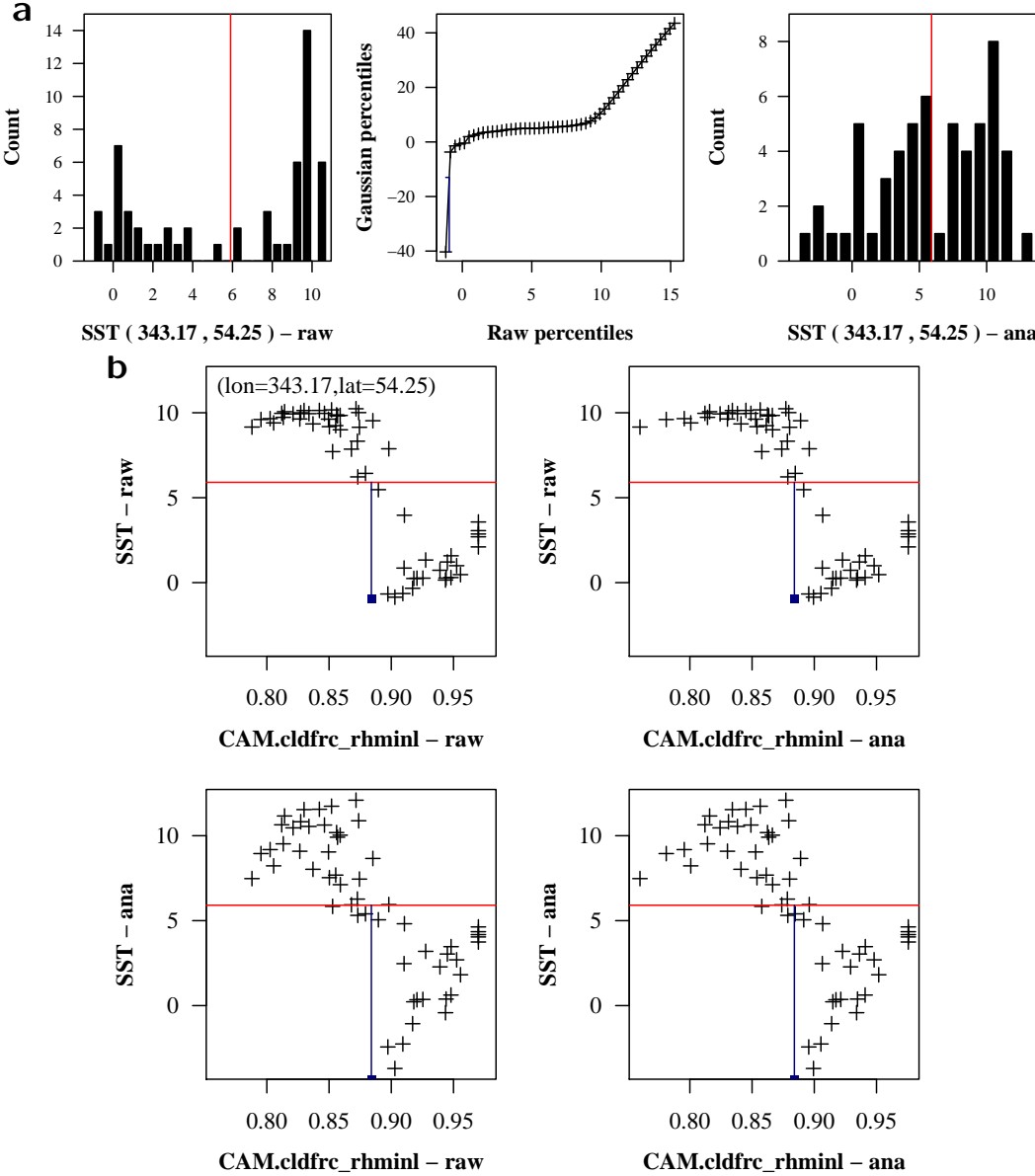

**Figure 4.** CESM experiment. Example of Gaussian anamorphosis transformation in a control variable (minimum relative humidity for high stable cloud formation) and the sea surface temperature (SST) for both the model equivalent of the observations and the observation (transformation details in text) at a location (343.17°E, 54.25°N; see Fig. 5) in the North Atlantic, where an apparent switch in SST takes place. Plot (a) shows the anamorphosis for the SST at the location, where the left panel shows the histogram of the original (raw) variable, the middle panel shows the empirical piecewise linear transformation which maps the raw variable percentiles into those of a normal variable with the two first moments (mean and standard deviation) preserved as those in the raw variable, and the right panel shows the histogram of the anamorphosed variable. Plot (b) show scatterplots between the two variables in for the original variables, the cases that only one is transformed, and the case with both variables transformed. Vertical red lines in (a) and horizontal lines in (b) indicate the mean of the model background SST. Blue squares indicate observations, vertically placed and connected with the mean of the control variable.

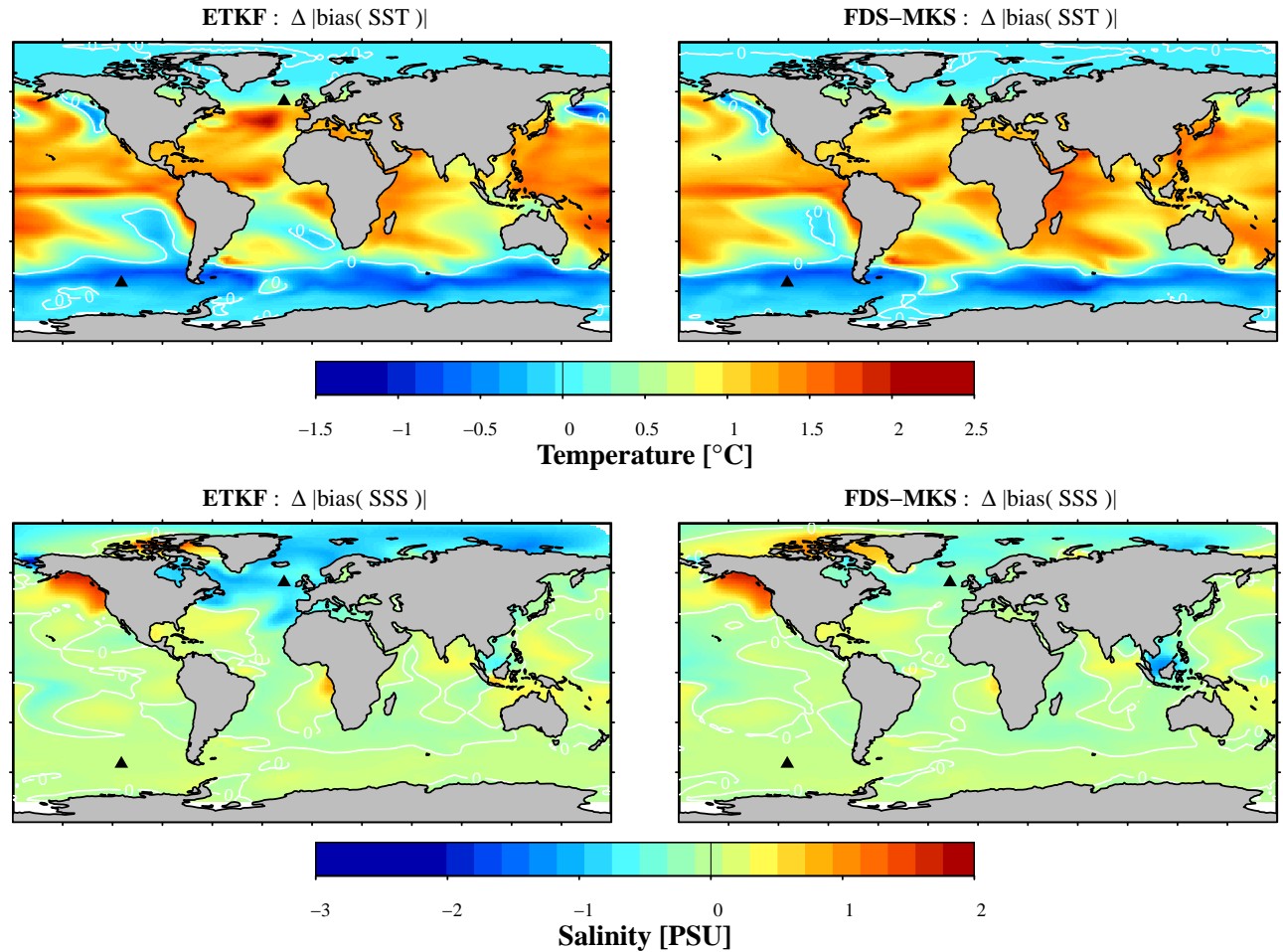

**Figure 5.** CESM experiment. Absolute bias reduction for SST and SSS as a result of a new integration with the parameters estimated with the ETKF and the FDS-MKS. The statistics are the absolute bias between the background and the truth minus the absolute bias between the analysis and the truth. Thus, positive values are a net bias reduction. Isolines at value 0 shown in white. The two triangles indicate the locations in the North Atlantic and South Pacific connected with Figs. 4 and B1.

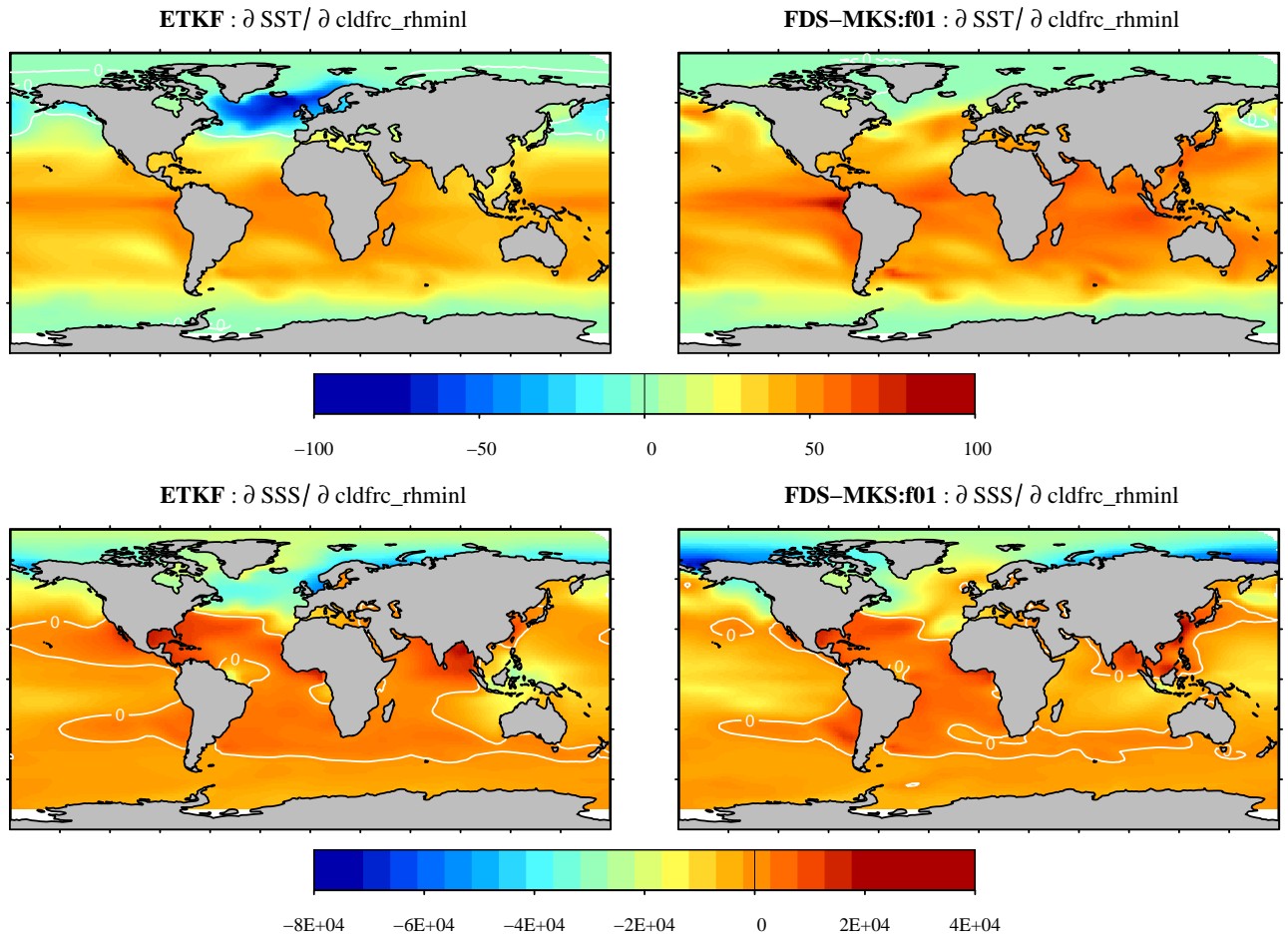

**Figure 6.** CESM experiment. Sensitivity of SST [°C] and SSS [PSU] to the minimum relative humidity for low stable cloud formation (CAM.clrfrc_rhmin) estimated from the ETKF background ensemble and the first iteration of the FDS-MKS. Isolines at value 0 shown in white.

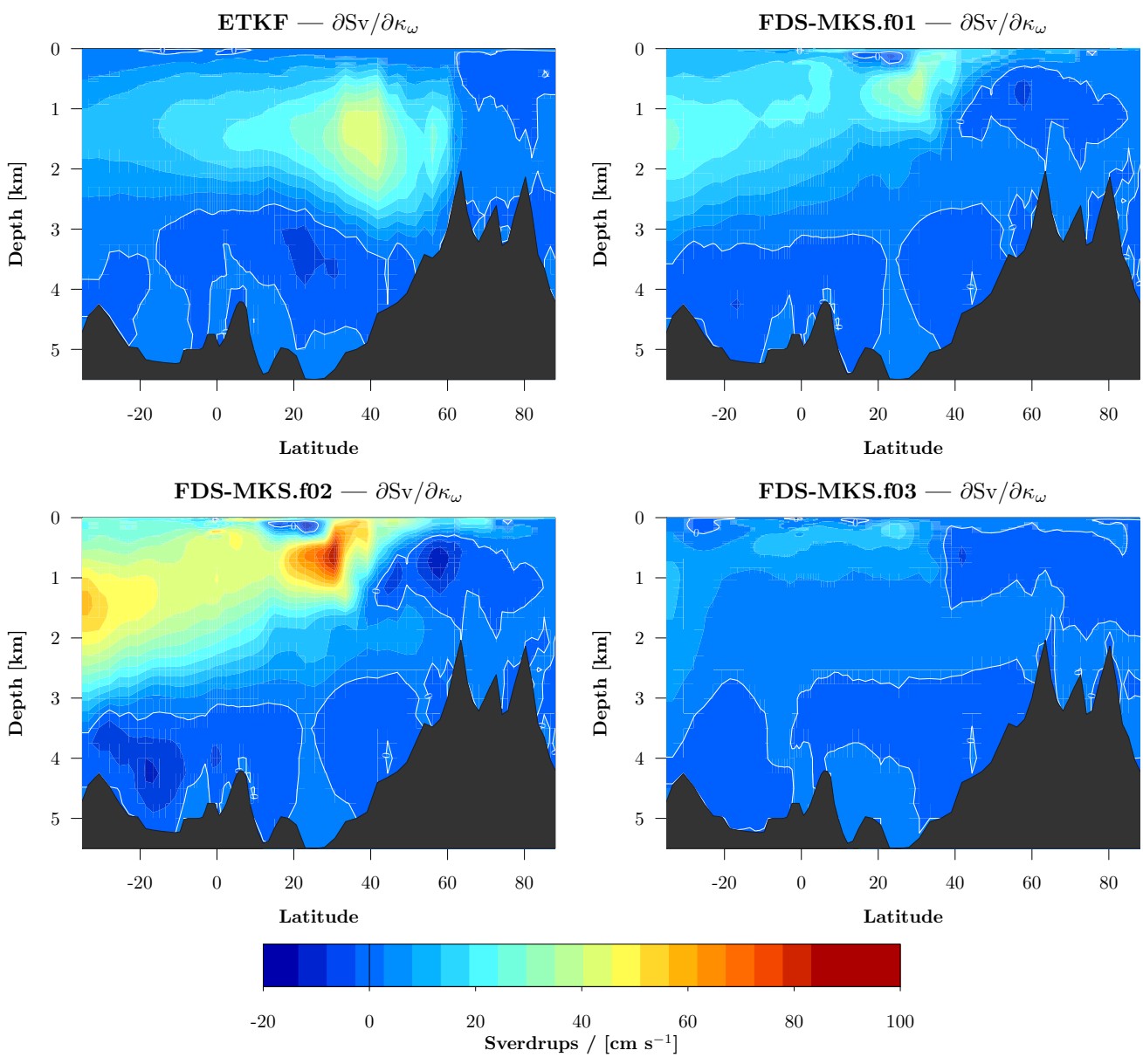

**Figure 7.** CESM experiment. Sensitivity of the Atlantic meridional overturning circulation (AMOC) to the ocean background vertical diffusion parameter ($\kappa_\omega$; POP2.bckgrnd_vdc1) in the ocean model estimated as ensemble sensitivity (from the ETKF background ensemble) and local FDS along the iterations of the 3-step FDS-MKS.

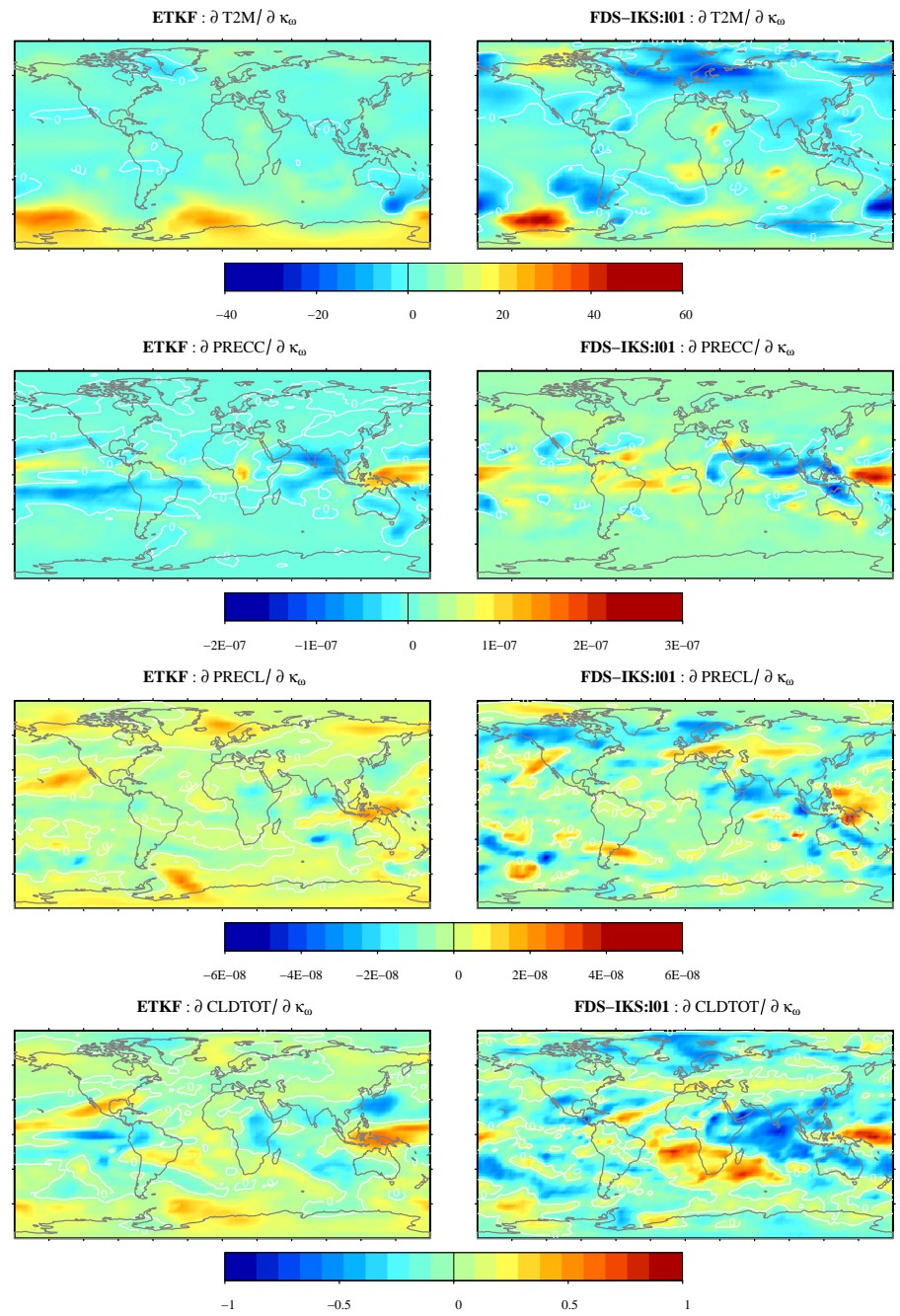

**Figure 8.** CESM experiment. Sensitivity of several atmospheric variables to the ocean background vertical diffusion parameter ($\kappa_\omega$; POP2.bckgrnd_vdc1) in the ocean model estimated as ensemble sensitivity (from the ETKF background ensemble) and the local FDS for the firs-iteration of both the FDS-MKS and the FDS-IKS. The atmospheric variables are the 2 meter air temperature (T2M), total convective precipitation rate (PRECC), large-scale (stable) precipitation rate (PRECL), and vertically integrated total cloud cover (CLDTOT).

**Table 4.** CESM definition of control variables.

| COMP.name[1] | Description | Units |
|---|---|---|
| CAM.cldfrc_rhminh | minimum relative humidity for high stable cloud formation | [-] |
| CAM.cldfrc_rhminl | minimum relative humidity for low stable cloud formation | [-] |
| CAM.ch4vmr | greenhouse gases, $CH_4$ volume mixing ratio | ppb |
| CAM.co2vmr | greenhouse gases, $CO_2$ volume mixing ratio | ppm |
| CAM.zmconv_c0_lnd | autoconversion coefficient over land in ZM deep convection | [-] |
| CAM.zmconv_c0_ocn | autoconversion coefficient over ocean in ZM deep convection | [-] |
| CAM.zmconv_ke | evaporation efficiency in ZM deep convection | [-] |
| POP2.bckgrnd_vdc1 | KPP mixing: background vertical diffusivity (Ledwell) | $cm^2s^{-1}$ |
| POP2.hmix_gm_nml.ah | Gent-Williams isopycnic tracer diffusion (Redi)[2] | $cm^2s^{-1}$ |
| POP2.freshwater_gis | freshwater influx homogeneously distributed around Greenland | Sv |

[1] *COMP:name* CESM component and parameter name.

[2] We constrained POP2.hmix_gm_nml.ah_bolus to equal POP2.ah.hmix_gm_nml.ah in the background and updates.

**Table 5.** CESM control vector estimation.[1]

| COMP.name | $\mathbf{x}^{\mathrm{t}}$ | $\mathbf{x}^{\mathrm{b}}$ | $\mathbf{x}^{\mathrm{a}}$ – ETKF$_{60}$ | $\mathbf{x}^{\mathrm{a}}$ – ETKF$_{60}$-GA$^2_{\theta,y}$ | $\mathbf{x}^{\mathrm{a}}$ – FDS-MKS | FDS-EKS | $\mathbf{x}^{\mathrm{a}}$ – FDS-IKS |
|---|---|---|---|---|---|---|---|
| CAM.cldfrc_rhminh | 0.80 | 0.75 | 0.76 | 0.82 | 0.81 | 0.76 | 0.69 |
| CAM.cldfrc_rhminl | 0.91 | 0.88 | 0.91 | 0.90 | 0.90 | 0.90 | 0.91 |
| CAM.ch4vmr | 791.6 | 800.0 | 801.0 | 798.2 | 798.7 | 797.4 | 800.5 |
| CAM.co2vmr | 284.7 | 300.0 | 300.0 | 301.2 | 298.8 | 301.4 | 302.5 |
| CAM.zmconv_c0_lnd | 0.0035 | 0.0202 | 0.0208 | 0.0167 | 0.0287 | 0.0238 | 0.0377 |
| CAM.zmconv_c0_ocn | 0.0035 | 0.0202 | 0.0155 | 0.0230 | 0.0290 | 0.0327 | 0.0478 |
| CAM.zmconv_ke | 1.0e-06 | 5.0e-06 | 2.3e-06 | 4.6e-06 | 4.1e-06 | 3.75e-06 | 3.02e-06 |
| POP2.bckgrnd_vdc1 | 0.16 | 0.19 | 0.18 | 0.19 | 0.19 | 0.19 | 0.18 |
| POP2.hmix_gm_nml.ah | 4.00e+07 | 4.20e+07 | 4.17e+07 | 4.30e+07 | 4.17e+07 | 4.30e+07 | 4.46e+07 |
| POP2.freshwater_gis | 0.0075 | 0.0 | 0.0038 | 0.0144 | -8.4e-04 | 6.28e-05 | 5.5e-04 |
| $\mathcal{J}_{\mathrm{o}}(\boldsymbol{\theta})$ | | 373.39 | 64.95 | 61.23 | 50.24 | 91.81 | 48.88 |
| $\mathcal{J}(\boldsymbol{\theta})$ | | 373.39 | 66.43 | 66.83 | 51.85 | 93.13 | 55.20 |

[1] Units as described in Table 1.

[2] ETKF subindex indicates the ensemble size. Cost function obtained by re-integration of the model with the mean updated parameters.

## 5 Conclusions

This study focuses on low-frequency climate field reconstruction (multidecadal and longer timescales) with comprehensive deterministic Earth system models (ESMs). Given the enormous computational requirements of this class of models, we evaluate two iterative schemes based on reduced-order control vectors and the Kalman filter as assimilation approaches for climate field reconstruction. The schemes use an explicit representation of the background-error covariance matrix, and the Kalman gain is based on finite difference sensitivity (FDS) experiments. As such, the schemes are computationally limited to the estimation of a low-dimensional control vector. The underlying assumption is so that a low-dimensional control vector and its background uncertainty, contanining the most sensitive variables for a given climate, can encapsulate most of the modelled internal and external climate variability. The control vector can contain parameterized errors in initial conditions, and parameters for the small-scale physics, as well as parameters for forcing and boundary condition errors (e.g.; a bias in a time-varying radiative constituent). In general, it is expected that errors in initial conditions are a low sensitive input for the low-frequency model climate response. Thus, these would be generally excluded from the control vector, which makes it relatively easier to keep its low-dimensionality.

The evaluated schemes are a FDS implementation of the iterative Kalman smoother (FDS-IKS, a Gauss-Newton scheme), and a so-called FDS-multistep Kalman smoother (FDS-MKS, based on repeated assimilation of the observations). We have conducted two assimilation experiments: (a) a simple 1D energy balance model (Ebm1D; which has an adjoint code) with present-day surface air temperature from the NCEP/NCAR reanalysis data as target, and (b) a multi-decadal synthetic case with the Community Earth System Model (CESM v1.2, with no adjoint). The methodological description and the first experiment serves to show that, under a strong constraint minimization and perfect-model framework, the FDS-IKS should converge to the

same minimum as incremental 4D-Var. Actually. in this experiment, the FDS-IKS converges substantially faster than 4D-Var and to the same minimum. The FDS-MKS does not theoretically converges to the same minimum (except for linear cases), but it is more stable than the FDS-IKS for poorly regularized cost functions.

In a second experiment with CESM, given the lack of an adjoint code, we included an ETKF (with $m = 60$ ensemble members) and an ETKF with Gaussian Anamorphosis (ETKF-GA), as nonlinear estimation approach alternative to 4D-Var, as benchmarking schemes. As far as the authors know, this is also the first time that the ETKF-GA is evaluated with a comprehensive ESM for multidecadal analysis. Regarding the cost function as performace criterion, the ETKF-GA was not clearly better nor worse than a standard ETKF. We have shown that the GA does not solve the strong nonlinearities which sensitivities may find at some observations. A clear example is the North Atlantic SST. The results cannot be extrapolated, for example, to shorter term climate analyses. Also, alternative anamorphosis strategies for low-frequency analysis could show an improvement in the assimilation due to the transformations. With relatively high perturbations, both FDS schemes resulted (with about half the computing cost) in lower cost function values than the ETKF and the ETKF-GA. We would expect more optimal (likely smaller) perturbations adapted to individual control variables to result in further improvement in the FDS schemes, mostly the FDS-IKS (which, according to the first experiment, should be less resilient to the perturbation size). Given the computational requirements of comprehensive ESM and current HPCs, the experiments here indicate that the FDS iterated schemes can be a relatively efficient strategy for dealing with the nonlinear relation between model inputs and the paleoclimate proxy observations. These nonlinearity being introduced by both the ESM and the forward operators (*proxy system models* in general). Out for these experiments, the general impresion is that one would choose the FDS-IKS over FDS-MKS as it should converge to the same minimum than incremental 4D-Var under the given assumptions. However, the experiments indicate that initially damped increments, as those with FDS-MKS, should improve the convergence. So, FDS-MKS iterations (one or two) could be used initially to update the control vector estimate (but not the covariance), and then be followed by FDS-IKS iterations. Note that alternatively a linear search along the same direction provided by the FDS-IKS could also be possible. Further evaluation of the possibilities is needed.

This study is a first attempt to use the described iterated schemes for assimilation with comprehensive ESMs and multidecadal and longer timescales. It has provided the context of the problem, described the schemes and conducted preliminary experiments. The study is limited by the same computational constraint that motivate itself. Further study is in clear need before this type of schemes can be applied soundly for low-frequency past climate analyses in real cases. This would include sensitivity analyses for control vector design, error compensation analyses, and the inclusion of model error (e.g.; Sakov and Bocquet, 2018; Sakov et al., 2018). In addition, the other paleo-assimilation issues, summarised in the paper, regarding the model-data comparison and the observational-error characteristics, whose discussion goes beyond the scope of this study, need to be considered.

*Code and data availability.* The code for the Ebm1D model is being made available as part of this paper. Data for both experiments are available upon request to the authors. CESM v.1.2 is available at http://www.cesm.ucar.edu. The authors are preparing the assimilation code for CESM to be made publicly available. In the interim time, this code is also available upon request to the authors.

## Appendix A: Experiment 1: convergence analysis for finite difference schemes

Here we give tabulated details and a summary of the convergence tests for the finite difference schemes in the experiment with the model Ebm1D. Table A1 shows the convergence results. The tests have the naming code SCN_FFFFxIT_mNN_sdfacSSSS, where

- SCN: weight scenario for the cost term $J_o$ in Eq. (35), with the alternatives:

    - PD1: present-day scenario. $\sum_{i=1}^{p} w_i = 1$
    - PD2: present-day scenario. $\sum_{i=1}^{p} w_i = 3$
    - PD3: present-day scenario. $\sum_{i=1}^{p} w_i = 5$

- FFFF: assimilation scheme, with the following options:

    - pIKS: FDS-IKS
    - pMKS: FDS-MKS

- IT: Number of iterations. Fixed for FDS-MKS. Maximum for FDS-IKS.

- NN: number of perturbations $m_\theta$ for each control variable. Including the estimate at each loop, the total ensemble size is $m = m_\theta \times q + 1$, where $q$ is the dimension of the control vector.

- SSSS: $1000 \times sdfac$. For example, SSSS code '0010' indicates $sdfac = 0.010$. For $m_\theta = 1$ (forward finite differences), the perturbation of each control variable for sensitivity estimation is $sdfac \times \sigma_{\theta_i}$, where $\sigma_{\theta_i}$ is the standard deviation of the control variable. For $m_\theta > 1$, perturbations for each control variable are drawn from $\mathcal{N}(0, (sdfac \times \sigma_{\theta_i})^2)$.

The tests are considered to evaluate the resilience of the Gauss-Newton scheme (FDS-IKS) to high perturbations and decreasing regularization, and how this affects the relative performance of the Gauss-Newton scheme versus the multistep scheme (FDS-MKS). Specific cost function values are to be compared quantitatively only within a specific weight scenario (PD1, PD2, or PD3). For higher weights (being PD3 the highest), the effect of regularization by the background term decreases, which increases the chances of the FDS-IKS not converging (STOPPED tests due to unstable model integrations). Further tests (not shown) with even higher observational weight than PD3 made more and more difficult for the FDS-IKS to converge.

Scheduling of computing resources is more uncertain with the FDS-IKS than with the FDS-MKS. However, regarding this experiment and model, when the FDS-IKS converges, the values of the cost function are lower than those of the corresponding FDS-MKS test. This happens for FDS-MKS with either 2 or 3 iterations. Thus, with adequate regularization, the FDS-IKS is favoured. With decreasing observation uncertainty (decreasing regularization), the FDS-MKS stays more stable (see PD2 and PD3 tests). Still, for low regularization (PD3) the FDS-MKS cost function values ($\sim 46$ for the 2-step FDS-MKS, and $\sim 43$ for the 3-step FDS-MKS) are higher than those by the FDS-IKS when it converges ($\sim 39$). The 3-step FDS-MKS always obtains lower values that the corresponding 2-step FDS-MKS, but (generally) the differences are not very high. Interestingly, in some

**Table A1.** 1D energy balance model experiment. Cost function values for the tests with the FDS schemes[1].

|  |  | 1 | 2 | 3 | 4 | 5 | 6 | 7 |
|---|---|---|---|---|---|---|---|---|
| 1 | PD1_pIKSx10_m01_sdfac0001 | 14.21 | 11.56 | 9.52 | 9.48 | 9.47 | 9.47 | 9.47 |
| 2 | PD1_pIKSx10_m01_sdfac0010 | 14.21 | 11.57 | 9.50 | 9.47 | 9.47 | 9.47 | 9.47 |
| 3 | PD1_pIKSx10_m01_sdfac0100 | 14.21 | 13.25 | 9.49 | 9.48 | 9.48 | 9.48 | 9.48 |
| 4 | PD1_pMKSx02_m01_sdfac0001 | 14.21 | 10.56 | 9.61 | | | | |
| 5 | PD1_pMKSx02_m01_sdfac0010 | 14.21 | 10.56 | 9.61 | | | | |
| 6 | PD1_pMKSx02_m01_sdfac0100 | 14.21 | 10.86 | 9.57 | | | | |
| 7 | PD1_pMKSx03_m01_sdfac0001 | 14.21 | 10.44 | 9.75 | 9.55 | | | |
| 8 | PD1_pMKSx03_m01_sdfac0010 | 14.21 | 10.45 | 9.75 | 9.55 | | | |
| 9 | PD1_pMKSx03_m01_sdfac0100 | 14.21 | 10.56 | 9.76 | 9.53 | | | |
| 10 | PD2_pIKSx10_m01_sdfac0001 | 42.63 | 46.90 | 27.45 | 25.45 | 25.33 | 25.32 | 25.32 |
| 11 | PD2_pIKSx10_m01_sdfac0010 | 42.63 | 47.52 | 26.74 | 25.39 | 25.33 | 25.32 | 25.32 |
| 12 | PD2_pIKSx10_m01_sdfac0100 | 42.63 | 186.80 | 25.71 | 25.41 | 25.34 | 25.33 | 25.33 |
| 13 | PD2_pMKSx02_m01_sdfac0001 | 42.63 | 33.72 | 27.58 | | | | |
| 14 | PD2_pMKSx02_m01_sdfac0010 | 42.63 | 33.74 | 27.62 | | | | |
| 15 | PD2_pMKSx02_m01_sdfac0100 | 42.63 | 38.93 | 27.36 | | | | |
| 16 | PD2_pMKSx03_m01_sdfac0001 | 42.63 | 32.05 | 27.55 | 26.79 | | | |
| 17 | PD2_pMKSx03_m01_sdfac0010 | 42.63 | 32.06 | 27.52 | 26.78 | | | |
| 18 | PD2_pMKSx03_m01_sdfac0100 | 42.63 | 34.13 | 27.41 | 26.61 | | | |
| 19 | PD3_pIKSx10_m01_sdfac0001 | 71.05 | STOPPED | | | | | |
| 20 | PD3_pIKSx10_m01_sdfac0010 | 71.05 | 141.33 | 44.54 | 39.63 | 39.26 | 39.26 | 39.26 |
| 21 | PD3_pIKSx10_m01_sdfac0100 | 71.05 | STOPPED | | | | | |
| 22 | PD3_pMKSx02_m01_sdfac0001 | 71.05 | 60.35 | 46.24 | | | | |
| 23 | PD3_pMKSx02_m01_sdfac0010 | 71.05 | 60.49 | 46.29 | | | | |
| 24 | PD3_pMKSx02_m01_sdfac0100 | 71.05 | 83.14 | 48.33 | | | | |
| 24 | PD3_pMKSx03_m01_sdfac0001 | 71.05 | 55.35 | 44.86 | 43.32 | | | |
| 26 | PD3_pMKSx03_m01_sdfac0010 | 71.05 | 55.38 | 44.80 | 43.25 | | | |
| 27 | PD3_pMKSx03_m01_sdfac0100 | 71.05 | 62.89 | 44.44 | 43.02 | | | |
| 28 | PD3_pIKSx10_m02_sdfac0001 | 71.05 | STOPPED | | | | | |
| 29 | PD3_pIKSx10_m02_sdfac0010 | 71.05 | 122.35 | 42.79 | 39.41 | 39.37 | 39.26 | 39.26 |
| 30 | PD3_pIKSx10_m02_sdfac0100 | 71.05 | 120.35 | 40.01 | 39.54 | 39.77 | 39.34 | 39.76 |

[1] Background cost function value in first column, followed by values along iterations. STOPPED indicates non-convergence (unstable model integration).

instances, the second step of the 3-step FDS-MKS has already lower cost function values than the final estimation of the 2-step FDS-MKS. This effect increases with decreased regularization. When the FDS-IKS starts to have convergence problems (see PD3 tests), increasing the ensemble size by conducting 2 perturbations per parameter (akin to central finite differences), makes the FDS-IKS more stable. Higher number of perturbations (not shown) make the scheme more stable, but this would not be practical for long term analyses with comprehensive ESMs.

## 5   Appendix B:  Experiment 2

### B1   Example of well behaved Gaussian anamorphosis

As an example, Fig. B1 shows scatterplots between the parameter CAM.cldfrc_rhminl and SST for a location in the South Pacific in the limits of the Antartic circumpolar current. The location is shown in 5. Scatterplots are shown for the original variables and those with a Gaussian anamorphosis transformation. As opossed to Fig. 4, in this case Gaussianity is improved in the anamorphosed SST, and also the scatterplots show that an implicit pseudolinearization appears between the two anamorphosed variables, mostly as a result of the SST transformation.

### B2   Convergence of the FDS-IKS and posterior standard deviations

Table B1 shows the convergence of the cost function for the FDS-IKS in the CESM experiment. Table B2 shows the posterior standard deviations of the control vector for the iterated FDS schemes, as well as for the ETKF and the ETKF-GA.

**Table B1.** CESM parameter estimation: convergence of FDS-IKS cost function.

| COMP.name | $\mathbf{x}^{\mathrm{b}}$ | $\mathbf{x}^{\mathrm{a}}$ – FDS-IKS$_1$ | $\mathbf{x}^{\mathrm{a}}$ – FDS-IKS$_2$ | $\mathbf{x}^{\mathrm{a}}$ – FDS-IKS$_3$ |
|---|---|---|---|---|
| $\mathcal{J}_{\mathrm{o}}(\boldsymbol{\theta})$ | 373.39 | 91.81 | 63.50 | 48.88 |
| $\mathcal{J}(\boldsymbol{\theta})$ | 373.39 | 93.13 | 69.21 | 55.20 |

[1] Units as described in Table 1.

[2] Subindices in FDS-IKS refer to loop number. First iteration, FDS-IKS$_1$ is also the FDS-EKS.

**Table B2.** CESM posterior standard deviation in control variables.[1]

| COMP.name | $\sigma(\mathbf{x}^{\mathrm{b}})$ | $\sigma(\mathbf{x}^{\mathrm{a}})$ – ETKF$_{60}$ | $\sigma(\mathbf{x}^{\mathrm{a}})$ – ETKF$_{60}$-GA | $\sigma(\mathbf{x}^{\mathrm{a}})$ – FDS-MKS | $\sigma(\mathbf{x}^{\mathrm{a}})$ – FDS-IKS |
|---|---|---|---|---|---|
| CAM.cldfrc_rhminh | 5.00e-02 | 2.91e-02 | 2.38e-02 | 8.93e-03 | 1.19e-02 |
| CAM.cldfrc_rhminl | 5.00e-02 | 6.99e-03 | 6.97e-02 | 2.13e-04 | 1.48e-03 |
| CAM.ch4vmr | 4.00 | 3.37 | 0.97 | 1.12 | 0.85 |
| CAM.co2vmr | 3.00 | 2.67 | 1.54 | 1.69 | 0.42 |
| CAM.zmconv_c0_lnd | 1.20e-02 | 1.03e-02 | 0.79e-02 | 3.68e-03 | 1.44e-03 |
| CAM.zmconv_c0_ocn | 1.20e-02 | 9.34e-03 | 3.51e-03 | 3.79e-03 | 8.28e-04 |
| CAM.zmconv_ke | 2.20e-06 | 1.28e-06 | 1.01e-06 | 3.68e-07 | 4.59e-07 |
| POP2.bckgrnd_vdc1 | 2.00e-02 | 1.76e-02 | 1.12e-02 | 3.31e-03 | 3.88e-03 |
| POP2.hmix_gm_nml.ah | 2.00e+06 | 1.54e+06 | 1.71e+05 | 6.95e+05 | 3.89e+05 |
| POP2.freshwater_gis | 5.00e-03 | 3.17e-03 | 1.66e-03 | 1.56e-03 | 1.16e-03 |

[1] Units as described in Table 1.

[2] ETKF subindex indicates the ensemble size.

*Competing interests.*  The authors declare that they have no conflict of interest.

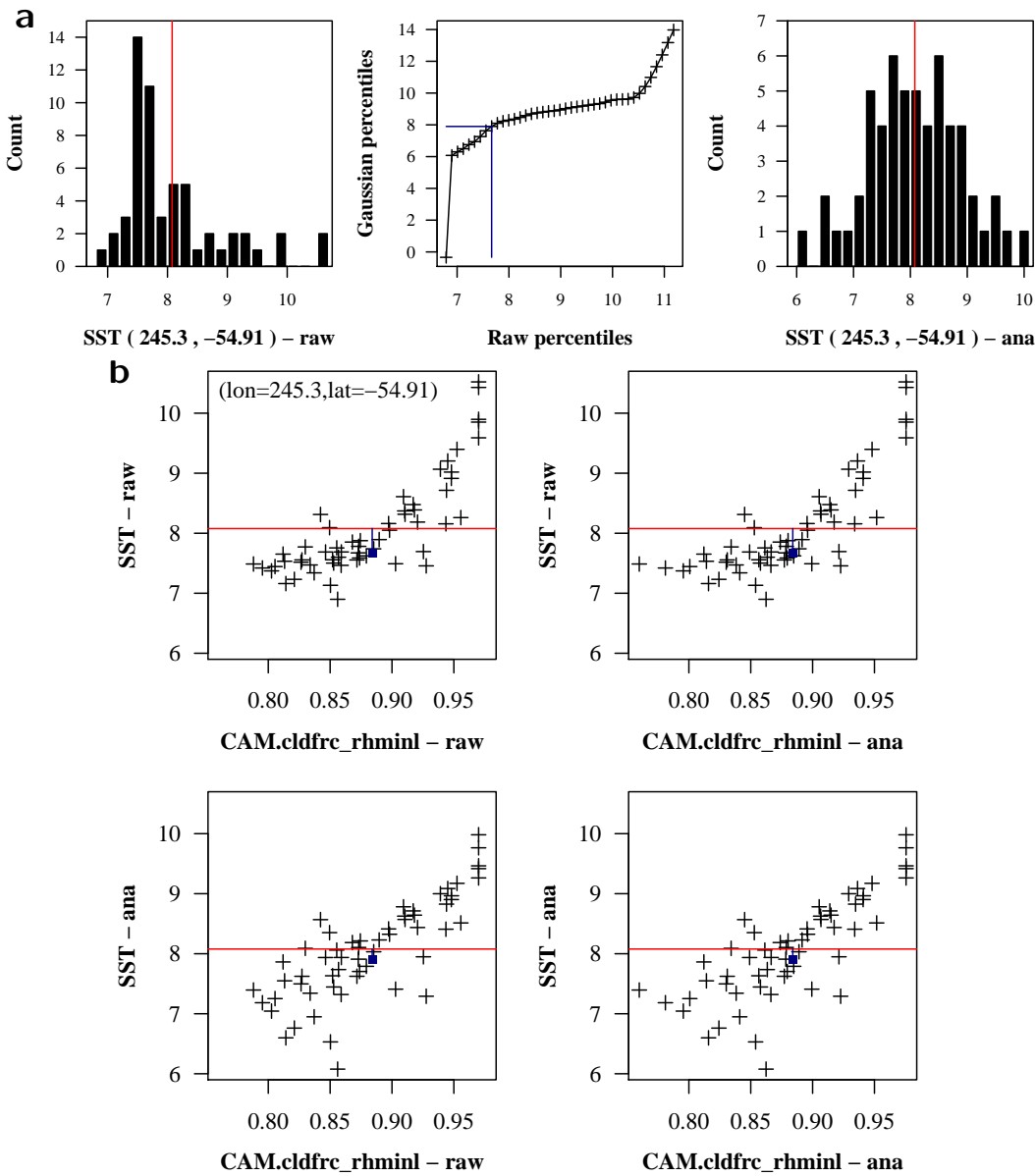

**Figure B1.** CESM experiment. Example of Gaussian anamorphosis transformation in a control variable (minimum relative humidity for high stable cloud formation) and the sea surface temperature (SST) for both the model equivalent of the observations and the observation (transformation details in text) at a location (245.3°E, 54.91°S) in the South Pacific (location shown in 5). Further description of symbols is as Fig. 4.

*Acknowledgements.* This work was supported by the PALMOD project (01LP1511D) sponsored by the German Ministry of Education and Research (BMBF). The authors acknowledge the North-German Supercomputing Alliance (HLRN) for providing HPC resources that have

contributed to the research results reported in this paper. We thank two anonymous reviewers for their positive comments, which have been very helpful to improve the manuscript.

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
