# Peer review of "Evaluation of iterative Kalman smoother schemes for multi-decadal past climate analysis with comprehensive Earth system models"

_Geoscientific Model Development, 2018_

## Referee Comment (RC1) · Anonymous Referee #1 · 7 Jun 2018

The authors present innovative low-cost strategies for online model parameter estimation, which can potentially be applied for climate field reconstruction with coupled GCMs. The manuscript is written in a transparent way and authors explain very well all the assumptions used in their study. The paper moves the time-averaged data assimilation efforts a small but very important step forward. I recommend publishing the paper after minor revision.

**General Comments:**

**1) Scientific significant:**

The manuscript represent a substantial contribution to modeling science within the scope of this journal. The ideas and methods are original.

**2) Scientific quality:**

The scientific approach and applied methods are valid and the assumptions are well introduced. However, the results are not discussed in a balanced way and could be improved (see below). The models, technical advances and/or experiments described have the potential to perform calculations leading to significant scientific results.

**3) Scientific reproducibility**

The modeling science seems not to be reproducible. I think if the authors share their code, this problem will be solved. However, their methodology is well described and traceable.

**4) Presentation quality**

The presentation quality is fair and could be improved in a new version. Number of figures presenting the results can be revised.

**5)** Overall, the manunscript is understandable for experts working in the field but not easy to follow for general readers. There are too many acronyms in the manuscript. Please spell out when possible.

**6)** Given that this paper belongs to the category of "Development and technical papers" (see:https://www.geoscientific-model-evelopment.net/about/manuscript_types.html#item2), I encourage the authors to make their code available online (at least for the 1D experiment). This might help the community very much and improve the code itself. According to the journal policies, the authors have to include the model's version in the title (e.g., Model XXX (version Y)).

**7) Page1Line3 (P1L3):** Explain how model's parameters have relationship with proxies!

**8) P1L6-7** is too complicated!

**9) P1L6 :** Authors mix two approaches: model's parameter estimation (atmosphere and ocean) and fresh water melting parameters estimation. Have they done separate experiments? They describe, the latter might be essential for North Atlantic circulations. However, there might be nonlinear relationships between these two and they could contaminate each other. Which one impacts the error reduction larger?

**10) P2L12:** Could you provide references for that?

**11) P2L32-33:** How should one do this? Reference?

**12) P2L5:** Explain the "generally positive results"!

**13) P3L20:** Aren't the parameters not being updated at each DAW? Aren't they time varying when the new observation is available? How could one do that in future projections without observations? What are the challenges? How could tuning the model for the past improve projections? Please clarify!

**14) P6L10:** Explain briefly the gradient descent algorithm, learning rate, number of iterations, etc...

**15) P8L6-7:** However, aren't the parameters updated based on time-averaged obs?

**16) P11L18-19:** How do you define the learning rate in gradient descent?

**17) P15L19:** Why analyzing only 10 years? 90 years for spin up of 1d model?

**18) P18L27:** "not shown", but is interesting to put in supplementary.

**19) P19L2 :** is it a typical set-up of CESM?

**20) P20L9:** Could you explain and discuss the problems of multi-component DA in your set-up? Differences in time-scales of proxies, etc.

**21) P20L13:** Have you done other experiments with other sets of parameters? For example more or fewer numbers of parameters?

**22) P24L10:** In figure 3 it is really hard to see the differences. Maybe centering the colorbar with zero might help. How do you explain that the error reduction is due to DA and not the lack of sensitivity of the model to perturbation of the parameters. For example Figure 4 upper left panel shows that the model is not sensitive to changes of cldfrc_rhminl in Arctic and Antarctic regions.

**23)** You focus on the ocean where the observations where assimilated, how about the atmospheric variables? Is there any error reduction happening there? Could you show for example global T2m quantities?

**24)** Figures similar to Fig.4 for other perturbed parameters could be shown in the supplementary. This will clarify the sensitivity of CESM.

**Specific Comments:**

**1) P3L6**: "in section?"

**2) P3L7:** "problem of CFR" which problem?

**3) P22L12:** How about the uneven time resolution of observations?

**4) figure 2, 3, 4:** for the results one has to switch between figure 2, 2 and 4 to follow the line of discussions. You could at least put the observation locations on figure 3 and 4. Note that the land-sea mask is also shifted between the figure 2 and 3-4.

---

## Referee Comment (RC2) · Anonymous Referee #2 · 19 Jun 2018

**Review of "Parameter space Kalman smoothers for multi-decadal climate analysis in high resolution coupled Global Circulation Models"**

by Javier García-Pintado and André Paul

18 June 2018

This is a very interesting study, supported by a substantial amount of work, both theoretical and numerical. The theoretical and experimental parts are well balanced; the experiments are well chosen and interesting. The manuscript certainly should be considered for publication. Nevertheless, I believe it has a few flaws and requires significant improvements before being acceptable for publication.

In my opinion, the main issues are:

1. The manuscript is too long. Several discussions are unnecessary and frankly a bit wordy, especially in the introductory parts.

2. There are quite a few typos that need to be corrected.

3. I believe that there is no need to give these methods new names just because they are applied to parameters. But this is certainly up to the authors.

4. Please avoid the use of the term "adjoint method" which is – as of today – rarely used, ill-defined or at best does not precisely correspond to 4D-Var. See comments below.

5. The synthetic experiments with the CESM is a great piece of work, but unfortunately quite inconclusive. This is unexpected since we could have hoped for clear and neat results with such experiments. Tables 4 and 5 point to uncleared problems in the experiments. It is nice to use an ETKF in conjunction with a Gaussian anamorphosis. However, if the outcome is inconclusive (is it?), then it casts doubts on the interest of such test, or more likely on the implementation of the method (bugs?). See additional comments below (sorry for the redundancy).

6. It would be worth introducing in the CESM synthetic experiment some additional model error that you do not control in order to check how the methods are compensating for this error. This would be realistic and convincing.

7. Why is the data assimilation code not available? I thought it would be mandatory to do so for GMD, is it?

List of remarks and suggestions, some pertaining to the main criticisms:

1. page 1, l.4-5: "In a model framework where we assume that model dynamic parameters account for (nearly) all forecast errors at observation times,": Right, but is this framework usually met?

2. page 1, 12: "are evaluated in numerical experiments": Are these twin/synthetic experiments? In other words do you use real observations or synthetic ones? It is necessary to mention it here in the abstract.

3. Page 1, l.14: "the pFKS obtains a cost function": This expression seems meaningless. Please rephrase.

4. Page 1, l.14: the expression "adjoint method" should be avoided as it is not well defined.

5. Page 1, l.14-15: Frankly, the whole sentence "Firstly, with Ebm1D the pFKS...behaves slightly worse." is difficult to understand, especially in an abstract. (For me, the technical terms are not the problem, since I am fluent in them.)

6. Page 1, l.17: You have to explain in the abstract why you would use an ETKF with a Gaussian anamorphosis or not mention it at all.

7. Page 1, l.18: Having the lowest cost function value is rarely a criterion as it depends much on the prior used in the cost function.

8. Page 1, l.21: "The issue of fusing data into models arises in all scientific areas that enjoy a profusion of data.": Not really. This is specific to areas where costly models are used!

9. Page 1, l.23-24: "Such methods can be considered as an approach for interpolating or smoothing a data set in space and time where a model acts as a dynamical constraint (Evensen, 1994a)": I don't believe you should use such outdated comment, all the more since nowadays there is a general consensus on a Bayesian view on data assimilation/inverse problems.

10. Page 2, l.16-19: "Other geophysical applications share this relevance of model parameters on the assimilation problem, as the estimation of distributed parameters and state for multiphase flow in petroleum reservoirs (e.g.; Gu and Oliver, 2007; Oliver et al., 2011), or hydraulic tomography for groundwater applications (e.g.; Schöniger et al., 2012).": You should mention atmospheric chemistry first, all the more since it quite close to climate (e.g., Bocquet et al., 2015).

11. Page 2, l.20: "A related issue is the enforcement of physically based conservation laws, which by default is not taken into account by (ensemble) Kalman filters." No! You are right in general, but all linear constraints are properly enforced. (Which is why the use of the EnKF is widely spread!)

12. Page 2, l.23: ";"

13. Page 2, l.23: "confirming re-integration": This is unclear to me. Please clarify.

14. Page 2, l.30: "under the assumption the errors" $\longrightarrow$ "under the assumption that the errors"

15. Page 3, l.3: "conduct" $\longrightarrow$ "conducted"

16. Page 3, l.6: "in section ,": Section number is missing.

17. Page 3, l.9: "adjoint method": please avoid this expression. It does not correspond to anything rigorous.

18. Page 3, l.17: "opposed" $\longrightarrow$ "as opposed"

19. Page 3, l.17-18: This is an outdated view. Today, it is considered a doable task to estimate uncertainty within a variational framework (this is actually operational at the ECMWF). Read for instance Bousserez et al. (2015).

20. Page 3, l.24-25: "Other than that the formulation is identical than it would be for the corresponding filtering versions.": unclear or awkward.

21. Page 3, l.31: Twin experiments? This should be mentioned here as well.

22. Page 4, l.2: "the not only" $\longrightarrow$ "not only"

23. Page 4, l.18-19: "We also assume that the model is weakly nonlinear, such that it can be linearized.": This is not a clear statement. Any smooth model (even very nonlinear ones!) can be linearised.

24. Page 4, l.20: "small": do you mean low-dimensional?

25. Page 4, l.28: "The problem is to fit three spatial dimensions in time.": the sentence is unclear. "in" ⟶ "and"?

26. Page 5, l.1: Assuming time-invariant system is very restrictive in climate models where most forcings are time-dependent. Please justify.

27. Page 5, l.6: "That is, that the system..." ⟶ "That is, the system..."

28. Page 5, line 23: "in 4D-Var then" ⟶ "in 4D-Var is then"?

29. Page 5, line 23: "non-linear" ⟶ "non-quadratic"

30. Page 7, line 6: "is the same that" ⟶ "is the same as"?

31. Page 7, line 7: "4D-Var, IKS" ⟶ "4D-Var, the IKS"

32. Page 7, Eq.(16) and around: Such an operator exists only if the observations are time-averaged values, right? In general observations will depend on the initial condition. This must discussed (this is actually better discussed in the introduction!).

33. Page 7, line 8: What is a "quasi-equilibrium"?

34. Page 7, line 14: In my opinion, there is no need to introduce a new term. This is just an IKS in parameter space.

[Figure]

35. Page 8, Eq.(17): This type of formulation is frequent in many areas of geo-sciences; there is no need to look as far as history matching in oil reservoir modelling. For instance, this is very often met in source/fluxes inverse problems in atmospheric chemistry.

36. Page 8, line 13: "While it": A typo?

37. Page 9: In my opinion the discussion on the computation of the sensitivities is not only convoluted but also not very useful. It is obvious to the reader (to me at least), that you will use finite-differences in the end. Essentially only the last paragraph of section 3.2 is needed.

38. Page 9, Eq.(22): The linearisation in parameter space should be carried out at the j-th estimate of the parameters, no the background parameters (except for j=1). What you wrote is just an approximation, which would make the iterative approach not as accurate as expected. Please clarify.

39. Page 10, line 20: "Iterative linear methods" is awkward, even though I guess I understand what you mean.

40. Page 10, line 24: Parentheses are needed around Bell and Cathey (1993).

41. Page 10, beginning of section 3.3: I don't see the point in the discussion with the EnRML. You can probably do without it.

42. Page 11, Eq.(25): The notation is unclear (I understand but many colleagues would not) and should be made consistent with Eq.(22).

43. Page 11, line 22: Actually the use of the MDA trick is slightly different in Bocquet and Sakov (2014) than in Emerick and Reynolds (2013), because the weights are adjusted over several data assimilation cycles.

44. Another reference relevant to your manuscript is a study of the iterative ensemble Kalman smoother applied to a joint state and parameters estimation problem (Bocquet and Sakov, 2013).

45. Page 12, line 21: "opposite to" ⟶ "as opposed to"

46. Page 13, line 21: The sentence is a bit ambiguous since the model integration is part of the analysis (and so-to-speak a part of the analysis!). Please reformulate.

47. Page 13, line 26: What is a "temporal solution"?

48. Page 13, line 27: "detect linearity assumption": This expression is unclear. Please rephrase.

49. Page 13, line 3: I am familiar with the Levenberg-Marquardt scheme(s) and I do not understand your sentence!

50. Page 13, line 12-14: You have to give more details of your implementation. First, I do not see why you would need localisation for the state variables, since you are not updating them. Second, it is well known that, without a few tweaks, one cannot update global parameters in a LETKF.

51. Page 14, line 21: "It is not standard, however, how the GA should be applied in the context of DA.": There have been reviews and papers about that; for instance Bertino et al. (2003), as you rightfully mentioned, but also Bocquet et al. (2010); and above all Simon and Bertino (2009) and Béal et al. (2010) who set the standard on this topic. As far as I can understand, you are using their method. Please amend.

52. Page 15, l.9: "adjoint method (4D-Var)" ⟶ "4D-Var (based on the adjoint)"

53. Page 15, l.11, L.13, l.27: Please avoid the "adjoint method" expression which is really outdated, and not use in data assimilation study. Refer instead to 4D-Var or variational method, possibly mentioning the use of the adjoint model.

54. Page 15, l.24, "standard 4D-Var applications" ⟶ "standard in 4D-Var applications"?

55. Page 15, l.29-30: I do not understand the last sentence.

56. Page 15-16, section 4.1: Where did you describe the parameters and how many are they? This is absolutely key to the feasibility of the problem. There are tables; but the parameter should be more clearly discussed in the text.

57. Page 17, line 4: "Note the original" ⟶ "Note that the original"

58. Page 17-18, section 4.3: In this section, you keep referring to the "ajoint method". Please do not use this term. This is a loose term, used in a loose way which generates confusion. At best, it refers to the computation of the gradient via the adjoint model, and not to the optimisation method you actually imply. That is why it is not used in written texts of the data assimilation community. You even refers on page 18 to the "adjoint", a short-cut which definitely lacks rigour.

59. Page 18, lines 1-14: It seems that it all boils down to the presence or absence of a prior for the parameters. Isn't it? If this is so, then this discussion is not really focused on what it should be.

60. Page 20, line 9: "multi-component data assimilation": To the best of my knowledge/understanding, this is rather called "strongly coupled data assimilation".

61. Page 20-21: I would more precisely enumerate/list/discuss the control variables. For instance, at some point, clearly mention: "Hence, our first control variable is..." etc.

62. Page 23, line 26: "that is could be" ⟶ "that it could be"

63. Page 22-24, section 5.5: The results are not very enlightening. This is frustrating since we were expecting clear and neat results in such a controlled synthetic experiment. It might point to problems with these experiments (bugs?, too weak sensitivities, hence bad conditioning?). The reduction of uncertainty, as seen in table 5, does not seem consistent (too high) with the estimates reported in table 4 compared to the truth. This is worrisome.

64. Page 25, line 16: "and the trust one" ⟶ "and the true one"?

65. Page 25, line 25-26: "The estimation of the flux correction in our example has not succeed for the pFKS.": I do not understand the sentence. Please rephrase.

66. Page 26, line 9: "(or adjoint method)": no, rigorously, 4D-Var is not and should not be called the "adjoint method".

67. Page 29, line 7-8: I am surprised that you do not make your data assimilation codes available. I thought this was a mandatory rule for a potential GMD publication.

**References**

Béal, D., Brasseur, P., Brankart, J.-M., Ourmières, Y., and Verron, J.: Characterization of mixing errors in a coupled physical biogeochemical model of the North Atlantic: implications for nonlinear estimation using Gaussian anamorphosis, Ocean Sci., 6, 1–16, 2010.

Bell, B. M. and Cathey, F. W.: The iterated Kalman filter update as a Gauss-Newton method, IEEE Transactions on Automatic Control, 38, 294–297, 1993.

Bertino, L., Evensen, G., and Wackernagel, H.: Sequential data assimilation techniques in oceanography, Int. Stat. Rev., 71, 223–241, 2003.

Bocquet, M. and Sakov, P.: Joint state and parameter estimation with an iterative ensemble Kalman smoother, Nonlin. Processes Geophys., 20, 803–818, doi:10.5194/npg-20-803-2013, 2013.

Bocquet, M. and Sakov, P.: An iterative ensemble Kalman smoother, Q. J. R. Meteorol. Soc., 140, 1521–1535, doi:10.1002/qj.2236, 2014.

Bocquet, M., Pires, C. A., and Wu, L.: Beyond Gaussian statistical modeling in geophysical data assimilation, Mon. Wea. Rev., 138, 2997–3023, doi:10.1175/2010MWR3164.1, 2010.

Bocquet, M., Elbern, H., Eskes, H., Hirtl, M., Žabkar, R., Carmichael, G. R., Flemming, J., Inness, A., Pagowski, M., Pérez Camaño, J. L., Saide, P. E., San Jose, R., Sofiev, M., Vira, J., Baklanov, A., Carnevale, C., Grell, G., and Seigneur, C.: Data Assimilation in Atmospheric Chemistry Models: Current Status and Future Prospects for Coupled Chemistry Meteorology Models, Atmos. Chem. Phys., 15, 5325–5358, doi:10.5194/acp-15-5325-2015, 2015.

Bousserez, N., Henze, D. K., Perkins, A., Bowman, K. W., Lee, M., Liu, J., Deng, F., and Jones, D. B. A.: Improved analysis-error covariance matrix for high-dimensional variational inversions: application to source estimation using a 3D atmospheric transport model, Q. J. R. Meteorol. Soc., 141, 1906–1921, doi:10.1002/qj.2495, 2015.

Emerick, A. A. and Reynolds, A. C.: Ensemble smoother with multiple data assimilation, Computers & Geosciences, 55, 3–15, 2013.

Simon, E. and Bertino, L.: Application of the Gaussian anamorphosis to assimilation in a 3-D coupled physical-ecosystem model of the North Atlantic with the EnKF: a twin experiment, Ocean Sci., 5, 495–510, 2009.

---

## Author Comment (AC2) · 5 Sep 2018

**Manuscript reference:** GMD-2018-48

**Title\*:** Evaluation of iterative Kalman smoother schemes for multi-decadal climate analysis with comprehensive Earth system models

**Date:** 05/09/2018

*Modified from the discussion paper

**Responses to Reviewer #2 (anonymous)**

We thank the reviewer for their positive comments, which will help improve the manuscript. Below we give each comment and describe how we are altering the manuscript to address the reviewer's concerns.

Let us note that indirectly prompted by Reviewer # 2, we have considered a new title for the manuscript as indicated above. In general, we are finishing a substantial rewriting of the manuscript, due to (a) specific request from the reviewers to shorten introductory parts and expand the results and discussions, and (b) reviewers' comments also have indirectly suggested us that some parts of the manuscript were in need of further explanation.

Thus the Introduction is now longer and the description of the paleoclimate context has been slightly expanded, but former section 2 (Problem definition) has been now dropped and compacted within the Introduction. The description of the nonlinear relation between the control variables and the observation space in the experiment with the Community Earth System Model (CESM) now receives more attention. The analysis with ETKF-GA (the Gaussian anamorphosis) is now more detailed, and the described scheme of the iterated Kalman smoother is now also included in the CESM experiment. Also, the first experiment, with the 1D energy balance model, has been updated with a new more adequate 4D-Var benchmarking.

**Responses**

**Rev#2:**

This is a very interesting study, supported by a substantial amount of work, both theoretical and numerical. The theoretical and experimental parts are well balanced; the experiments are well chosen and interesting. The manuscript certainly should be considered for publication. Nevertheless, I believe it has a few flaws and requires significant improvements before being acceptable for publication.

In my opinion, the main issues are:

1. The manuscript is too long. Several discussions are unnecessary and frankly a bit wordy, especially in the introductory parts.

**Auth:** We have done a thorough revision of the manuscript, with a substantial number of sentences rephrased and shortened/dropped. Alternative we have expanded the Introduction to give some context of paleoclimate and support the rationale for the assumptions and applicability of the given schemes. Given the comments, we have also expanded the section rgarding the tests with Gausian Anamorphosis, and conducted additional tests in both experiments (now including the IKS in the CESM twin case). Please see new manuscript.

**Rev#2:** 2. There are quite a few typos that need to be corrected.

**Auth:** Typos have been corrected as found.

**Rev#2:** 3. I believe that there is no need to give these methods new names just because they are applied to parameters. But this is certainly up to the authors.

**Auth:** In the same sense that the EnKF provides a low-rank representation of the background covariance matrix, making it akin to but different from the KF, the methods used here differ

from the standard KF (iterations apart) in that they are explicitly solved as a function of numerically-based local sensitivity analysis (LSA), where sensitivites from observations to input parameters are estimated (conditionally) one-by-one to each of the considered inputs

While the IEnKS also uses (in its different versions) a strategy to get local sensitivites (e.g., differing from the average ensemble sensitivites of the EnRML, and the batch-EnRML), it is a low-rank scheme. We feel it is adequate to prepend some tag to "KF" (or "KS" for smoother) to clarify the schemes. We chose "parameter-space" as an implicit way to indicate the way sensitivities were constructed. With the comment we have considered that it is likely better to prepend "NLS-" for "numerical local sensitivity-". Still we indicate the rationale for the labeling in the manuscript. On the other hand, to relate better to the "multiple data assimilation" strategy, we have replaced the "F" for fractional by an "M" for multistep in the formerly named pFKS.

With this, the names now used for the schemes are NLS-IKS (numerical local sensitivity, iterative Kalman smoother) and NLS-MKS (numerical local sensitivity, multistep Kalman smoother). Let us insist that the aim is not to claim these are fundamentally new filters, but to clarify their specificity. We hope the reviewer agrees with the need for clarification and that a labelling is suited.

**Rev#2:** 4. Please avoid the use of the term "adjoint method" which is – as of today – rarely used, ill-defined or at best does not precisely correspond to 4D-Var. See comments below.

**Auth:** In general, although we explicitly referred to "4D-Var" in Section 3 regarding the methodological description, we chose to use the "adjoint" term in other parts (mostly in Section 4), to comply with the terminology used by Paul and Losch (2012) –--for which this section was considered as an extension---, and by some oceanographers and climate scientists (e.g., regarding terminology, see Wunsch and Heimbach, 2013). We understand, however, the reviewer's point and have now used the "4D-Var" term throughout the manuscript.

**Rev#2:** 5. The synthetic experiments with the CESM is a great piece of work, but unfortunately quite inconclusive. This is unexpected since we could have hoped for clear and neat results with such experiments. Tables 4 and 5 point to uncleared problems in the experiments. It is nice to use an ETKF in conjunction with a Gaussian anamorphosis. However, if the outcome is inconclusive (is it?), then it casts doubts on the interest of such test, or more likely on the implementation of the method (bugs?). See additional comments below (sorry for the redundancy).

**Auth:** The main conclusion of the CESM experiment is that the iterative/recursive schemes based on simple numerically-estimated local sensitivities (even with non-optimal perturbations) are more able to deal with the non-linear relation between the inputs and the observation space that a single linear step (in the experiment), even if this is based on a denser sampling of the prior input PDF (as the EnKF). We acknowledge that more detailed experimentation is needed.

We have not found any bug in the Gaussian Anamorphosis (GA) as implemented. However, given the comment we felt that some clarification was needed. We have so provided further explanation on the specific GA analysis, including the justification of the tests, and an additional analysis where only the "inputs" (control variables) were transformed but the dual of the observation space (sea surface temperature, SST) was not (the formed GA transformed both in univariate way). Also, we have included additional plots regarding the transformations. Hopefully, these are informative about the non-linear relation between inputs and SST and how the two used forms of GA were able or not to provide a "pseudo-linearization". The plots also support a reason of why the GA (as applied in either case) was not completely successful despite general increase in univariate Gaussianity. Irrespective of the success of the GA

experiment, we consider it is worth reporting the results as support to guide/encourage further possible experimentation.

**Rev#2:** 6. It would be worth introducing in the CESM synthetic experiment some additional model error that you do not control in order to check how the methods are compensating for this error. This would be realistic and convincing.

**Auth:** We agree. For real applications, the selection of the (computationally feasible) control variables should be done based in sensitivity analysis. The control variables are responsible in the assimilation for all model errors, including compensation of model biases (in the tuned model) elsewhere. We have added a paragraph in the introduction to clarify this and that the purpose of included uncertain (deterministic) parameters in the control variables here is not to tune the model. The model is assumed to be previously tuned. A subset of the more "sensitive" but possibly uncertain parameters should be chosen as control vector to generate the background state needed for the assimilation. The purpose if the schemes is to produce mean climate field reconstructions for past climates at long scales. Differences between updated parameters and "tuned" values can be evaluated to diagnose the possible reasons for these. This is now clarified in the introduction.

Regarding the experiments, as a step-by-step approach we chose a scenario in which all sources of uncertainty were included in the control vector (a subset of model parameters, plus a freshwater flux term from Greenland and forcing from greenhouse gases). We understand this is the first time a fully coupled CESM is evaluated (even for a identical twin experiment) for the assimilation of data from a past climate multiproxy database (the MARGO Last Glacial Maximum in this case), and the experiments conducted have taken a considerable computing effort (possible thanks to HLRN III, the North Germany HPC). We have included some furthr test so that the IKS is now also evaluated with the CESM twin experiment. But it is not feasible for us to expand the analyses within the scope of this manuscript. We are looking forward (from ourselves or other colleagues in the community) for additional experiments, including the evaluation of specific error compensations (which will never be general).

**Rev#2:** 7. Why is the data assimilation code not available? I thought it would be mandatory to do so for GMD, is it?

**Auth:** We are making the DA and Ebm1D codes available. CESM v1.2 is already available.

**Rev#2:** List of remarks and suggestions, some pertaining to the main criticisms:

1. page 1, l.4-5: "In a model framework where we assume that model dynamic parameters account for (nearly) all forecast errors at observation times,": Right, but is this framework usually met?

**Auth:** In a past-climate context (paleo-climate) deterministic Earth System models (ESMs) converge to their own climatology and the memory of (reasonable) initial conditions is lost after some integration time. This assumption is related to the time footprint of paleoclimate proxy observations. We believe this comment is implicitly answered in the answers to comments 26 & 27 below.

**Rev#2:** 2. page 1, 12: "are evaluated in numerical experiments": Are these twin/synthetic experiments? In other words do you use real observations or synthetic ones? It is necessary to mention it here in the abstract.

**Auth:** Clarified that the first experiment uses "present-day surface air temperature from the NCEP/NCAR reanalysis data as target" and the second one (our object of study) is a synthetic experiment with the Community Earth System Model (CESM v1.2).

**Rev#2:** 3. Page 1, l.14: "the pFKS obtains a cost function": This expression seems meaningless. Please rephrase.

**Auth:** Removed.

**Rev#2:** 4. Page 1, l.14: the expression "adjoint method" should be avoided as it is not well defined.

**Auth:** Replaced by "4D-Var".

**Rev#2:** 5. Page 1, l.14-15: Frankly, the whole sentence "Firstly, with Ebm1D the pFKS...behaves slightly worse." is difficult to understand, especially in an abstract. (For me, the technical terms are not the problem, since I am fluent in them.)

**Auth:** Rewritten.

**Rev#2:** 6. Page 1, l.17: You have to explain in the abstract why you would use an ETKF with a Gaussian anamorphosis or not mention it at all.

**Auth:** Explained.

**Rev#2:** 7. Page 1, l.18: Having the lowest cost function value is rarely a criterion as it depends much on the prior used in the cost function.

**Auth:** Cost functions at each experiment use the same prior for each scheme. We indicate now that we focus here on the analysis step.

**Rev#2:** 8. Page 1, l.21: "The issue of fusing data into models arises in all scientific areas that enjoy a profusion of data.": Not really. This is specific to areas where costly models are used!

**Auth:** Modified to "...in scientific areas that enjoy a profusion of data and costly models are used."

**Rev#2:** 9. Page 1, l.23-24: "Such methods can be considered as an approach for interpolating or smoothing a data set in space and time where a model acts as a dynamical constraint (Evensen, 1994a)": I don't believe you should use such outdated comment, all the more since nowadays there is a general consensus on a Bayesian view on data assimilation/inverse problems.

**Auth:** We do not see why this comment by Evensen (1994a), which is a point of view, clashes with the Bayesian perspective. It is often echoed with similar wording in recent DA literature, while acknowledging the Bayesian view. In any case, in a now shortened introduction we have rewritten the paragraph and indicated now the Bayesian view of DA methods.

**Rev#2:** 10. Page 2, l.16-19: "Other geophysical applications share this relevance of model parameters on the assimilation problem, as the estimation of distributed parameters and state for multiphase flow in petroleum reservoirs (e.g.; Gu and Oliver, 2007; Oliver et al., 2011), or hydraulic tomography for groundwater applications (e.g.; Schöniger et al., 2012).": You should mention atmospheric chemistry first, all the more since it quite close to climate (e.g., Bocquet et al., 2015).

**Auth:** We have now given a short introduction to the context of Earth system modelling of climate, which is more relevant. Then, to shorten the manuscript, we have decided to remove the complete reference, which is more distant to the manuscript.

**Rev#2:** 11. Page 2, l.20: "A related issue is the enforcement of physically based conservation laws, which by default is not taken into account by (ensemble) Kalman filters." No! You are right in general, but all linear constraints are properly enforced. (Which is why the use of the EnKF is widely spread!)

**Auth:** Yes, but this is exactly the justification for the cited work of Janjic et al (2014) and other work, who deal with the incorporation of constraints in the EnKF to preserve mass, angular momentum and energy. Still, we have removed the comment to shorten the manuscript.

**Rev#2:** 12. Page 2, l.23: ";"

**Auth:** Done.

**Rev#2:** 13. Page 2, l.23: "confirming re-integration": This is unclear to me. Please clarify.

**Auth:** Clarified.

**Rev#2:** 14. Page 2, l.30: "under the assumption the errors" −! "under the assumption that

the errors"

**Auth:** Modified.

**Rev#2:** 15. Page 3, l.3: "conduct" −! "conducted"

**Auth:** Done.

**Rev#2:** 16. Page 3, l.6: "in section ,": Section number is missing.

**Auth:** Corrected.

**Rev#2:** 17. Page 3, l.9: "adjoint method": please avoid this expression. It does not correspond

to anything rigorous.

**Auth:** Changed to "4D-Var".

**Rev#2:** 18. Page 3, l.17: "opposed" −! "as opposed"

**Auth:** Corrected.

**Rev#2:** 19. Page 3, l.17-18: This is an outdated view. Today, it is considered a doable task to estimate uncertainty within a variational framework (this is actually operational at the ECMWF). Read for instance Bousserez et al. (2015).

**Auth:** The point here was to indicate B is 4D-Var is not evolving (although it could). The hybrid methods ---En4DVar----in operational centers (e.g. ECMWF, UKMO, GMAO, Meteo-France), use an ensemble, in several ways (as ensemble of 4Dvars, etc.),  for the flow-dependent term of B. We have removed the comment in any case.

**Rev#2:** 20. Page 3, l.24-25: "Other than that the formulation is identical than it would be for the corresponding filtering versions.": unclear or awkward.

**Auth:** Removed in new version.

**Rev#2:** 21. Page 3, l.31: Twin experiments? This should be mentioned here as well.

**Auth:** Done

**Rev#2:** 22. Page 4, l.2: "the not only" −! "not only"

**Auth:** Done.

**Rev#2:** 23. Page 4, l.18-19: "We also assume that the model is weakly nonlinear, such that it can be linearized.": This is not a clear statement. Any smooth model (even very nonlinear ones!) can be linearised.

**Auth:** This referred to the sense used in pp.65 in Tarantola (2005), where discussing various degrees on nonlinearity he refers to "forward equations that cannot be linearized, so the a posteriori probability density may be far from a Gaussian and special methods must be used...". In any case, we have removed this comment and the complete Section to shorten the manuscript as requested.

**Rev#2:** 24. Page 4, l.20: "small": do you mean low-dimensional?

**Auth:** Yes. Modified.

**Rev#2:** 25. Page 4, l.28: "The problem is to fit three spatial dimensions in time.": the sentence is unclear. "in" −! "and"?

**Auth:** Modified to: "The problem is to estimate the state of a past climate state along a time window for multidecadal and longer time scales."

**Rev#2:** 26. Page 5, l.1: Assuming time-invariant system is very restrictive in climate models where most forcings are time-dependent. Please justify.

**Auth:**

We consider two applications of the simplified described schemes: a) reconstruction of climate with equilibrium simulations (e.g.; mean (annual, and seasonal) climate reconstruction for the Last Glacial maximum) or mid-Holocene. Here, solar forcing (variability and orbital parameters) is inter-annually stationary. Greenhouse gas forcings (GHGs) may be control variables or not, depending on whether they are part of the prognostic variables. Ozone-aerosols land use and volcanic eruptions would normally be set as certain, and not estimated. That is, for equilibrium simulations the model is left to converge, and the control vector would most commonly a set of the (deterministic) parameters for model physics. This is the more straightforward application.

For transient simulation, with time-evolving forcing, we would not generally include most of the common forcings in the control variables.

In any case, for GHGs, we would generally use the most recent reconstuction by Peter Köhler et al. (2017) more than prognostic GHGs, which reaches until 156 kyr. We could include an error term for these GHGs that would be constant within each DAW (of the order of some hundred years), and estimated as part of the assimilation for subsequent DAWs. Flux correction term can be treated similarly (which we can consider as a parametric way of dealing with model error).

**Rev#2:** 27. Page 5, l.6: "That is, that the system..." −! "That is, the system..."

**Auth:** Done.

**Rev#2:** 28. Page 5, line 23: "in 4D-Var then" −! "in 4D-Var is then"?

**Auth:** Done.

**Rev#2:** 29. Page 5, line 23: "non-linear" −! "non-quadratic"

**Auth:** Yes! Sorry. Done.

**Rev#2:** 30. Page 7, line 6: "is the same that" −! "is the same as"?

**Auth:** Done.

**Rev#2:** 31. Page 7, line 7: "4D-Var, IKS" −! "4D-Var, the IKS"

**Auth:** This sentence has been removed (redundant).

**Rev#2:** 32. Page 7, Eq.(16) and around: Such an operator exists only if the observations are time-averaged values, right? In general observations will depend on the initial condition. This must discussed (this is actually better discussed in the introduction!).

**Auth:** We would say the operator (as the simplified schemes) is applicable under the condition that the observations have are long-time averages. Most of the paleoclimate proxy observations have temporal resolution longer that decadal (some much longer). For example, the simplified approach followed in the describe schemes would not be suited to assimilate coral records (Sr/Ca or d18O) with its full annual resolution (when available) for the last two hundred years (an IEnKS could be used instead). This would result in the initial conditions being too influential in the background state at the observation times (an IEnKS could be used instead to include these in a low-rank formulation). This is now discussed in the introduction.

**Rev#2:** 33. Page 7, line 8: What is a "quasi-equilibrium"?

**Auth:** Clarified as: "(we denote this here as *quasi-equilibrium*). That is, it is possible that the deep ocean circulation still has not converged to its dynamical attractor, but this has a negligible effect on the model climate at the surface."

**Rev#2:** 34. Page 7, line 14: In my opinion, there is no need to introduce a new term. This is just an IKS in parameter space.

**Auth:** We have replaced the pIKS terminology by NLS-IKS. See answer to general comment 3.

**Rev#2:** 35. Page 8, Eq.(17): This type of formulation is frequent in many areas of geosciences; there is no need to look as far as history matching in oil reservoir modelling. For instance, this is very often met in source/fluxes inverse problems in atmospheric chemistry.

**Auth:** We have removed these comments.

**Rev#2:** 36. Page 8, line 13: "While it": A typo?

**Auth:** Yes; 'it' removed.

**Rev#2:** 37. Page 9: In my opinion the discussion on the computation of the sensitivities is not only convoluted but also not very useful. It is obvious to the reader (to me at least), that you will use finite-differences in the end. Essentially only the last paragraph of section 3.2 is needed.

**Auth:** A know drawback in finite differences approximations to sensitivity is the rounding issue, related to the selection of optimal perturbations. If computationally feasible, a regression around a small univariate ensemble helps instead of a single perturbation. As described, we did so in the first experiment (Section 4.3). Still, we have simplified this discussion.

**Rev#2:** 38. Page 9, Eq.(22): The linearisation in parameter space should be carried out at the j-th estimate of the parameters, no the background parameters (except for j=1). What you wrote is just an approximation, which would make the iterative approach not as accurate as expected. Please clarify.

**Auth:** Yes, as indicated later in Eqs. (25) and (30), but agree this is confusing. We have removed the "b" superindex to make this general (as the iterative methods come after this), and clarified the point.

**Rev#2:** 39. Page 10, line 20: "Iterative linear methods" is awkward, even though I guess I understand what you mean.

**Auth:** Rephrased.

**Rev#2:** 40. Page 10, line 24: Parentheses are needed around Bell and Cathey (1993).

**Auth:** Done.

**Rev#2:** 41. Page 10, beginning of section 3.3: I don't see the point in the discussion with the EnRML. You can probably do without it.

**Auth:** Removed

**Rev#2:** 42. Page 11, Eq.(25): The notation is unclear (I understand but many colleagues would not) and should be made consistent with Eq.(22).

**Auth:** Clarified and Eq.(22) modified to be consistent with this. The loop index is dropped (and explained) in the general sensitivity description, and explicitly indicated in the algorithms.

**Rev#2:** 43. Page 11, line 22: Actually the use of the MDA trick is slightly different in Bocquet and Sakov (2014) than in Emerick and Reynolds (2013), because the weights are adjusted over several data assimilation cycles.

**Auth:** Yes, we know both papers. This has been specified.

**Rev#2:** 44. Another reference relevant to your manuscript is a study of the iterative ensemble Kalman smoother applied to a joint state and parameters estimation problem (Bocquet and Sakov, 2013).

**Auth:** Included.

**Rev#2:** 45. Page 12, line 21: "opposite to" −! "as opposed to"

**Auth:** Done.

**Rev#2:** 46. Page 13, line 21: The sentence is a bit ambiguous since the model integration is part of the analysis (and so-to-speak a part of the analysis!). Please reformulate.

**Auth:** Rewritten ("analysis" was used to mean "assimilation"; we understand the ambiguity).

**Rev#2:** 47. Page 13, line 26: What is a "temporal solution"?

**Auth:** We have redone this small section. Actually, more than "truncated" solutions these are "early stopped" iterations, which provide an alternatives (also approximate) solutions.

**Rev#2:** 48. Page 13, line 27: "detect linearity assumption": This expression is unclear. Please rephrase.

**Auth:** Rephrased .

**Rev#2:** 49. Page 13, line 3: I am familiar with the Levenberg-Marquardt scheme(s) and I do not understand your sentence!

**Auth:** We removed the sentence. It is actually wrong (it would be an alternative, not a combination of these two) and not needed. Now, we cite earlier (Section 2.1) schemes combining Gauss-Newton with the multiple assimilation approach (the IenKS of Bocquet and Sakov, 2013, 2014).

**Rev#2:** 50. Page 13, line 12-14: You have to give more details of your implementation. First, I do not see why you would need localisation for the state variables, since you are not updating them. Second, it is well known that, without a few tweaks, one cannot update global parameters in a LETKF.

**Auth:** True. We do not use localization. This was a remainder of former versions in which we did use LETKF for independent state estimate, but this does not apply here. Any reference to localization has been removed, and we have clarified that we use a mean-preserving (or the "spherical simplex") ETKF. See new text and references.

**Rev#2:** 51. Page 14, line 21: "It is not standard, however, how the GA should be applied in the context of DA.": There have been reviews and papers about that; for instance Bertino et al. (2003), as you rightfully mentioned, but also Bocquet et al. (2010); and above all Simon and Bertino (2009) and Béal et al. (2010) who set the standard on this topic. As far as I can understand, you are using their method. Please amend.

**Auth:** Clarified. The section has been expanded according the comments.

**Rev#2:** 52. Page 15, l.9: "adjoint method (4D-Var)" −! "4D-Var (based on the adjoint)".

**Auth:** Done.

**Rev#2:** 53. Page 15, l.11, L.13, l.27: Please avoid the "adjoint method" expression which is really outdated, and not use in data assimilation study. Refer instead to 4D-Var or variational method, possibly mentioning the use of the adjoint model.

**Auth:** Done.

**Rev#2:** 54. Page 15, l.24, "standard 4D-Var applications" −! "standard in 4D-Var applications"?

**Auth:** Done.

**Rev#2:** 55. Page 15, l.29-30: I do not understand the last sentence.

**Auth:** Now splitted in two and merged with a previous paragraph: "…,which we considered as reasonable uncertainty values. Other than the parametric uncertainty we considered a perfect-model framework.".

**Rev#2:** 56. Page 15-16, section 4.1: Where did you describe the parameters and how many are they? This is absolutely key to the feasibility of the problem. There are tables; but the parameter should be more clearly discussed in the text.

**Auth:** We considered sufficient to refer to Paul and Losch (2012) [PL2012], as stated in former l.14, for the model (and parameter) description. We still believe it is not worth to reproduce the description in PL2012, which would make this manuscript longer, considering that in this manuscript this a first test and the experiment with CESM is the main focus. We have now, however, clarified that there are only five (scalar) parameters in this experiment, and included a short description of these parameters, referring to PL2012 as considered adequate and for broader explanation.

**Rev#2:** 57. Page 17, line 4: "Note the original" −! "Note that the original".

**Auth:** Done.

**Rev#2:** 58. Page 17-18, section 4.3: In this section, you keep referring to the "adjoint method". Please do not use this term. This is a loose term, used in a loose way which generates confusion. At best, it refers to the computation of the gradient via the adjoint model, and not to the optimisation method you actually imply. That is why it is not used in written texts of the data assimilation community. You even refers on page 18 to the "adjoint", a short-cut which definitely lacks rigour.

**Auth:** Replaced by "4D-Var". See answer to general comment 4 above.

**Rev#2:** 59. Page 18, lines 1-14: It seems that it all boils down to the presence or absence of a prior for the parameters. Isn't it? If this is so, then this discussion is not really focused on what it should be.

**Auth:** We agree that the comparison with Paul and Losch (2012) [PL2012], in which the regularization term for the parameters was not considered, was far from ideal. We have now conducted a new 4D-Var test using exactly the same cost function as for the other methods in the experiment, so that the benchmarking is now fair. The description of the experiment and results have been updated accordingly. We have also now dropped the $ETKF_{10}$ test (ETKF with $m$=10 members). Considering that the number of integrations in this case is substantially smaller than the rest of the schemes, it is no wonder it does not behave very well. We have left the $ETKF_{60}$, which is computationally more comparable with the iterated schemes in this experiment. Finally, in our previous test, weights given to individual observations in term $J_y$ in the cost function in PL2012 ranged from ~1 for observations close to the Equator to ~0 for observations toward the Poles. We have realised that PL2012, forced then these weights to sum to one, with the net effect that $J_y$ was about five times higher in out case for similar innovations. In the updated version weights sum to one as in PL2012, which leads to higher effect of the regularization term. Ultimately, this makes the (now called) NLS-IKS more stable, which now obtains a lower total cost function value than the (now called) NLS-MKS. This could be expected but now is explicitly quantified. A major result is that the NLS-IKS results in posterior parameters and cost value that are nearly identical than 4D-Var.

**Rev#2:** 60. Page 20, line 9: "multi-component data assimilation": To the best of my knowledge/ understanding, this is rather called "strongly coupled data assimilation".

**Auth:** Replaced by "strongly coupled". The "multi-component data assimilation" term is often used in the ESMs data assimilation context (e.g.; NCAR teams), in this sense. We agree "strongly coupled" is more clearly defined.

**Rev#2:** 61. Page 20-21: I would more precisely enumerate/list/discuss the control variables.

For instance, at some point, clearly mention: "Hence, our first control variable

is..." etc.

**Auth:** Done.

**References (not included in the paper)**

Alll references here are included in the manuscript

---

## Author Response (AR1)

**Manuscript reference: GMD-2018-48**

**Title\*: Evaluation of iterative Kalman smoother schemes for multi-decadal past climate analysis with comprehensive Earth system models**

**Date: 03/10/2018**

\*Modified from the discussion paper

**General Response to Editor**

Dear Editor,

Please find our responses to the reviewers' comments in the attached documents. In addition to specific requests, prompted by reviewers' comments, we have included a number of modifications to the original submission that we consider suited to improve the study.

Prompted by Rev#2's comment about the "parameter-space" terminology (see specific reply), we have changed the main title of the manuscript to:

"Evaluation of iterative Kalman smoother schemes for multi-decadal past climate analysis with comprehensive Earth system models"

The previously prepended "parameter" in the filter names aimed to indicated that these schemes (smoothers) were specifically formulated in the control vector space (with control variables generally being a set of model deterministic physics parameters, forcings and boundary conditions). Assumed simplifications regarding background and model error allow for the numerical estimation of the sensitivity of the dual of the observation space to inputs based on finite differences. The use of this numerical local sensitivity analysis (LSA) with simple Finite Difference sensitivities (FDS) to compose for the Kalman gain, make these schemes different from both Monte Carlo methods –-as the schemes based on EnKF with samples representing the background covariance matrix ---in which "mean sensitivities" are obtained throughout the sampled input space representative of the prior distribution e.g. the batch-EnRML)--- and from standard KF schemes. Let us note that although the IEnKS schemes attempt to obtain a local sensitivity, they do so by deflation and inflation mechanisms, and they are still low-rank schemes designed for high-dimensional control vectors. Still, given Rev#2's comments, we have considered that "finite difference sensitivities" (FDS) based schemes is a more specific and clarifying label. We hope Rev#2 agrees. Further clarification is given in the updated manuscript and in the reply to Rew#2-.

Also, according to our experiment with CESM, we have preferred to use the term "Earth System Models" in the title itself, more than the former "coupled Global Circulation Models". We have modified the manuscript accordingly.

We are sending the responses to reviewers in specific documents.

We believe we have addressed all reviewers' comments adequately. We are so sending our responses to reviewers' comments and an updated manuscript, which we hope you find suited for publication in GMD.

We would like to note two points about the new manuscript: a) Despite the reviewers indicated that the manuscript needed a minor revision, we have felt that, in order to address properly their comments, the manuscript actually would benefit from a thorough rewriting. After the initial modification tracking, the restructuring of paragraphs and sections made the tracking unusable, and were forced to abandon the track of changes. b) The new version includes two appendices. The Copernicus latex template is not able to label properly the Figures and Tables if these are placed in the end after the whole document when appendices

are included (an issue we did not have without appendices in the former version). Thus we have included them in line with the text, so that the numbering is correct.

Best regards,

Javier García-Pintado, and André Paul
* * *
**Manuscript ID:** GMD-2018-48

**Title*:** Evaluation of iterative Kalman smoother schemes for multi-decadal past climate analysis with comprehensive Earth system models

Authors: J García-Pintado, André Paul

**Date:** 05/09/2018

*Modified from the discussion paper

**Responses to Reviewer #1 (anonymous)**

We thank the reviewer for their positive comments, which will help improve the manuscript. Below we give each comment and describe how we are altering the manuscript to address the reviewer's concerns.

Let us note that indirectly prompted by Reviewer # 2, we have considered a new title for the manuscript as indicated above. In general, we have conducted a substantial rewriting of the manuscript, due to (a) specific request from the reviewers to shorten introductory parts and expand the results and discussions, and (b) reviewers' comments also have indirectly suggested us that some parts of the manuscript were in need of further explanation.

Thus the Introduction is now longer and the description of the paleoclimate context has been slightly expanded, but former section 2 (Problem definition) has been now dropped and compacted within the Introduction. The description of the nonlinear relation between the control variables and the observation space in the experiment with the Community Earth System Model (CESM) now receives more attention. The analysis with ETKF-GA (the Gaussian anamorphosis) is now more detailed, and the described scheme of the iterated Kalman smoother is now also included in the CESM experiment. Also, the first experiment, with the 1D energy balance model, has been updated with a new more adequate 4D-Var, as benchmarking scheme.

**Responses**

**Rev#1:**

The authors present innovative low-cost strategies for online model parameter estimation, which can potentially be applied for climate field reconstruction with coupled GCMs. The manuscript is written in a transparent way and authors explain very well all the assumptions used in their study. The paper moves the time-averaged data assimilation efforts a small but very important step forward. I recommend publishing the paper after minor revision.

General Comments:

**Rev#1:**

1) Scientific significant:

The manuscript represent a substantial contribution to modeling science within the scope of this journal. The ideas and methods are original.

2) Scientific quality:

The scientific approach and applied methods are valid and the assumptions are well introduced. However, the results are not discussed in a balanced way and could be improved (see below). The models, technical advances and/or experiments described have the potential to perform calculations leading to significant scientific results.

**Auth:** According to both reviewers, we have done a thorough revision of the manuscript, shortening the introductory sections and expanding the result sections.

**Rev#1:**

3) Scientific reproducibility

The modeling science seems not to be reproducible. I think if the authors share their code, this problem will be solved. However, their methodology is well described and traceable.

**Auth:** We are sharing the code.

**Rev#1:**

4) Presentation quality

The presentation quality is fair and could be improved in a new version. Number of figures presenting the results can be revised.

**Auth:** See answer to general comment 2 above. The updated version involves additional figures showing an example of the nonlinear relation between a model parameter and the dual of the observation space for experiment 2 (CESM experiment) as well as the effect of Gaussian transformation, and an example of sensitivity estimates of T2m and additional atmospheric variables to a model parameter (as example of atmospheric variable, according with comment 23 below).

**Rev#1:**

5) Overall, the manuscript is understandable for experts working in the field but not easy to follow for general readers. There are too many acronyms in the manuscript. Please spell out when possible.

**Auth:** We hope the new version is more clear.

**Rev#1:**

6) Given that this paper belongs to the category of "Development and technical papers"

(see:https://www.geoscientific-model-development.net/about/manuscript_types.html#item2),

I encourage the authors to make their code available online (at least for the 1D experiment). This might help the community very much and improve the code itself. According to the journal policies, the authors have to include the model's version in the title (e.g., Model XXX (version Y)).

**Auth:** We now are sharing the code online for the 1D experiment. We also share the DA code for CESM code under request to the authors. Let us note there is a number of more efficient available DA software based on compiled language (C, F90), as , for example, SANGOMA, EMPIRE or PDAF. Our code is instead for prototyping and research purposes, mostly based on R scripting with netCDF4 as interchange format, and resorting to calls to CDO, and occasionally to bash and perl. Note the computational demand of the assimilation is minor in our case by comparison with the computational demand for the model integrations, with CESM runs in ensemble mode for mutidecadal time spans. We are currently arranging our CESM+DA software as an R package with included documentation, in a way it can be directly usable by the paleoclimate community.

**Rev#1:** 7) Page1Line3 (P1L3): Explain how model's parameters have relationship with proxies!

**Auth:** Explained.

**Rev#1:** 8) P1L6-7 is too complicated!

**Auth:** Removed in the new abstract.

**Rev#1:** 9) P1L6 : Authors mix two approaches: model's parameter estimation (atmosphere and ocean) and fresh water melting parameters estimation. Have they done separate experiments? They describe, the latter might be essential for North Atlantic circulations.

However, there might be nonlinear relationships between these two and they could contaminate each other. Which one impacts the error reduction larger?

**Auth:** The sensitivity experiment for the parameter-space schemes (now renamed as Finite Dfference Sensitivity (FDS) schemes are based on individual sensitivity analysis, with no possibility of cross-influencing the sensitivities. We have now described which parameter has shown a higher sensitivity in the analysis (specifically, the atmosphere parameter cldlfrc_rhminl: minimum relative humidity fro low stable cloud formation). Overall, we have clarified the importance of previous sensitivity analyses for the design of the control vector.

**Rev#1:** 10) P2L12: Could you provide references for that?

**Auth:** We have modified the order of the paragraphs. The reference now given for this is Annan et al. (2005b), which was indicated later in the former manuscript.

**Rev#1:** 11) P2L32-33: How should one do this? Reference?

**Auth:** We give now a few references to sensitivity analysis. Please see updated text.

**Rev#1:** 12) P2L5: Explain the "generally positive results"!

**Auth:** This part of the sentence has been removed. We have considered it is not really needed.

**Rev#1:** 13) P3L20: Aren't the parameters not being updated at each DAW? Aren't they time varying when the new observation is available? How could one do that in future projections without observations? What are the challenges? How could tuning the model for the past improve projections? Please clarify!

**Auth:** Yes. Their estimates vary, but this does not mean that they are time-varying. This has been clarified. We also clarify that the goal is to conduct past climate field reconstruction at long time scales, and the the model is assumed to be previously tuned. The control variables in the assimilation (model deterministic parameters, and other inputs) are used to carry on the responsibility to generate an uncertain background and cope with the overall uncertainty. Thus the estimated control variables at each DAW serve as a mechanism to minimise the cost functions and to obtain a climate filed reconstruction fusing model and data. Differences (or increments) between corresponding tuned parameters (for present day dense datasets) and those estimated by the assimilation based on proxy databases of past climates may serve to diagnose model differences, and be a very useful tool (as opposed to direct updating of the climatic full-field as in standard EnKFs, here there is a physical mechanism in the model explaining the climatic increments resulting from the assimilation). But estimated parameters as part of the paleoclimate assimilation based on proxy data would not in principle be meant to replace the originally tuned model for future projections. We have clarified this in the manuscript.

**Rev#1:** 14) P6L10: Explain briefly the gradient descent algorithm, learning rate, number of iterations, etc…

**Auth:** There are several options for this (conjugate gradient methods in general). Then, tipically 4D-Var uses about 3 inner loops and one (e.g.; UKMO) or two outer loops (e.g.; ECMWF). But we think it is better not to expand too much on this here as it is not central to the manuscript. Currently operational centres have mostly moved to hybrid methods ('En4DVar') and either use ensemble of 4D-Vars (as ECMWF) or use EnKF to get hybrid B matrices, and the scenario of options is rather wide. Alternatively, for the interested reader, we have given the new sentence:

"The current implementation of variational assimilation (with atmospheric models) is now different in each operational NWP center, who have mostly moved to hybrid methods. A

recent review of operational methods of variational and ensemble-variational daa assimilation is given by Bannister (2017)"

**Rev#1:** 15) P8L6-7: However, aren't the parameters updated based on time-averaged obs?

**Auth:** Yes. The estimates of the parameters. Please see response to comment 13.

**Rev#1:** 16) P11L18-19: How do you define the learning rate in gradient descent?

**Auth:** We have removed these lines. The strategy for the (now called) FDS-MKS scheme stepping is described later in the section.

**Rev#1:** 17) P15L19: Why analyzing only 10 years? 90 years for spin up of 1d model?

**Auth:** This follows the protocol in Paul and Losch (2012) to make the experiment more comparable with theirs.

**Rev#1:** 18) P18L27: "not shown", but is interesting to put in supplementary.

**Auth:** Included in supplementary material

**Rev#1:** 19) P19L2 : is it a typical set-up of CESM?

**Auth:** Yes, it is the scientifically validated compset with short name B1850CN, as found in http://www.cesm.ucar.edu/models/cesm1.2/cesm/doc/modelnl/compsets.html. Is is now indicated in the manuscript.

**Rev#1:** 20) P20L9: Could you explain and discuss the problems of multi-component DA in your set-up? Differences in time-scales of proxies, etc.

**Auth:** Please note we have indicated this now by the more standard ("strongly coupled" data assimilation). A brief discussion has been included.

**Rev#1:** 21) P20L13: Have you done other experiments with other sets of parameters? For example more or fewer numbers of parameters?

**Auth:** For other model configuration yes, but not specifically for this study. We understand that additional experiments based on a wider number of error sources (e.g. from biases in other parameters not included in the control vector) would be very illustrative. We comment now on this in the introduction of the new manuscript in reference to error compensation in real applications. We also indicate the study should clearly be expanded in further research.

**Rev#1:** 22) P24L10: In figure 3 it is really hard to see the differences. Maybe centering the colorbar with zero might help. How do you explain that the error reduction is due to DA and not the lack of sensitivity of the model to perturbation of the parameters. For example Figure 4 upper left panel shows that the model is not sensitive to changes of cldfrc_rhminl in Arctic and Antarctic regions.

**Auth:** We have tried to add observation locations to Figures 4 and 5, and this renders the plots too noisy. Also, making colour scales centred around 0 made a bit more difficult to see the patterns in some cases (not in other cases, in which we have followed the reviewer's suggestion). The neatest general solution we have found is to include isolines at level 0 for both Figure 5 and Figure6, and to centre all maps in this experiments at longitude 0. We have also included tick marks matching the figure of the observation locations and the other ones.

**Rev#1:** 23) You focus on the ocean where the observations where assimilated, how about the atmospheric variables? Is there any error reduction happening there? Could you show for example global T2m quantities?24) Figures similar to Fig.4 for other perturbed parameters could be shown in the supplementary. This will clarify the sensitivity of CESM.

**Auth:** We have added sensitivity plots for T2m and other atmospheric variables in the manuscript with respect to the ocean background vertical diffusivity, linking them with the

AMOC Figure. From many other possibilities we have considered this is a nice example, in connection with strongly coupled DA.

**Rev#1:** Specific Comments:

1) P3L6: "in section?"

**Auth:** Corrected.

**Rev#1:** 2) P3L7: "problem of CFR" which problem?

**Auth:** This sentence has been removed.

**Rev#1:** 3) P22L12: How about the uneven time resolution of observations?

**Auth:** We obtain model equivalent of the observations at the resolution of the observation time. This includes, for example, seasonal means or annual means during specific sub-spans of the DAW, which are specific for each proxy type. As the paleoclimate proxies represent an integrated effect longer that the model timestep (e.g. commonly get monthly output from CESM) forward models (proxy system models; PSM) can include this integration effect. In this study we do not discuss the problem of paleoclimate proxy modelling, although now we refer to it more clearly.

**Rev#1:** 4) figure 2, 3, 4: for the results one has to switch between figure 2, 2 and 4 to follow theline of discussions. You could at least put the observation locations on figure 3 and 4. Note that the land-sea mask is also shifted between the figure 2 and 3-4.

**Auth:** We have reorganized the discussion. Please, see also the answer to comment 22 above.

**References (not included in the paper)**

Alll references here are included in the manuscript

**Manuscript reference:** GMD-2018-48

**Title\*:** Evaluation of iterative Kalman smoother schemes for multi-decadal past climate analysis with comprehensive Earth system models

**Date:** 03/10/2018

\*Modified from the discussion paper

**Responses to Reviewer #2 (anonymous)**

We thank the reviewer for their positive comments, which will help improve the manuscript. Below we give each comment and describe how we are altering the manuscript to address the reviewer's concerns.

Let us note that indirectly prompted by Reviewer # 2, we have considered a new title for the manuscript as indicated above. In general, we have conducted a substantial rewriting of the manuscript, due to (a) specific request from the reviewers to shorten introductory parts and expand the results and discussions, and (b) reviewers' comments also have indirectly suggested us that some parts of the manuscript were in need of further explanation.

Thus the Introduction is now longer and the description of the paleoclimate context has been slightly expanded, but former section 2 (Problem definition) has been now dropped and compacted within the Introduction. The description of the nonlinear relation between the control variables and the observation space in the experiment with the Community Earth System Model (CESM) now receives more attention. The analysis with ETKF-GA (the Gaussian anamorphosis) is now more detailed, and the described scheme of the iterated Kalman smoother is now also included in the CESM experiment. Also, the first experiment, with the 1D energy balance model, has been updated with a new more adequate 4D-Var test as benchmark scheme.

**Responses**

**Rev#2:**

This is a very interesting study, supported by a substantial amount of work, both theoretical and numerical. The theoretical and experimental parts are well balanced; the experiments are well chosen and interesting. The manuscript certainly should be considered for publication. Nevertheless, I believe it has a few flaws and requires significant improvements before being acceptable for publication.

In my opinion, the main issues are:

1. The manuscript is too long. Several discussions are unnecessary and frankly a bit wordy, especially in the introductory parts.

**Auth:** We have done a thorough revision of the manuscript, with a substantial number of sentences rephrased and shortened/dropped. Alternative we have expanded the Introduction to give some context of paleoclimate and support the rationale for the assumptions and applicability of the given schemes. Given the comments, we have also expanded the section regarding the tests with Gausian Anamorphosis, and conducted additional tests in both experiments (now including the IKS in the CESM twin case). Please see new manuscript.

**Rev#2:** 2. There are quite a few typos that need to be corrected.

**Auth:** Typos have been corrected as found.

**Rev#2:** 3. I believe that there is no need to give these methods new names just because they are applied to parameters. But this is certainly up to the authors.

**Auth:** In the same sense that the EnKF provides a low-rank representation of the background covariance matrix, making it akin to but different from the KF, the methods used here differ from the standard KF (iterations apart) in that they are explicitly solved as a function of local sensitivity analysis (LSA) with finite difference sensitivity (FDS) experiments. While the IEnKS also uses (in its different versions) a strategy to get local sensitivities (e.g., differing from the average ensemble sensitivities of the EnRML, and the batch-EnRML), it is a low-rank scheme. We feel it is adequate to prepend some tag to "KF" (or "KS" for smoother) to clarify the schemes. Earlier, we chose "parameter-space" as an implicit way to indicate the way sensitivities were constructed. Given the comment, we have preferred to simply use FDS to indicate the specific way in which sensitivites are obtained. On the other hand, to relate better to the "multiple data assimilation" strategy, we have replaced the "F" for fractional by an "M" for multistep in the formerly named pFKS.

With this, the names now used for the schemes are FDS-IKS (finite difference sensitivity, iterative Kalman smoother) and FDS-MKS (finite difference sensitivity, multistep Kalman smoother). Let us insist that the aim is not to claim these are new schemes, but to clarify their specificity. We hope the reviewer agrees with the need for clarification and that some labelling is suited.

**Rev#2:** 4. Please avoid the use of the term "adjoint method" which is – as of today – rarely used, ill-defined or at best does not precisely correspond to 4D-Var. See comments below.

**Auth:** In general, although we explicitly referred to "4D-Var" in Section 3 regarding the methodological description, we chose to use the "adjoint" term in other parts (mostly in Section 4), to comply with the terminology used by Paul and Losch (2012) –--for which this section was considered as an extension---, and by some oceanographers and climate scientists (e.g., regarding terminology, see Wunsch and Heimbach, 2013). We understand, however, the reviewer's point and have now used the "4D-Var" term throughout the manuscript.

**Rev#2:** 5. The synthetic experiments with the CESM is a great piece of work, but unfortunately quite inconclusive. This is unexpected since we could have hoped for clear and neat results with such experiments. Tables 4 and 5 point to uncleared problems in the experiments. It is nice to use an ETKF in conjunction with a Gaussian anamorphosis. However, if the outcome is inconclusive (is it?), then it casts doubts on the interest of such test, or more likely on the implementation of the method (bugs?). See additional comments below (sorry for the redundancy).

**Auth:** The main conclusion of the CESM experiment is that the iterative schemes based on simple finite difference sensitivities (FDS) –-even with non-optimal perturbations--- are more able to deal with the nonlinear relation between the inputs and the observation space that a single linear step (in the experiment), even if this is based on a denser sampling of the prior input PDF (as the EnKF) and in our tests. We acknowledge and agree that more detailed experimentation is needed.

We have not found any bug in the Gaussian Anamorphosis (GA) as implemented. However, given the comment we felt that some clarification was needed. We have so provided further explanation on the specific GA analysis, including the justification of the tests. Also, we have included additional plots regarding the transformations. Hopefully, these are informative about the non-linear relation between inputs and SST. The plots also support an explanation about why the GA (as applied) was not completely successful despite general increase in univariate Gaussianity. Irrespective of the success of the GA experiment, we consider it is worth reporting the results as support to guide/encourage further possible experimentation.

**Rev#2:** 6. It would be worth introducing in the CESM synthetic experiment some additional model error that you do not control in order to check how the methods are compensating for this error. This would be realistic and convincing.

**Auth:** We agree. For real applications, the selection of the (computationally feasible) control variables should be done based in sensitivity analysis. The control variables are responsible in the assimilation for all model errors, including compensation of model biases (in the tuned model) elsewhere. We have added a paragraph in the introduction to clarify this and that the purpose of included uncertain (deterministic) parameters in the control variables here is not to tune the model. The model is assumed to be previously tuned. A subset of the more "sensitive" but possibly uncertain parameters should be chosen as control vector to generate the background state needed for the assimilation. The purpose if the schemes is to produce mean climate field reconstructions for past climates at long scales. Differences between updated parameters and "tuned" values can be evaluated to diagnose the possible reasons for these. This is now clarified in the introduction.

Regarding the experiments, as a step-by-step approach we chose a scenario in which all sources of uncertainty were included in the control vector (a subset of model parameters, plus a freshwater flux term from Greenland and forcing from greenhouse gases). We understand this is the first time a fully coupled CESM is evaluated (even for a identical twin experiment) for the assimilation of data from a past climate multiproxy database (the MARGO Last Glacial Maximum in this case), and the experiments conducted have taken a considerable computing effort (possible thanks to HLRN III, the North Germany HPC). We have included some further test so that the IKS is now also evaluated with the CESM twin experiment. But it is not feasible for us to expand the analyses within the scope of this manuscript. We are looking forward for additional experiments (by ourselves or other colleagues in the paleoclimate community). The evaluation of specific error compensations is for sure one of the tasks ahead.

**Rev#2:** 7. Why is the data assimilation code not available? I thought it would be mandatory to do so for GMD, is it?

**Auth:** We are making the DA and Ebm1D codes available. CESM v1.2 is already available.

**Rev#2:** List of remarks and suggestions, some pertaining to the main criticisms:

1. page 1, l.4-5: "In a model framework where we assume that model dynamic parameters account for (nearly) all forecast errors at observation times,": Right, but is this framework usually met?

**Auth:** Deterministic Earth System models (ESMs) converge to their own climatology and the memory of (reasonable) initial conditions is lost after some integration time (from a few years to some decades). This is now further discussed in the manuscript.

**Rev#2:** 2. page 1, 12: "are evaluated in numerical experiments": Are these twin/synthetic experiments? In other words do you use real observations or synthetic ones? It is necessary to mention it here in the abstract.

**Auth:** Clarified that the first experiment uses "present-day surface air temperature from the NCEP/NCAR reanalysis data as target" and the second one (our object of study) is a synthetic experiment with the Community Earth System Model (CESM v1.2).

**Rev#2:** 3. Page 1, l.14: "the pFKS obtains a cost function": This expression seems meaningless. Please rephrase.

**Auth:** Removed.

**Rev#2:** 4. Page 1, l.14: the expression "adjoint method" should be avoided as it is not well defined.

**Auth:** Replaced by "4D-Var".

**Rev#2:** 5. Page 1, l.14-15: Frankly, the whole sentence "Firstly, with Ebm1D the pFKS...behaves slightly worse." is difficult to understand, especially in an abstract. (For me, the technical terms are not the problem, since I am fluent in them.)

**Auth:** Rewritten.

**Rev#2:** 6. Page 1, l.17: You have to explain in the abstract why you would use an ETKF with a Gaussian anamorphosis or not mention it at all.

**Auth:** Explained.

**Rev#2:** 7. Page 1, l.18: Having the lowest cost function value is rarely a criterion as it depends much on the prior used in the cost function.

**Auth:** Cost functions at each experiment use the same prior for each scheme. We indicate now that we focus here on the analysis step.

**Rev#2:** 8. Page 1, l.21: "The issue of fusing data into models arises in all scientific areas that enjoy a profusion of data.": Not really. This is specific to areas where costly models are used!

**Auth:** Modified to "...in scientific areas that enjoy a profusion of data and use costly models."

**Rev#2:** 9. Page 1, l.23-24: "Such methods can be considered as an approach for interpolating or smoothing a data set in space and time where a model acts as a dynamical constraint (Evensen, 1994a)": I don't believe you should use such outdated comment, all the more since nowadays there is a general consensus on a Bayesian view on data assimilation/inverse problems.

**Auth:** We do not see why this comment by Evensen (1994a), which is a point of view, clashes with the Bayesian perspective. It is often echoed with similar wording in recent DA literature, while acknowledging the Bayesian view. In any case, in a now shortened introduction we have rewritten the paragraph and indicated now the Bayesian view of DA methods.

**Rev#2:** 10. Page 2, l.16-19: "Other geophysical applications share this relevance of model parameters on the assimilation problem, as the estimation of distributed parameters and state for multiphase flow in petroleum reservoirs (e.g.; Gu and Oliver, 2007; Oliver et al., 2011), or hydraulic tomography for groundwater applications (e.g.; Schöniger et al., 2012).": You should mention atmospheric chemistry first, all the more since it quite close to climate (e.g., Bocquet et al., 2015).

**Auth:** We have now given a short introduction to the context of Earth system modelling of climate, which is more relevant. Then, to shorten the manuscript, we have decided to remove the complete reference, which is more distant to the manuscript.

**Rev#2:** 11. Page 2, l.20: "A related issue is the enforcement of physically based conservation laws, which by default is not taken into account by (ensemble) Kalman filters." No! You are right in general, but all linear constraints are properly enforced. (Which is why the use of the EnKF is widely spread!)

**Auth:** Yes, but this is exactly the justification for the cited work of Janjic et al (2014) and other work, who deal with the incorporation of constraints in the EnKF to preserve mass, angular momentum and energy. Still, we have removed the comment to shorten the manuscript.

**Rev#2:** 12. Page 2, l.23: ";"

**Auth:** Done.

**Rev#2:** 13. Page 2, l.23: "confirming re-integration": This is unclear to me. Please clarify.

**Auth:** Clarified.

**Rev#2:** 14. Page 2, l.30: "under the assumption the errors" −! "under the assumption that the errors"

**Auth:** Modified.

**Rev#2:** 15. Page 3, l.3: "conduct" −! "conducted"

**Auth:** Done.

**Rev#2:** 16. Page 3, l.6: "in section ,": Section number is missing.

**Auth:** Corrected.

**Rev#2:** 17. Page 3, l.9: "adjoint method": please avoid this expression. It does not correspond to anything rigorous.

**Auth:** Changed to "4D-Var".

**Rev#2:** 18. Page 3, l.17: "opposed" −! "as opposed"

**Auth:** Corrected.

**Rev#2:** 19. Page 3, l.17-18: This is an outdated view. Today, it is considered a doable task to estimate uncertainty within a variational framework (this is actually operational at the ECMWF). Read for instance Bousserez et al. (2015).

**Auth:** The point here was to indicate B is 4D-Var operational implementation is not evolving. The hybrid methods ---En4DVar----in operational centers (e.g. ECMWF, UKMO, GMAO, Meteo-France), use an ensemble, in several ways (as ensemble of 4Dvars, etc.),  for the flow-dependent term of B. We have removed the comment in any case.

**Rev#2:** 20. Page 3, l.24-25: "Other than that the formulation is identical than it would be for the corresponding filtering versions.": unclear or awkward.

**Auth:** Removed in new version.

**Rev#2:** 21. Page 3, l.31: Twin experiments? This should be mentioned here as well.

**Auth:** Done

**Rev#2:** 22. Page 4, l.2: "the not only" −! "not only"

**Auth:** Done.

**Rev#2:** 23. Page 4, l.18-19: "We also assume that the model is weakly nonlinear, such that it can be linearized.": This is not a clear statement. Any smooth model (even very nonlinear ones!) can be linearised.

**Auth:** This referred to the sense used in pp.65 in Tarantola (2005), where discussing various degrees on nonlinearity he refers to "forward equations that cannot be linearized, so the a posteriori probability density may be far from a Gaussian and special methods must be used...". In any case, we have removed this comment and the complete Section to shorten the manuscript as requested.

**Rev#2:** 24. Page 4, l.20: "small": do you mean low-dimensional?

**Auth:** Yes. Modified.

**Rev#2:** 25. Page 4, l.28: "The problem is to fit three spatial dimensions in time.": the sentence is unclear. "in" −! "and"?

**Auth:** Modified to: "The problem is to estimate the state of a past climate state along a time window for multidecadal and longer time scales."

**Rev#2:** 26. Page 5, l.1: Assuming time-invariant system is very restrictive in climate models where most forcings are time-dependent. Please justify.

**Auth:** We do not assume a time-invariant system, but that the forcing by radiative constituents can possibly be fixed to a specific calendar year to simulate relatively stable past climate conditions (e.g. Last Glacial Maximum). As we think this is a misunderstanding, we have clarified the two scenarios commonly studied in the paleoclimate community in this sens (i.e. the equilibrium and the transient forcing). Please see new section 2.1, and new specific comment within the experiment with CESM in section 4.

**Rev#2:** 27. Page 5, l.6: "That is, that the system..." −! "That is, the system..."

**Auth:** Done.

**Rev#2:** 28. Page 5, line 23: "in 4D-Var then" −! "in 4D-Var is then"?

**Auth:** Done.

**Rev#2:** 29. Page 5, line 23: "non-linear" −! "non-quadratic"

**Auth:** Yes! Sorry. Done.

**Rev#2:** 30. Page 7, line 6: "is the same that" −! "is the same as"?

**Auth:** Done.

**Rev#2:** 31. Page 7, line 7: "4D-Var, IKS" −! "4D-Var, the IKS"

**Auth:** This sentence has been removed (redundant).

**Rev#2:** 32. Page 7, Eq.(16) and around: Such an operator exists only if the observations are time-averaged values, right? In general observations will depend on the initial condition. This must discussed (this is actually better discussed in the introduction!).

**Auth:** This is now discussed here in much more detail. Please see new text.

**Rev#2:** 33. Page 7, line 8: What is a "quasi-equilibrium"?

**Auth:** Clarified in the new discussion.

**Rev#2:** 34. Page 7, line 14: In my opinion, there is no need to introduce a new term. This is just an IKS in parameter space.

**Auth:** We have replaced the pIKS terminology by FDS-IKS (FDS for finite difference sensitivities), only to clarify the way that the scheme is expressed and the sensitivities are obtained. See answer to general comment 3 and new manuscript.

**Rev#2:** 35. Page 8, Eq.(17): This type of formulation is frequent in many areas of geosciences; there is no need to look as far as history matching in oil reservoir modelling. For instance, this is very often met in source/fluxes inverse problems in atmospheric chemistry.

**Auth:** We have removed these comments.

**Rev#2:** 36. Page 8, line 13: "While it": A typo?

**Auth:** Yes; 'it' removed.

**Rev#2:** 37. Page 9: In my opinion the discussion on the computation of the sensitivities is not only convoluted but also not very useful. It is obvious to the reader (to me at least), that you will use finite-differences in the end. Essentially only the last paragraph of section 3.2 is needed.

**Auth:** A know drawback in finite differences approximations to sensitivity is the rounding issue, related to the selection of optimal perturbations. We have reduced the discussion of the sensitivities, now focused on clarifying how the ensemble sensitivities from the ensemble backround for the ETKF and the FDS are obtained, which is relevant for the plots in experiment 2.

**Rev#2:** 38. Page 9, Eq.(22): The linearisation in parameter space should be carried out at the j-th estimate of the parameters, no the background parameters (except for j=1). What you wrote is just an approximation, which would make the iterative approach not as accurate as expected. Please clarify.

**Auth:** Yes, as indicated later in Eqs. (25) and (30), but agree this is confusing. We have removed the "b" superindex to make this general (as the iterative methods come after this), and clarified the point.

**Rev#2:** 39. Page 10, line 20: "Iterative linear methods" is awkward, even though I guess I understand what you mean.

**Auth:** Rephrased.

**Rev#2:** 40. Page 10, line 24: Parentheses are needed around Bell and Cathey (1993).

**Auth:** Done.

**Rev#2:** 41. Page 10, beginning of section 3.3: I don't see the point in the discussion with the EnRML. You can probably do without it.

**Auth:** Removed

**Rev#2:** 42. Page 11, Eq.(25): The notation is unclear (I understand but many colleagues would not) and should be made consistent with Eq.(22).

**Auth:** Clarified and Eq.(22) modified to be consistent with this. The loop index is dropped (and explained) in the general sensitivity description, and explicitly indicated in the algorithms.

**Rev#2:** 43. Page 11, line 22: Actually the use of the MDA trick is slightly different in Bocquet and Sakov (2014) than in Emerick and Reynolds (2013), because the weights are adjusted over several data assimilation cycles.

**Auth:** Yes, we know both papers. This has been specified.

**Rev#2:** 44. Another reference relevant to your manuscript is a study of the iterative ensemble Kalman smoother applied to a joint state and parameters estimation problem (Bocquet and Sakov, 2013).

**Auth:** Included.

**Rev#2:** 45. Page 12, line 21: "opposite to" −! "as opposed to"

**Auth:** Done.

**Rev#2:** 46. Page 13, line 21: The sentence is a bit ambiguous since the model integration is part of the analysis (and so-to-speak a part of the analysis!). Please reformulate.

**Auth:** Rewritten ("analysis" was used to mean "assimilation"; we understand the ambiguity).

**Rev#2:** 47. Page 13, line 26: What is a "temporal solution"?

**Auth:** We have redone this small section. Actually, more than "truncated" solutions these are "early stopped" iterations. Please see new section.

**Rev#2:** 48. Page 13, line 27: "detect linearity assumption": This expression is unclear. Please rephrase.

**Auth:** Rephrased .

**Rev#2:** 49. Page 13, line 3: I am familiar with the Levenberg-Marquardt scheme(s) and I do not understand your sentence!

**Auth:** We removed the sentence. It is actually wrong (it would be an alternative, not a combination of these two) and not needed.

**Rev#2:** 50. Page 13, line 12-14: You have to give more details of your implementation. First, I do not see why you would need localisation for the state variables, since you are not updating them. Second, it is well known that, without a few tweaks, one cannot update global parameters in a LETKF.

**Auth:** True. We do not use localization. This was a remainder of former versions in which we did use LETKF for independent state estimate, but this does not apply here. Any reference to localization has been removed, and we have clarified that we use a mean-preserving (or the "spherical simplex") ETKF. See new text and references.

**Rev#2:** 51. Page 14, line 21: "It is not standard, however, how the GA should be applied in the context of DA.": There have been reviews and papers about that; for instance Bertino et al. (2003), as you rightfully mentioned, but also Bocquet et al. (2010); and above all Simon and Bertino (2009) and Béal et al. (2010) who set the standard on this topic. As far as I can understand, you are using their method. Please amend.

**Auth:** Clarified. The section has been expanded according the comments.

**Rev#2:** 52. Page 15, l.9: "adjoint method (4D-Var)" −! "4D-Var (based on the adjoint)".

**Auth:** Done.

**Rev#2:** 53. Page 15, l.11, L.13, l.27: Please avoid the "adjoint method" expression which is really outdated, and not use in data assimilation study. Refer instead to 4D-Var or variational method, possibly mentioning the use of the adjoint model.

**Auth:** Done.

**Rev#2:** 54. Page 15, l.24, "standard 4D-Var applications" −! "standard in 4D-Var applications"?

**Auth:** Done.

**Rev#2:** 55. Page 15, l.29-30: I do not understand the last sentence.

**Auth:** Now splitted in two and merged with a previous paragraph: "…,which we considered as reasonable uncertainty values. Other than the parametric uncertainty we considered a perfect-model framework.".

**Rev#2:** 56. Page 15-16, section 4.1: Where did you describe the parameters and how many are they? This is absolutely key to the feasibility of the problem. There are tables; but the parameter should be more clearly discussed in the text.

**Auth:** We considered sufficient to refer to Paul and Losch (2012) [PL2012], as stated in former l.14, for the model (and parameter) description. We still believe it is not worth to reproduce the description in PL2012, which would make this manuscript longer, considering that in this

manuscript this a first test and the experiment with CESM is the main focus. We have now, however, clarified that there are only five (scalar) parameters in this experiment, and included a short description of these parameters, referring to PL2012 as considered adequate and for broader explanation.

**Rev#2:** 57. Page 17, line 4: "Note the original" −! "Note that the original".

**Auth:** Done.

**Rev#2:** 58. Page 17-18, section 4.3: In this section, you keep referring to the "adjoint method". Please do not use this term. This is a loose term, used in a loose way which generates confusion. At best, it refers to the computation of the gradient via the adjoint model, and not to the optimisation method you actually imply. That is why it is not used in written texts of the data assimilation community. You even refers on page 18 to the "adjoint", a short-cut which definitely lacks rigour.

**Auth:** Replaced by "4D-Var". See answer to general comment 4 above.

**Rev#2:** 59. Page 18, lines 1-14: It seems that it all boils down to the presence or absence of a prior for the parameters. Isn't it? If this is so, then this discussion is not really focused on what it should be.

**Auth:** We agree that the comparison with Paul and Losch (2012) [PL2012], in which the regularization term for the parameters was not considered, was far from ideal. We have now conducted a new 4D-Var test using exactly the same cost function as for the other methods in the experiment, so that the benchmarking is now fair. The description of the experiment and results have been updated accordingly. We have also now dropped the $ETKF_{10}$ test (ETKF with $m$=10 members). Considering that the number of integrations in this case is substantially smaller than the rest of the schemes, it is no wonder it does not behave very well. We have left the $ETKF_{60}$, which is computationally more comparable with the iterated schemes in this experiment. Finally, in our previous test, weights given to individual observations in term $J_y$ in the cost function in PL2012 ranged from ~1 for observations close to the Equator to ~0 for observations toward the Poles. We have realised that PL2012, forced then these weights to sum to one, with the net effect that $J_y$ was about five times higher in out case for similar innovations. In the updated version weights sum to one as in PL2012, which leads to higher effect of the regularization term. Ultimately, this makes the (now called) FDS-IKS more stable, which now obtains a lower total cost function value than the (now called) FDS-MKS. This could be expected but now is explicitly quantified. A major result is that the FDS-IKS converges to the same minimum (although faster in this test) than 4D-Var.

**Rev#2:** 60. Page 20, line 9: "multi-component data assimilation": To the best of my knowledge/ understanding, this is rather called "strongly coupled data assimilation".

**Auth:** Replaced by "strongly coupled". The "multi-component data assimilation" term is often used in the ESMs data assimilation context (e.g.; NCAR teams), in this sense. We agree "strongly coupled" is more clearly defined.

**Rev#2:** 61. Page 20-21: I would more precisely enumerate/list/discuss the control variables.

For instance, at some point, clearly mention: "Hence, our first control variable

is..." etc.

**Auth:** Done.

**References (not included in the paper)**

Alll references here are included in the manuscript

---

## Author Response (AR2)

**Manuscript reference: GMD-2018-48**

**Title\*: Evaluation of iterative Kalman smoother schemes for multi-decadal past climate analysis with comprehensive Earth system models**

**Date: 21/11/2018**

\*Modified from the discussion paper

**General Response to Editor**

Dear Editor,

In response to the requested technical corrections, we are uploading a revised manuscript. As requested, the new manuscript has new colour scales for figures, and a revised 'code availability' section. Our software for the manuscript is now freely available subject to open source licenses via Github. It is composed of four different pieces of software, and their corresponding DOIs have been obtained via Zenodo. Code DOIs are now also included as additional references. Also, we have found and corrected a few remaining grammar errors.

As in the previous version, the latex compiler only indicates correct labelling of Figures and Tables if these are placed in line at the end of the document, before the appendices, as we have done.

Best regards,

Javier García-Pintado, and André Paul